# Beyond Pass@1: Self-play with Variational Problem Synthesis Sustains RLVR

**Xiao Liang**[1]*, **Zhongzhi Li**[3]*, **Yeyun Gong**[2]†, **Yelong Shen**[2], **Ying Nian Wu**[1],
**Zhijiang Guo**[4,5]†, **Weizhu Chen**[2]†
[1]University of California, Los Angeles    [2]Microsoft
[3]School of Artificial Intelligence, Chinese Academy of Sciences
[4]Hong Kong University of Science and Technology
[5]Hong Kong University of Science and Technology (Guangzhou)

## Abstract

Reinforcement Learning with Verifiable Rewards (RLVR) has recently emerged as a key paradigm for post-training Large Language Models (LLMs), particularly for complex reasoning tasks. However, standard RLVR training has been shown to improve *Pass@1* performance at the expense of policy entropy, leading to reduced generation diversity and limiting the *Pass@k* performance, which typically represents the upper bound of LLM reasoning capability. In this paper, we systematically analyze the policy's generation diversity from the perspective of training data and find that augmenting and updating training problems helps mitigate *entropy collapse* during training. Based on these observations, we propose an online **S**elf-play with **V**ariational problem **S**ynthesis (**SvS**) strategy for RLVR training, which uses the policy's correct solutions to synthesize variational problems while ensuring their reference answers remain identical to the originals. This self-improving strategy effectively preserves policy entropy during training and substantially improves *Pass@k* compared with standard RLVR, sustaining long-term improvements and achieving absolute gains of **18.3%** and **22.8%** in *Pass@32* performance on the competition-level AIME 24 and AIME 25 benchmarks. Experiments on 12 reasoning benchmarks across varying model sizes from 3B to 32B consistently demonstrate the generalizability and robustness of SvS. The code is available at https://github.com/MasterVito/SvS.

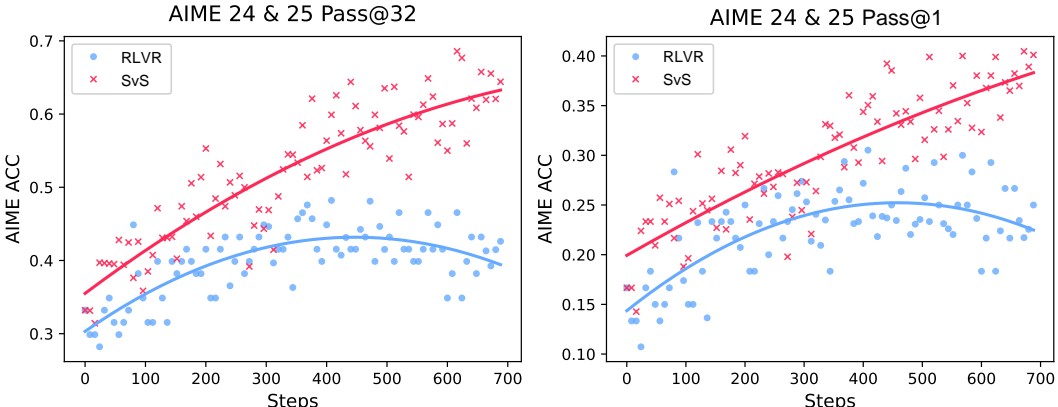

Figure 1: We train Qwen2.5-32B-Instruct on the DAPO-17k dataset using our SvS strategy and standard RLVR. SvS achieves superior efficiency and effectiveness on competition-level AIME benchmarks, showing significant improvements in *Pass@32* and *Pass@1* (average 32 times) scores.

---

*Equal contribution. Work done during Xiao's and Zhongzhi's internships at Microsoft.

†Corresponding authors: Yeyun Gong, Zhijiang Guo, and Weizhu Chen. ✉: yegong@microsoft.com; zhijiangguo@hkust-gz.edu.cn; wzchen@microsoft.com

# 1   INTRODUCTION

The reasoning capabilities of Large Language Models (LLMs) have been significantly enhanced by Reinforcement Learning with Verifiable Rewards (RLVR; Guo et al. 2025a). However, recent studies (Yue et al., 2025; Cui et al., 2025b) have shown that standard RLVR training, such as GRPO (Shao et al., 2024) optimization, may diminish the generation diversity of the policy model, enhancing sampling efficiency and *Pass@1* performance at the expense of output richness, thereby failing to improve *Pass@k* over the base model. In RLVR, training entropy is used to quantify the diversity of model outputs (Cui et al., 2025b; Zhu et al., 2025; Cheng et al., 2025), while improvements in *Pass@k* indicate more advanced exploration. Together, these metrics reflect the model's potential to continue improving in RLVR training. When training entropy collapses to zero, the policy tends to produce homogeneous solutions to training problems, thus losing the opportunity to explore more advanced reasoning trajectories and causing *Pass@k* performance to plateau. Ultimately, the *Pass@1* score also plateaus due to the lack of further exploration opportunities. Therefore, maintaining training entropy and ensuring *Pass@k* improvement are both critical factors for sustainable RLVR training. Recent works such as (Cui et al., 2025b; Li et al., 2025b) also show that regulating policy entropy can enhance rollout diversity, thereby improving RLVR training.

The primary cause of *entropy collapse* and *plateaued Pass@k* is RLVR training on limited problems, where the policy is easily rewarded for repeatedly generating memorized correct solutions—a behavior akin to "hacking" the RLVR training. Intuitively, maintaining policy entropy and generation diversity requires using a broad and diverse range of problems, or entirely new problems in each training step. However, collecting large problem sets with verifiable answers for RLVR is non-trivial. High-quality, human-annotated problem sets are scarce and may not align with the strong reasoning capabilities of modern LLMs (Cobbe et al., 2021; Hendrycks et al., 2021). While synthetic data is a common alternative (Yu et al., 2023; Huang et al., 2024; Liang et al., 2025), a critical limitation is the absence of precise reference answers, which are difficult to derive. These challenges naturally raise the question: *Can we develop a simple yet effective problem augmentation strategy that maintains sustainable data diversity, aligns with the model's capabilities, and ensures accurate labeled answers?*

To answer this question, we propose an online **S**elf-play with **V**ariational problem **S**ynthesis (**SvS**) strategy for RLVR training, where the policy model is prompted to generate *variational problems* based on its correct solutions to challenging and underperforming training-set problems. The rationale for augmenting only the challenging problems is to efficiently target the policy's weakest capabilities (Liang et al., 2025). Since the correct solutions must capture all essential information from the original problems, the policy is naturally encouraged to produce variational problems with rephrased descriptions and structures while preserving the original semantics. Most importantly, the variational problems should share the same reference answers as the original ones, ensuring precision and eliminating the need for additional labeling computation. After synthesis, the policy model is prompted to solve its self-generated variational problems, and the consistency between its produced answers and the reference answers of the corresponding original problems serves to validate the correctness of the variational problems. Finally, the solutions to original problems, the self-generated variational problems, and the solutions to variational problems are gathered for policy updating, enabling it to jointly learn both problem solving and problem synthesis. Notably, the SvS framework relies exclusively on the policy model itself, without any external guidance or distillation, achieving all improvements through end-to-end self-improvement. Moreover, the SvS augmentation is agnostic to RLVR optimization algorithms and can be flexibly incorporated into other methods, such as PPO (Schulman et al., 2017), GSPO (Zheng et al., 2025) and Reinforce++ (Hu et al., 2025a).

To validate the effectiveness and generalizability of SvS, we conduct experiments on LLMs ranging from 3B to 32B and evaluate their performance across 12 widely used reasoning benchmarks. The results show that SvS consistently outperforms standard RLVR across all model sizes and benchmark levels, achieving an average absolute improvement of approximately 3% over the baseline in all experiments. Thanks to the online data updating strategy, SvS training consistently maintains policy entropy within a stable range without noticeable decline or explosion, indicating more sustainable training and prolonged self-improvement. Most importantly, SvS achieves substantial gains of **18.3%** and **22.8%** in *Pass@32* on AIME 24 and AIME 25 (MAA, b), where the standard RLVR shows little improvement. Experiments in Section 5.2 and results in Table 1 provide a detailed demonstration that SvS achieves scalable *Pass@k* improvements across four authoritative benchmarks, highlighting that

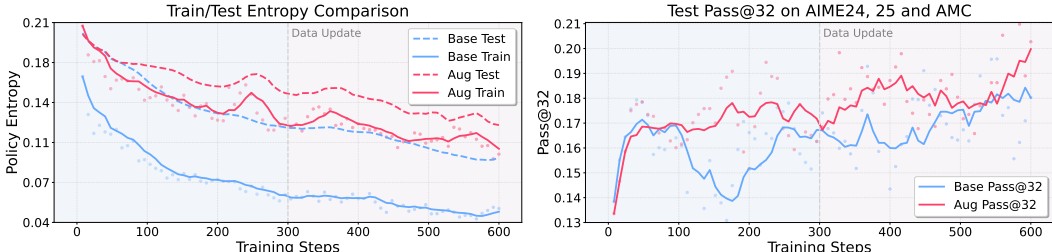

Figure 2: Policy entropy and *Pass@k* during RLVR training under different data strategies. The dashed line indicates policy entropy on evaluated competition-level benchmarks in the right figure. The augmented problems in the *Aug* experiment are updated at the 300th step. All curves are smoothed with a window size of 5, and the original data points are marked with faint dots.

our framework can significantly extend the model's reasoning boundaries (Yue et al., 2025). We also provide a comprehensive study of SvS from multiple dimensions in Section 5.3 and Appendix F.

Our **contributions** can be summarized as:

**(i)** We propose an online **S**elf-play with **V**ariational problem **S**ynthesis (SvS) strategy for RLVR training, where the policy's correct solutions for underperforming training samples are used to synthesize variational problems without additional answer labeling, enabling self-improvement without any external guidance or distillation. **(ii)** The variational problems synthesizing in SvS supports online data augmentation, thereby maintaining stable policy entropy and output diversity during training and improving overall performance, particularly in *Pass@k* on competition-level benchmarks. **(iii)** Extensive experiments across models of varying sizes, together with evaluations on a wide range of benchmarks and additional analyses, demonstrate the generalizability of our proposed SvS.

## 2 RETHINKING THE ENTROPY–PERFORMANCE TRADE-OFF IN RLVR

Recent study (Cui et al., 2025b) demonstrates a trade-off between policy entropy and model performance, where gains in test accuracy come at the expense of response diversity. Specifically, when using a fixed RL training set without entropy intervention, the policy's performance improves over time while its entropy steadily degrades, with the two variables exhibiting a logarithmic relationship: Performance $= -a \exp(\text{Entropy} + b)$. Meanwhile, Yue et al. (2025) shows that RLVR training improves *Pass@k* on evaluation benchmarks only when $k$ is small, with no further gains when $k$ scales to tens or thousands. This suggests that standard RLVR training narrows the reasoning trajectory toward most reward-prone solutions, reducing exploration capacity without fostering more general or advanced reasoning beyond that of base models.

When the policy is iteratively trained on a limited problem set, it tends to memorize specific correct solutions and repeatedly produce similar correct trajectories to obtain positive rewards, leaving less and less room for improvement as training progresses. Intuitively, increasing training data diversity and incorporating online updates can help mitigate policy entropy collapse during training. If each iteration involves different problems, the policy is forced to continually explore optimal solutions to new challenges rather than repeating high-reward solutions from previously seen problems, which promotes continuous exploration of advanced reasoning strategies and enables sustainable learning.

To explore how data diversity affects policy entropy and performance, we conducted experiments using RLVR to train the same policy model with different data strategies. We demonstrate the policy entropy and *Pass@k* scores during training in Figure 2. The blue line shows results on the MATH-12k (Hendrycks et al., 2021) dataset throughout training, while the orange line begins with a mixture of MATH-12k and 36k rephrased problems from MetaMath (Yu et al., 2023); at the 300th step, the rephrased problems are updated with similar ones. Notably, augmented training sets consistently slow the decline of policy entropy for both training and test problems. Furthermore, when the training data is updated at the 300th step, policy entropy stops decreasing and begins to rise, indicating that the policy is re-exploring new reasoning patterns and thereby sustaining learning. Concurrently, evaluation results illustrate that training with an augmented and periodically updated problem set consistently improves *Pass@32* performance, particularly near the update steps.

> **Takeaways for Problem Diversity in RLVR**
>
> - **Impact of Problem Diversity on Entropy** (*Figure 2, left*): Adding augmented problems with diverse formulations, even when the knowledge and domains are close to the originals, can effectively counteracts the entropy drop during RLVR training.
> - **Impact of Problem Diversity on *Pass@k*** (*Figure 2, right*): Diverse problems significantly improve *Pass@k* during RLVR training compared to vanilla problems.

Although effective, rephrasing-based augmentation has notable limitations. Rephrased problems generated by external LLMs may introduce semantic inconsistencies, thereby compromising the accuracy of reference answer annotations and undermining the training stability. Moreover, since rephrasings often use the original problem as context, their diversity cannot be guaranteed. Based on our preliminary experiments, the limitations of rephrasing-based augmentation, and recent studies (Wen et al., 2025; Chen et al., 2025; Liang et al., 2025) advocating the selection of problems appropriate to the model's capabilities, we conclude that *ideal data augmentation for RLVR should be iterative, provide precise reference answers, and be aligned with the policy's capabilities.*

To this end, we propose the **S**elf-play with **V**ariational problem **S**ynthesis (SᴠS) strategy for RLVR training, which features targeted online problem augmentation and a pure self-improvement paradigm. This strategy augments training problems using the policy's correct solutions to underperforming problems, ensuring that the golden answers of synthetic problems precisely match the originals. Sections 3, 4, and 5 present the framework, experiments, and detailed analysis, respectively.

## 3 METHOD

### 3.1 OVERVIEW FOR SᴠS

To achieve the ideal data augmentation for RLVR as discussed in Section 2, we propose the SᴠS framework, which uses the policy itself to online augment training problems through self-play, leading to self-improvement. The policy synthesizes variational problems from its correct solutions to underperforming training set problems and then attempts to solve these synthetic problems. Ideally, these variational problems preserve the semantics and, crucially, the reference answers of the original ones, while their structures and descriptions may differ significantly, thereby eliciting novel or diverse reasoning strategies from the policy.

Specifically, as shown in Figure 3 and Algorithm 1, the full online augmented training batch at each step $t$ comprises three components: (1) **Original Problem Solving**: The policy generates solutions to training set problems, with the underperforming ones retained for augmentation. (2) **Variational Problem Synthesis**: The correct responses containing full information of the underperforming problems are used as context to synthesize variation problems for online training data augmentation. (3) **Synthetic Problem Solving**: The policy is prompted to solve the self-synthesized variational problems, which share the same reference answers as the original ones. Following strategic filtering and reward shaping, the three types of training data are mixed for policy updating.

### 3.2 SELF-PLAY WITH VARIATIONAL PROBLEM SYNTHESIS

Each experience collection step in SᴠS training alternates between problem solving and problem synthesis, enriching the training data buffer **B** online throughout the RLVR iterations. Without any external guidance or distillation, the policy independently generates and solves its synthetic variational problems in a self-improving paradigm.

**Original Problem Solving**. At the beginning of each RLVR iteration, the policy $\pi_\theta$ is prompted to solve problems sampled from the original training set $\mathcal{D}$. For each sampled problem-answer pair $(x, a)$ in $\mathcal{D}$, the policy $\pi_\theta$ generates a group of $G$ solutions $\{y_i\}_{i=1}^{G}$. The correctness reward $\mathbf{R}_\mathrm{c}$ for each response $y_i$ is determined by its consistency with the ground truth answer $a$:

$$\mathbf{R}_\mathrm{c}(y_i, a) = \mathbb{I}(\mathrm{Extract}(y_i) = a) \tag{1}$$

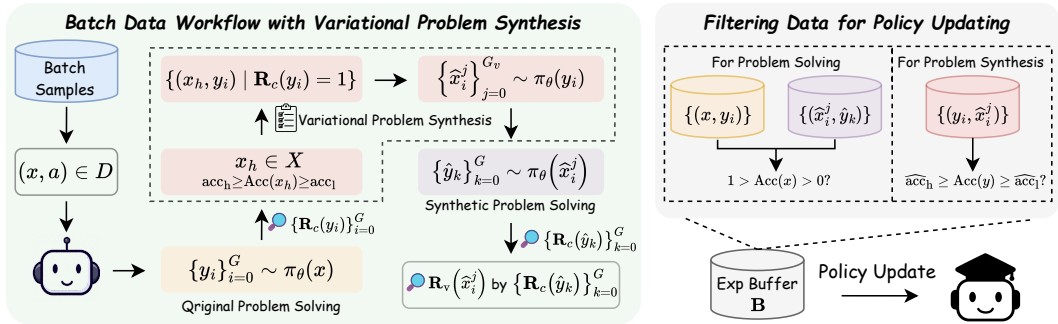

Figure 3: The data workflow of our SvS in a training iteration, comprising original problem solving, variational problem synthesis, synthetic problem solving, and policy update data filtering.

where $\mathbb{I}(\cdot)$ is the indicator function, and $\text{Extract}(\cdot)$ extracts the final answer from the reasoning trajectories. Since the advantage for groups with all-correct or all-incorrect solutions degrades to zero in GRPO, we filter out problems with group accuracy equal to 1 or 0. The remaining problems with solution groups $\{(x, y_i)\}_{i=1}^G$ and their corresponding rewards are added to the training buffer $\mathbf{B}$.

**Variational Problem Synthesis from Responses**. After generating solutions to the original problems, SvS identifies underperforming problems with low solve rates and synthesizes their variants to online augment the training set. Specifically, underperforming problems are defined as those with group average accuracy $\text{Acc}(x)$ falling within the range $[\text{acc}_l, \text{acc}_h]$ (Line 11 in Algorithm1), thereby excluding problems that are either too easy or unsolvable. This filtering strategy focuses the augmentation effort on problems that match the current model's frontier capabilities.

After identifying underperforming problems, SvS leverages the policy's correct solutions to synthesize corresponding variational problems for augmentation. Since a correct response $y_i$ contains the full informational content of the original problem $x$, each solution $y_i$ serves as context to generate a group of $G_v$ variational problems, $\{\hat{x}_i^j\}_{j=1}^{G_v}$, enriching the originals with more diverse structures and descriptions. The detailed prompt is present in Figure 18. Because the variational problems are derived from correct responses to the original problems, they are expected to share the same reference answers. This constraint not only serves as a criterion for validating the correctness of the variational problems, but also bypasses the need for additional answer annotations, which is crucial for RLVR data augmentation, where the reference answers provide the only training signal. Except for problem-solving augmentation, the correctness of generated variational problems is also incorporated into RLVR training, encouraging the policy to learn the inverse mapping from a solution to its problem statement and fostering a deeper understanding of the problems' semantics and structure. In Appendix F.2, we provide further analysis of how problem synthesis training helps problem solving.

**Synthetic Problem Solving**. Once a set of variational problems $\{\hat{x}_i^j\}_{j=1}^{G_v}$ is generated from $y_i$, the policy is tasked with solving them in the same way as solving the original training problems. For each variation problem $\hat{x}_i^j$, the policy produces a group of $G$ solutions $\{\hat{y}_k\}_{k=1}^G$, and the original ground-truth answer $a$ paired with $x$ is reused to evaluate their correctness. The corresponding correctness reward $\mathbf{R}_c$ is computed as:

$$\mathbf{R}_c(\hat{y}_k, a) = \mathbb{I}(\text{Extract}(\hat{y}_k) = a) \qquad (2)$$

Similar to the original problem solving filtering for experience buffering, we retain only variational problems for which the policy produces a mix of correct and incorrect solutions, i.e., $0 < \sum_{k=1}^G \mathbf{R}_c(\hat{y}_k, a) < G$, as they provide effective training signals in Eq. 6 of GRPO.

**Reward Shaping for Problem Synthesis**. Ideally, the correctness reward for variational problem synthesis, $\mathbf{R}_v$, is determined by whether the reference answer matches the original answer. Since precise reference answers for synthetic problems are unavailable and they must align with the policy's capabilities, as an intuitive implementation, we adopt a proxy criterion for validating them: a synthetic problem $\hat{x}_i^j$ is considered correct if the policy can produce solutions whose extracted answers match

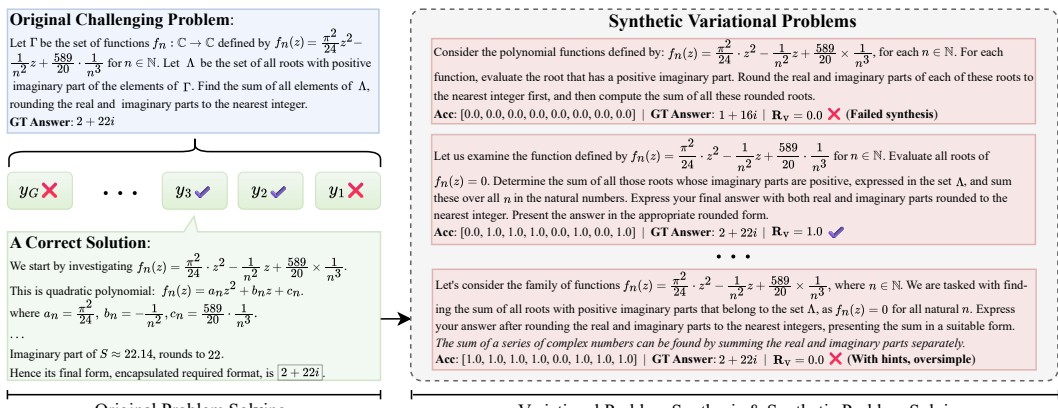

Figure 4: Illustrations of a challenging problem, its correct solution from policy, the synthetic variational problems from the solution, and the reward-shaping strategy for the synthetic problems.

the original answer $a$, formulated as

$$\mathbf{R}_{\mathrm{v}}(\hat{x}_i^j) = \mathbb{I}\Big(\mathrm{Acc}(\hat{x}_i^j, a) > 0\Big) \tag{3}$$

While straightforward, we find this reward can be easily exploited by the policy, which may embed excessive hints or even directly include the correct answer in the synthetic problems. Since they are generated given the correct responses, they can become trivial to solve, allowing the policy to obtain the reward in Eq. 3. Consequently, such variational problems are over-simplified and fail to encourage advanced reasoning of the policy, making the pipeline unsustainable and convergence suboptimal.

To ensure that variational problems remain diverse and effectively elicit stronger reasoning of the policy, we introduce a reward-shaping constraint to validate them, requiring that they maintain an appropriate level of difficulty for the policy. Specifically, we assign positive rewards to a synthetic problem only if the policy achieves a moderate level of group accuracy—neither too high nor entirely incorrect—rather than simply rewarding it for which a correct answer is sampled. The reward for each variational problem $\hat{x}$ is defined as:

$$\mathbf{R}_{\mathrm{v}}(\hat{x}_i^j) = \mathbb{I}\Big(\mathrm{a\hat{c}c}_{\mathrm{l}} \leqslant \mathrm{Acc}(\hat{x}_i^j, a) \leqslant \mathrm{a\hat{c}c}_{\mathrm{h}}\Big) \tag{4}$$

Notably, as shown in Figure 4, if a synthetic variational problem can be fully addressed or no solution aligning with $a$ can be sampled, it receives a negative reward. This discourages the policy from generating overly hint-laden, unverifiable, or unsolvable problems, ensuring that synthetic problems remain challenging while providing effective learning signals.

**Full Training Data**. After experience collection, for each training step, the final training buffer $\mathbf{B}$ contains three distinct types of prompt-response-reward tuples: **(1)** Original Problem Solving: $(x, y_i, \mathbf{R}_{\mathrm{c}}(y_i, a))$; **(2)** Variational Problem Synthesis: $(y_i, \hat{x}_i^j, \mathbf{R}_{\mathrm{v}}(\hat{x}_i^j))$ **(3)** Synthetic Problem Solving: $(\hat{x}_i^j, \hat{y}_k, \mathbf{R}_{\mathrm{c}}(\hat{y}_k, a))$. Utilizing the augmented buffer $\mathbf{B}$, the SvS framework updates the policy $\pi_\theta$ according to the GRPO gradient update objective in Eq. 6. By jointly training on the problem solving and synthesis tasks, the policy learns to solve the given training problems, generate challenging problems for itself, and solve the self-generated problems, forming a powerful self-improving loop.

# 4 EXPERIMENTS

## 4.1 SETTINGS

**Models and Datasets**. We employ models of various sizes (3B to 32B) for validating the effectiveness of our proposed SvS, including Qwen2.5-3B-Instruct, LLaMA-3.1-8B-Instruct (Grattafiori et al., 2024), and Qwen2.5-32B-Instruct (Yang et al., 2024). All models are trained on the MATH-12k dataset (Hendrycks et al., 2021), with the 32B model additionally trained on the DAPO-17k dataset to enhance competition-level reasoning capabilities.

| Model | Pass@1 | | | | | | | Pass@32 | | | | | | |
|---|---|---|---|---|---|---|---|---|---|---|---|---|---|---|
| | AIME24 | AIME25 | BAIME | Math24o | OlymE | OlymH | Avg. | AIME24 | AIME25 | BAIME | Math24o | OlymE | OlymH | Avg. |
| *Open-Source Models* | | | | | | | | | | | | | | |
| Qwen2.5-32B | 4.3 | 1.2 | 2.4 | 8.0 | 3.7 | 1.6 | 3.5 | 38.9 | 15.6 | 18.7 | 34.0 | 24.6 | 15.2 | 24.5 |
| Qwen2.5-32B-IT | 10.0 | 13.0 | 7.4 | 26.0 | 8.6 | 2.0 | 11.2 | 40.2 | 34.6 | 24.0 | 67.8 | 35.2 | 9.5 | 35.2 |
| SimpleRL-32B | 22.1 | 13.9 | 8.3 | 25.5 | 9.4 | 3.7 | 13.8 | 62.0 | 38.5 | 27.4 | 69.9 | 42.5 | 19.4 | 43.3 |
| ORZ-32B | 24.2 | 26.3 | 10.9 | 16.1 | 12.2 | 1.1 | 15.1 | 55.7 | 47.0 | 29.4 | 58.0 | 45.9 | 12.3 | 41.4 |
| *MATH-12k* | | | | | | | | | | | | | | |
| → RLVR | 22.2 | 15.8 | 11.5 | 34.5 | 11.7 | **4.1** | 16.6 | 47.4 | 36.4 | 29.2 | 66.0 | 36.2 | 16.4 | 38.6 |
| → SvS | **30.3** | **21.7** | **13.8** | **42.7** | **20.1** | 3.3 | **22.0** | **63.6** | **55.1** | **41.5** | **79.2** | **63.6** | **24.8** | **54.6** |
| Δ | +8.1 | +5.9 | +2.3 | +8.2 | +8.4 | -0.8 | +5.4 | +16.2 | +18.7 | +12.3 | +13.2 | +27.4 | +8.4 | +16.0 |
| *DAPO-17k* | | | | | | | | | | | | | | |
| → RLVR | 28.8 | 30.0 | 14.0 | 39.6 | 17.9 | **4.8** | 22.5 | 52.5 | 42.4 | 35.9 | 71.2 | **47.1** | **18.3** | 44.6 |
| → SvS | **39.3** | **40.5** | **19.2** | **44.1** | **21.8** | 2.7 | **27.9** | **70.8** | **65.2** | **45.9** | **76.5** | 43.4 | 16.7 | **53.1** |
| Δ | +10.5 | +10.5 | +5.2 | +4.5 | +3.9 | -2.1 | +5.4 | **+18.3** | **+22.8** | +10.0 | +5.3 | -3.7 | -1.6 | +8.5 |

Table 1: Comparison of model performance on challenging benchmarks using the *Pass@1* (average 32 times) and *Pass@32* metrics. The Δ row shows the improvement of SvS over standard RLVR. The BAIME, Math24o, OlymE, and OlymH benchmarks correspond to BeyondAIME, Math24o, and the en-easy and en-hard subsets of OlymMATH, respectively.

**Evaluation**. We evaluated the models on a wide range of mathematical reasoning benchmarks, including GSM8K (Cobbe et al., 2021), MATH-500 (Lightman et al., 2023), Minerva Math (Lewkowycz et al., 2022), Olympiad-Bench (He et al., 2024), Gaokao-2023 (Zhang et al., 2023), AMC (MAA, a), AIME (MAA, b) and Beyond-AIME (ByteDance-Seed, 2025). To more comprehensively evaluate the models' advanced reasoning capabilities, we also evaluated their *Pass@k* and *Pass@1* (average 32 times) performance on additional challenging benchmarks, including OlymMATH (Sun et al., 2025) and Math-24o (CLUEbenchmark, 2024). Details of the training, evaluation implementation, and baseline settings are provided in Appendix E.

## 4.2 MAIN RESULTS

**SvS significantly improves both *Pass@1* and *Pass@k***. As shown in Figure 1, naive RLVR training plateaus at *Pass@32* and *Pass@1* on competition-level AIME benchmarks after roughly 450 steps. In contrast, the model trained with the SvS strategy achieves substantial and sustained improvements in both metrics on these challenging benchmarks. Table 1 shows that models trained on the DAPO dataset with the SvS strategy achieve absolute gains of **18.3** and **22.8** points on *Pass@32* for AIME 24 and AIME 25, respectively, compared to the standard RLVR baseline. These results not only demonstrate the effectiveness of SvS, but also highlight the potential of self-play–style RLVR training to enhance *Pass@k* and expand the model's reasoning capabilities. The rising *Pass@k* during training also facilitates greater exploration, which in turn improves *Pass@1*.

**SvS boosts RLVR across all settings**. Table 2 presents experimental results for models ranging from 3B to 32B across all evaluated benchmarks using the *Pass@1* metric. To mitigate high randomness, we evaluate models smaller than 8B on AIME-level benchmarks using an average of 32 inferences. Notably, the SvS strategy consistently outperforms standard RLVR across all model sizes, yielding overall improvements of 2.9%, 1.7%, and 2.5% for the 3B, 8B, and 32B models when trained on the MATH-12k dataset. Notably, for Qwen2.5-3B-Instruct, RLVR training on MATH-12k does not improve performance on the MATH-500 benchmark, whereas SvS yields a 3.0-point gain, demonstrating its generalizability. Experiments for the Qwen2.5-32B-Instruct model are conducted using both the MATH-12k and DAPO-17k training sets. When trained on MATH-12k, our model demonstrates improved performance across all benchmarks, with an overall gain of 2.5 absolute points. On the DAPO-17k experiments, SvS significantly enhances performance on AIME 24, AIME 25, and Beyond-AIME, with improvements of 20.0, 6.7, and 6.0 points, respectively. Nevertheless, it results in reduced performance on benchmarks with open-ended answers, likely because the model overfits to DAPO-17k's integer-only format during augmentation. By training the model using SvS on DAPO-17k with 8k open-ended problems from DeepMath (He et al., 2025b), the model restores its performance on related benchmarks and achieves the best overall results.

| Model | Training Data | GSM8K | MATH 500 | Minerva Math | Olympiad Bench | GaoKao 2023 | AMC23 | AIME24 | AIME25 | Beyond AIME | Avg. |
|---|---|---|---|---|---|---|---|---|---|---|---|
| *Qwen2.5-3B-Instruct* | | | | | | | | | | | |
| Init Model | - | 87.3 | 67.8 | 29.4 | 30.7 | 59.0 | 37.5 | 4.8 | 1.7 | 1.7 | 35.5 |
| ↳ RLVR | M12k | 86.4 | 67.4 | 29.4 | 30.2 | 57.7 | 57.5 | 6.7 | 3.4 | 2.7 | 37.9 |
| ↳ SvS | M12k | 88.9 | 70.8 | 31.2 | 38.4 | 61.6 | 55.0 | 10.5 | 7.8 | 2.8 | 40.8 |
| *LLaMA-3.1-8B-Instruct* | | | | | | | | | | | |
| Init Model | - | 85.6 | 48.2 | 24.6 | 18.8 | 39.7 | 22.5 | 2.5 | 0.3 | 0.5 | 27.0 |
| ↳ RLVR | M12k | 90.2 | 57.4 | 33.8 | 22.4 | 47.8 | 45.0 | 8.1 | 1.2 | 1.5 | 34.2 |
| ↳ SvS | M12k | 90.3 | 62.2 | 32.4 | 26.4 | 54.8 | 45.0 | 8.5 | 1.8 | 2.0 | 35.9 |
| *Qwen2.5-32B-Instruct* | | | | | | | | | | | |
| Init Model | - | 95.4 | 82.6 | 43.0 | 49.2 | 73.2 | 65.0 | 13.3 | 13.3 | 7.0 | 49.0 |
| ↳ RLVR | M12k | 95.8 | 86.4 | 45.6 | 52.7 | 74.5 | 77.5 | 26.7 | 23.3 | 11.0 | 54.8 |
| ↳ SvS | M12k | **96.1** | 87.2 | 46.0 | 56.7 | 78.7 | 80.0 | 30.0 | 26.7 | 14.0 | 57.3 |
| ↳ RLVR | D17k | 95.6 | 87.0 | 45.6 | 54.8 | 78.7 | 82.5 | 33.3 | 36.7 | 13.0 | 58.6 |
| ↳ SvS | D17k | 95.9 | 75.6 | 42.3 | 45.9 | 62.9 | 82.5 | **53.3** | **43.3** | 19.0 | 57.9 |
| ↳ SvS | D25k | 95.2 | **88.6** | **47.8** | **59.9** | **79.2** | **87.5** | 50.0 | 40.0 | 17.0 | **62.8** |

Table 2: Performance comparison between the standard RLVR and our SvS strategy on mainstream reasoning benchmarks, using different training sets and models of varying scales and families. The datasets M12k, D17k, and D25k correspond to MATH-12k, DAPO-17k, and DAPO-17k augmented with 8k problems with open-ended answers from DeepMath, respectively.

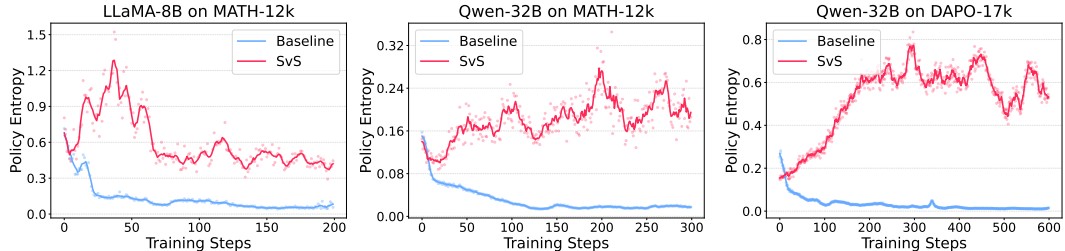

Figure 5: Policy entropy trajectories during training for standard RLVR and the SvS strategy across various models and datasets. Lines are smoothed with a window of 5 steps.

# 5 ANALYSIS

## 5.1 SvS STABLY MAINTAINS POLICY ENTROPY IN TRAINING

In RLVR training, policy entropy reflects the model's capacity for sustained exploration (Cui et al., 2025b; Cheng et al., 2025). Standard RLVR algorithms typically result in a steady decline in entropy, enhancing policy sampling efficiency and *Pass@1* performance but reducing generation diversity (Cui et al., 2025b). To evaluate whether the SvS strategy faces the same limitation, we record the entropy trajectories of both SvS and RLVR (GRPO with Clip-Higher) throughout the training in Figure 5. Notably, the RLVR baseline shows a continuous decline in entropy, whereas SvS maintains entropy within a relatively stable range, supporting sustained exploration and avoiding training collapse. Such advantages stem from the ever-updating problems in SvS that prevent the policy from memorization. The entropy stability explains the continuous improvements in both *Pass@1* and *Pass@32* achieved by SvS, as shown in Figure 1, whereas RLVR saturates after a certain number of training steps.

## 5.2 SvS PUSHES THE REASONING BOUNDARY

Recent study (Yue et al., 2025) discusses that standard RLVR often fails to expand the reasoning boundary of the base model, yielding improvements in *Pass@k* only for small values of $k$. Since our SvS training achieves a substantial improvement in *Pass@32*, we further evaluate its effectiveness and limits in incentivizing reasoning by scaling *Pass@k* from 1 to 1024, testing whether the SvS-trained model can solve problems beyond the capability of the base model. As presented in Figure 6, our experiments demonstrate that both standard RLVR and SvS improve *Pass@k* scores on the competition-level AIME benchmarks across all $k$, with SvS significantly outperforming the RLVR baseline. For *Pass@k* scaling on MATH-500, standard RLVR outperforms the initial model at small

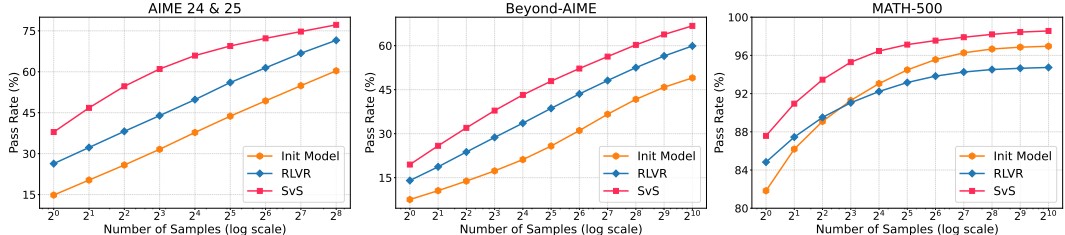

Figure 6: Evaluating the scaled-up *Pass@k* performance on the AIME 24 & 25, Beyond-AIME, and MATH-500 benchmarks. The maximum response tokens here is set to 24k.

| Model | Pass@1 | | | | | | | Pass@32 | | | | | | |
|---|---|---|---|---|---|---|---|---|---|---|---|---|---|---|
| | AIME24 | AIME25 | BAIME | Math24o | OlymE | OlymH | Avg. | AIME24 | AIME25 | BAIME | Math24o | OlymE | OlymH | Avg. |
| **RLVR** | 28.8 | 30.0 | 14.0 | 39.6 | 17.9 | **4.8** | 22.5 | 52.5 | 42.4 | 35.9 | 71.2 | 47.1 | 18.3 | 44.6 |
| **Ext** | 31.7 | 31.3 | 15.8 | 43.2 | 21.5 | 4.1 | 24.6 | 57.0 | 48.0 | 33.3 | 72.6 | **51.8** | 15.0 | 46.3 |
| **Eup** | 28.7 | 26.8 | 13.9 | 40.7 | 16.0 | 4.2 | 21.7 | 64.2 | 54.6 | 40.7 | 76.2 | 50.2 | **19.6** | 50.9 |
| **SvS-Asp** | 31.6 | 27.3 | 13.7 | 43.4 | 16.5 | 3.9 | 22.8 | 50.7 | 48.0 | 30.4 | 66.4 | 44.3 | 17.0 | 42.8 |
| **Full SvS** | **39.3** | **40.5** | **19.2** | **44.1** | **21.8** | 2.7 | **27.9** | **70.8** | **65.2** | **45.9** | **76.5** | 43.4 | 16.7 | **53.1** |

Table 3: Comparison between SvS and alternative augmentation strategies and ablation study, including extending RLVR training (Ext), enhancing underperforming problems in RLVR (Eup), and augmenting simpler problems (SvS-Asp). For benchmark abbreviations, see Table 1.

$k$ values but is surpassed at larger $k$. In contrast, SvS consistently outperforms both RLVR and the initial model as $k$ increases, demonstrating its strong generalization and robust reasoning diversity. We attribute this enhanced diversity to the diversity maintenance of SvS, which supports exploration of more advanced reasoning strategies for solving complex problems throughout training.

## 5.3 ALTERNATIVE AUGMENTATION STRATEGIES AND ABLATION STUDIES

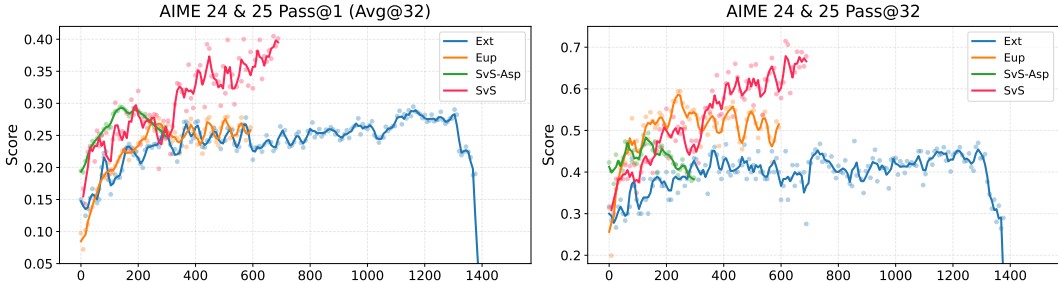

Figure 7: The intermediate evaluations on the AIME 24 and 25 benchmarks of all alternative augmentation strategies in Section 5.3, as well as the Full SvS setting.

In SvS training, we incorporated additional self-synthesized variational problems into policy optimization. In this section, we investigate whether the improvements in SvS arise merely from scaling up training samples. To this end, we compare SvS with alternative augmentation strategies and perform an ablation on the augmentation of underperforming problems. Specifically, they are:

1. **Extending standard RLVR training**: we prolong the standard RLVR training until the policy is exposed to the same number of problem–solution pairs as in SvS.

2. **Enhancing underperforming problems in RLVR**: a second rollout stage is assigned to underperforming problems, ensuring that the training pairs at each step align with the SvS trajectory.

3. **Augmentating simpler problems in SvS**: we augment simpler problems (accuracy in 37.5%–75.0%) as an alternative to underperforming ones used in the original SvS.

All experiments in Table 3 are conducted using Qwen2.5-32B-Instruct on the DAPO-17k training set, with checkpoints selected based on the best average scores from AIME 24 and 25. Notably, neither strategy surpasses the full SvS. *Extending standard RLVR training* yields overall performance improvements, aligning with the results of (Liu et al., 2025a). *For Enhancing underperforming problems* with additional rollouts, it achieves much higher *Pass@32* but lower *Pass@1* compared to

standard RLVR, suggesting that it prioritizes reasoning exploration over exploiting generated correct responses. This corresponds to the conclusion in (Zhu et al., 2025), as this strategy introduces more negative samples from underperforming problem augmentation, and such exploration effectively improves the model's *Pass@k* scores. *For Augmenting simpler problems in SvS*, it achieves similar *Pass@1* as standard RLVR but yields a lower overall *Pass@32*, indicating that this augmentation accelerates overfitting to the policy's already mastered capabilities while limiting exploration.

From these observations, two conclusions emerge: (1) Response-based augmentation in RLVR should focus primarily on underperforming problems (*Eup v.s. SvS-Asp*); and (2) Maintaining diversity in problem augmentation, rather than fixing the training set, is also crucial (*Eup v.s. Full SvS*). We also provide additional multidimensional analyses of SVS, presented in Appendix F.

## 5.4 SVS GENERALIZES BEYOND MATHEMATICS: RESULTS ON CODE GENERATION

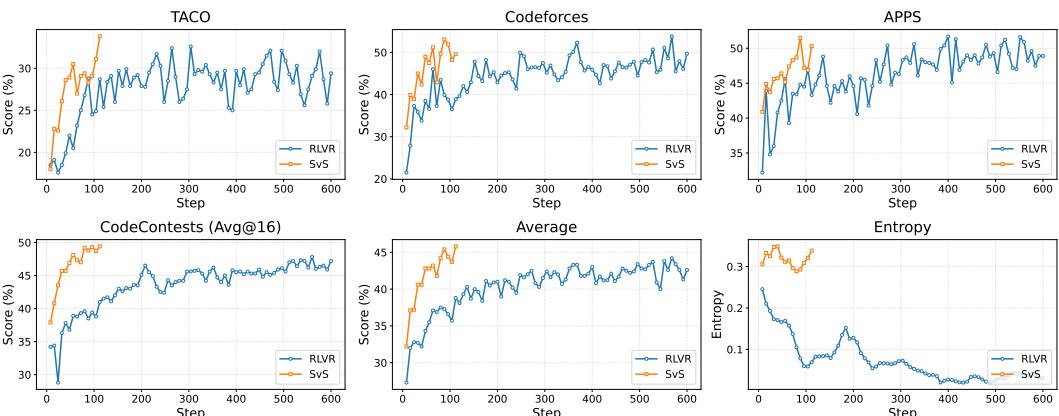

Figure 8: The intermediate evaluation comparing SvS and RLVR baseline on code generation tasks.

We incorporate the SvS strategy into RLVR training for code generation tasks to demonstrate its generalizability beyond mathematical reasoning. Specifically, we use the Qwen2.5-7B-Instruct model to perform RL on 12k code generation problems from PRIME-RL (Cui et al., 2025a), covering sources such as Apps, CodeContest, Taco, and Codeforces. For evaluation, we sample 100 instances from each validation set. The hyperparameters remain unchanged from our other experiments. The intermediate evaluation is presented in Figure 8. Notably, SvS training achieves significant improvements over the RLVR baseline and maintains stable policy entropy, demonstrating the strong generalization of this online self-play augmentation strategy in RLVR.

## 6 CONCLUSION

In this work, we propose an online **S**elf-play with **V**ariational problem **S**ynthesis (**SvS**) strategy for RLVR training, in which the policy model independently synthesizes variational problems to improve its performance on underperforming training samples, enabling a sustainable self-improvement process. By generating structurally diverse yet semantically aligned problems without requiring additional ground-truth annotations, our method ensures both diversity and verifiability of the training data throughout RLVR iterations, effectively maintaining policy entropy and generation diversity for sustained exploration. Extensive experiments show that SVS consistently outperforms standard RLVR across various model scales and benchmarks, particularly improving *Pass@k* scores at larger $k$ on competition-level benchmarks, where standard RLVR exhibits limited gains.

## ETHICS STATEMENT

The authors have read and agree to comply with the ICLR Code of Ethics. In this work, we propose a self-play-based problem synthesis strategy for reinforcement learning in LLM reasoning. This study does not involve human subjects, nor collect any new datasets containing personally identifiable or sensitive information, apart from mathematical reasoning–focused problem synthesis and problem-solving generations derived from open-source LLMs. Both the training and evaluation datasets are publicly available, widely used in the research community, and do not raise immediate privacy or data bias concerns. While we acknowledge that we cannot fully guarantee all responses from the trained models are helpful and harmless, this research project focuses solely on improving the robustness of LLM reasoning. We do not foresee any direct negative societal impacts stemming from this research.

## REPRODUCIBILITY STATEMENT

We provide an anonymous code package in the supplementary materials, which includes source files, experiment configurations, and full algorithm implementations for reproducing all figures and tables. Practical training and evaluation details, along with the experimental settings, are provided in Section E. The complete algorithmic workflow is described in Appendix 1, while the overall framework is illustrated in Figure 3. The prompts used in this work are provided in Section J. The training models, datasets, and evaluation benchmarks are provided in Appendix 4, while descriptions of all baseline methods are included in Appendix E.

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

APPENDIX CONTENTS FOR SvS

## A    LLM USAGE DISCLOSURE

In preparing this paper, we used ChatGPT, an OpenAI Large Language Model (LLM), exclusively to polish the manuscript's language for enhancing clarity and readability. The LLM was not involved in research ideation, experimental design, data analysis, or drawing conclusions. All substantive content, technical contributions, and interpretations are the authors' own. The authors take full responsibility for the accuracy and integrity of the manuscript.

## B    PRELIMINARY FOR GRPO

Group Relative Policy Optimization (GRPO; Shao et al. 2024) is an efficient optimization algorithm tailored for reinforcement learning in LLMs, where the advantages for each token in a rollout are computed in a group-relative manner without requiring an additional critic model to estimate token values. Specifically, given an input prompt $x$, the policy model $\pi_{\theta_{old}}$ generates a group of $G$ responses $\mathbf{Y} = \{y_i\}_{i=1}^G$, with acquired rewards $\mathbf{R} = \{r_i\}_{i=1}^G$. The advantage $A_{i,t}$ for each token in response $y_i$ is computed as the group-level normalized rewards:

$$A_{i,t} = \frac{r_i - \text{mean}(\{r_i\}_{i=1}^G)}{\text{std}(\{r_i\}_{i=1}^G)}. \tag{5}$$

To improve the stability of policy optimization, GRPO clips the probability ratio $k_{i,t}(\theta) = \frac{\pi_\theta(y_{i,t}|x,y_{i,<t})}{\pi_{\theta_{old}}(y_{i,t}|x,y_{i,<t})}$ within a trust region (Schulman et al., 2017), and constrains the policy distribution from deviating too much from the reference model using a KL term. The final optimization objective is defined as follows:

$$\mathcal{J}_{\text{GRPO}}(\theta) = \mathbb{E}_{x \sim \mathcal{D}, \mathbf{Y} \sim \pi_{\theta_{old}}(\cdot|x)}$$
$$\left[ \frac{1}{G} \sum_{i=1}^G \frac{1}{|y_i|} \sum_{t=1}^{|y_i|} \left( \min\left( k_{i,t}(\theta)A_{i,t}, \ \text{clip}\left( k_{i,t}(\theta), 1-\varepsilon, 1+\varepsilon \right) A_{i,t} \right) - \beta D_{\text{KL}}(\pi_\theta || \pi_{\text{ref}}) \right) \right]. \tag{6}$$

## C    EXTENDED RELATED WORK

### C.1    REINFORCEMENT LEARNING WITH VERIFIABLE REWARDS

Reinforcement Learning with Verifiable Rewards (RLVR) has significantly improved LLMs in complex reasoning tasks (Luong et al., 2024; Guo et al., 2025a). Algorithms such as PPO (Schulman et al., 2017) and GRPO (Shao et al., 2024) have shown strong generalization and effectiveness in LLM post-training. Existing efforts in scaling up RLVR optimization have focused on enhancing exploration (Yu et al., 2025a; Yuan et al., 2025; Liu et al., 2025b; Yeo et al., 2025) and adapting RLVR to the Long-CoT conditions (Jaech et al., 2024; Guo et al., 2025a; Li et al., 2025d; Yang et al., 2025). Yu et al. (2025a) found that removing the KL constraint and incorporating the Clip-Higher strategy on top of GRPO facilitates better exploration during training.

However, Yue et al. (2025) raised an insightful question of whether RLVR truly incentivizes capability expansion beyond the base LLM, with experiments showing that it does not enhance *Pass@k*—a metric associated with the reasoning boundaries of LLMs. Some studies (Gao et al., 2025; Cui et al., 2025b; Zhu et al., 2025) have also found that the entropy of model outputs declines during RLVR training, especially in the early stages, which hinders sustained exploration in later training. To mitigate entropy decline, Cheng et al. (2025) proposes augmenting the token advantage with an entropy-based term, while An et al. (2025) and Chen et al. (2025) find that tuning the temperature appropriately helps maintain rollout diversity during training. Concurrent with our work, CURE Li et al. (2025b) introduces an effective mechanism for promoting exploration through high-entropy token expansion, which successfully helps prevent entropy collapse in RLVR. In this paper, we analyze policy entropy from the perspective of training data diversity and introduce a self-play-style problem augmentation strategy (SvS) for RLVR training, which effectively maintains training entropy within a stable range and significantly boosts model *Pass@k* performance, as $k$ scaled up to 1024.

## C.2 Data Construction for LLM Reasoning

The construction of training data is crucial for enhancing the model's reasoning capabilities (Luo et al., 2025; Yu et al., 2025a; Hu et al., 2025b; Zhang et al., 2025; He et al., 2025a; Shen et al., 2025; Li et al., 2025c; Liang et al., 2025). However, high-quality human-labeled mathematical problems are limited and overly simplistic for advanced modern LLMs (Cobbe et al., 2021; Hendrycks et al., 2021). To augment training data for LLM reasoning, existing data synthesis approaches have explored generating problem-response pairs (Huang et al., 2024; Tang et al., 2024; Yu et al., 2023; Zhao et al., 2025; Liang et al., 2024; Wang et al., 2024a; Li et al., 2024; Tan et al., 2024) or augmenting responses to existing questions (Toshniwal et al., 2024; He et al., 2025a; Face, 2025; Yu et al., 2025b; Li et al., 2025c). Li et al. (2025a); Jiang et al. (2025) propose incorporating latent representation data to model LLM reasoning. Targeting the training paradigm of RLVR, Guo et al. (2025b) proposes to synthesize question and answer pairs from the task definition and documents, while SwS (Liang et al., 2025) generates synthetic problems based on the model's failure cases during RLVR training. Most related to our work, Cheng et al. (2024) introduces utilizing self-play-style instruction data to enhance model reasoning through adversarial training. In contrast to existing approaches, SVS enables online data augmentation without requiring ground-truth answer annotations. Our strategy effectively maintains training entropy in a stable range throughout RLVR, supports end-to-end training, and performs augmentation using the policy itself without external dependencies, expanding the policy's reasoning boundaries through full self-improvement.

## D  Full Algorithm of SVS

We present the full algorithm of SVS training in Algorithm 1. The inputs to the SVS training framework include the Training set $\mathcal{D}$, the Initial policy $\pi_\theta$, the Underperforming accuracy range $[\text{acc}_l, \text{acc}_h]$ for selecting problems whose correct responses are used to generate variational problems, the Positive synthesis range $[\hat{\text{acc}}_l, \hat{\text{acc}}_h]$ for defining reward shaping in the response-to-problem synthesis task, the group sizes $G$ and $G_v$ for problem solving and problem synthesis, and the total number of training steps $T$.

Generally, Lines 4–32 describe a complete training step of SVS, with experience collection and policy updating. Lines 5-10 detail the use of the policy to generate solutions for the problem batch sampled from the training set, where only problems with accuracy in $(0, 1)$ are filtered into the buffer for policy updates. Lines 11–12 indicate that problems with accuracy within $[\text{acc}_l, \text{acc}_h]$ are chosen, and their correct responses serve as context for later variational problem synthesis. Lines 13–20 describe both the variational problem synthesis process and the utilization of the same policy to solve the synthesized problems. Rewards for the solutions to these synthetic problems are assigned based on whether their extracted answers match the reference answers of the original problems. Similarly, only variational problems with accuracy in $(0, 1)$ are retained for policy updates. Lines 21–26 describe the reward assignment for the variational problem synthesis training pairs. Specifically, only synthetic variational problems whose policy-sampled solutions achieve accuracies within $[\hat{\text{acc}}_l, \hat{\text{acc}}_h]$ are assigned positive rewards; otherwise, the synthesized problem receives a negative reward. This design compels the policy to generate problems of appropriate difficulty that support effective problem synthesis training. Finally, only correct responses (input of this task) with a mix of positive and negative synthesis rewards are retained in the buffer for policy updates, consistent with the problem-solving task. During policy updates, all three types of training pairs in the buffer are mixed for gradient optimization, then the buffer is cleared for the next training step.

## E  Implementation Details

### E.1  RLVR Training

We choose GRPO (Shao et al., 2024) as our RLVR optimization strategy and incorporate several techniques from (Yu et al., 2025a), including Clip-Higher with $\varepsilon$ set to $0.28$, Token-Level Loss, and Dynamic Sampling. We set the learning rate to $1e^{-6}$ with a constant schedule. The sampling temperature is fixed to 1.0. The batch sizes for sampled problems and policy updates in each iteration are both set to 256. The group size $G$ of solutions generated from each original and synthetic problem, as well as $G_v$ for variational problems derived from each response, is set to 8. The underperforming

---

**Algorithm 1** Self-play RLVR with Variational Problem Synthesis

---

1: **Input:** Training set $\mathcal{D}$, Initial policy $\pi_\theta$, Underperforming accuracy range $[\mathrm{acc_l}, \mathrm{acc_h}]$, Positive synthesis range $[\mathrm{a\hat{c}c_l}, \mathrm{a\hat{c}c_h}]$, Group size $G$ and $G_v$, Total training steps $T$.
2: **Initialize:** Training experience buffer $\mathbf{B} \leftarrow \varnothing$
3: **for** $t = 1, \ldots, T$ **do**
4:     Sample a data batch from the training set $\mathcal{D}$
5:     **for** input problem-answer pair $(x, a)$ in the batch **do**
6:         Generate a group of solutions $\{y_i\}_{i=1}^G$ to $x$ using $\pi_\theta$
7:         Compute correctness rewards $\{\mathbf{R}_c\}_{i=1}^G$ using $a$ for each solution $y_1, \ldots, y_G$
8:         **if** $0 < \mathrm{Acc}(x) < 1$ **then**
9:           $\mathbf{B} \leftarrow \mathbf{B} \cup \{(x, y_1), \ldots, (x, y_G)\}$
10:         **end if**
11:         **if** $\mathrm{acc_l} < \mathrm{Acc}(x) < \mathrm{acc_h}$ **then**
12:           Select $\{(x, y_i)\}_{i \in \mathcal{I}}$ such that $\mathcal{I} = \{i \mid \mathbf{R}_c(y_i, a) = 1\}$
13:           **for** accurate solution $y_i$ in $\{(x, y_i)\}_{i \in \mathcal{I}}$ **do**
14:               Synthesize a group of variational problems $\{\hat{x}_i^j\}_{j=1}^{G_v}$ from $y_i$ using $\pi_\theta$
15:               **for** variational problem $\hat{x}_i^j$ in $\{\hat{x}_i^j\}_{j=1}^G$ **do**
16:                   Generate a group of solutions $\{\hat{y}_k\}_{k=1}^G$ for $\hat{x}_i^j$ using $\pi_\theta$
17:                   Compute correctness rewards $\{\mathbf{R}_c\}_{i=1}^G$ using $a$ for each generation $\hat{y}_1, \ldots, \hat{y}_G$
18:               **end for**
19:               Select $\{\hat{x}_i^j\}_{j \in \mathcal{J}_1}$ such that $\mathcal{J}_1 = \{j \mid 0 < \mathrm{Acc}(\hat{x}_i^j) < G\}$
20:               $\mathbf{B} \leftarrow \mathbf{B} \cup \{(\hat{x}_i^j, \hat{y}_1), \ldots, (\hat{x}_i^j, \hat{y}_G) \mid j \in \mathcal{J}_1\}$
21:               Select $\{\hat{x}_i^j\}_{j \in \mathcal{J}_2}$ such that $\mathcal{J}_2 = \{j \mid \mathrm{a\hat{c}c_l} \leqslant \mathrm{Acc}(\hat{x}_i^j) \leqslant \mathrm{a\hat{c}c_h}\}$
22:               **if** $|\mathcal{J}_2| > 0$ **then**
23:                   **for** variational problem $\hat{x}_i^j$ in $\{\hat{x}_i^j\}_{j=1}^G$ **do**
24:                     Assign $\mathbf{R}_c(\hat{x}_i^j) = 1.0$ if $j \in \mathcal{J}_2$, and $\mathbf{R}_c(\hat{x}_i^j) = 0.0$ otherwise
25:                   **end for**
26:                   $\mathbf{B} \leftarrow \mathbf{B} \cup \{(y_i, \hat{x}_i^1), \ldots, (y_i, \hat{x}_i^G)\}$
27:               **end if**
28:           **end for**
29:         **end if**
30:     **end for**
31:     Update the policy $\pi_\theta$ according to Equation 6, using the experience buffer $\mathbf{B}$
32:     Remove collected samples from $\mathbf{B}$: $\mathbf{B} \leftarrow \varnothing$
33: **end for**

---

problem range $[\mathrm{acc_l}, \mathrm{acc_h}]$ is set to 12.5%–50.0%, while the positive reward range $[\mathrm{a\hat{c}c_l}, \mathrm{a\hat{c}c_h}]$ for variational problem synthesis is defined as 12.5%–62.5%. The prompt used to synthesize variational problems from correct responses to underperforming problems is shown in Figure 18. Models trained on the MATH-12k dataset run for 300 steps, while 32B models trained on the DAPO-17k dataset run for 600 steps for more comprehensive exploration.

### E.2 EVALUATION

During evaluation, we use vLLM (Kwon et al., 2023) with inference hyperparameters set to a temperature of 1.0, a top-p value of 0.7, and a max response length of 8,192, except in *Pass@k* scaling experiments, where the length is increased to 24,576. For *Pass@k* evaluation, we employ an unbiased estimation method (Chen et al., 2021) to reduce the high variance from single evaluations. We employ a hybrid rule-based verifier by integrating Math-Verify and the DAPO verifier in veRL (Sheng et al., 2024). We use the default chat template and enable CoT prompting by appending the instruction: "Let's think step by step and output the final answer within \boxed{}" after each question.

**Baselines**. We compare the SVS trained models primarily with the instruction-tuned initial policy and the standard RLVR models, trained using GRPO with the same techniques described in Section E.1. We also compare the SVS-trained models with models of the same size from SimpleRL (Zeng et al., 2025) and Open-Reasoner-Zero (Hu et al., 2025b).

## F    ADDTIONAL ANALYSIS ON SVS

### F.1    SVS ELICITS DEEPER REASONING

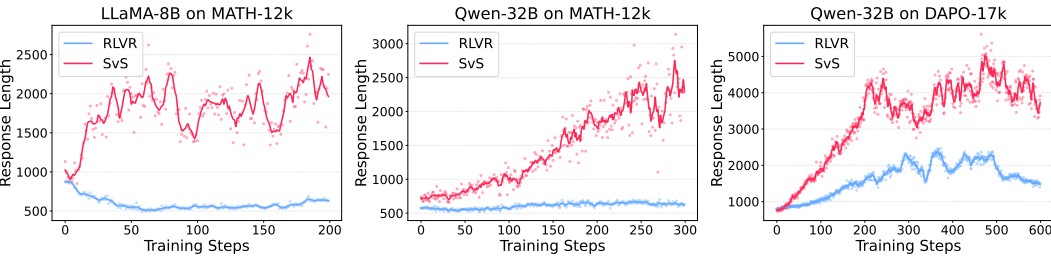

Figure 9: Training response lengths for standard RLVR and SVS across various models and datasets.

DeepSeek-R1 Guo et al. (2025a) has demonstrated that RLVR promotes deeper reasoning by encouraging longer solution paths, where extended reasoning trajectories often involve advanced strategies such as reflection, verification, and exploration. To assess the reasoning patterns induced by SVS, we compare its reasoning depth with that of standard RLVR training. We first compare reasoning lengths throughout training, as shown in Figure 9. We also present an exemplary case in Section K, illustrating a problem and the corresponding solutions from RLVR and SVS trained models. Notably, SVS consistently produces longer reasoning on training batches compared with RLVR. In experiments with LLaMA-8B and Qwen-32B on MATH-12k, standard RLVR consistently failed to generate extended reasoning paths from the initial policy, whereas SVS succeeded. This advantage arises because SVS requires the model to tackle new variational problems at each step, promoting continuous exploration of advanced reasoning strategies, whereas standard RLVR often secures high rewards by reusing memorized correct solutions.

### F.2    HOW PROBLEM SYNTHESIS ENHANCES PROBLEM SOLVING?

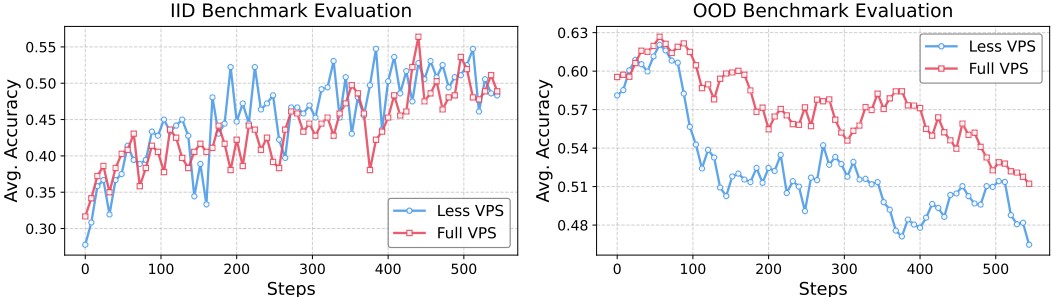

Figure 10: Evaluating *Pass@1* accuracy of intermediate checkpoints for the SVS training with full and reduced variational problem synthesis samples on IID and OOD benchmarks.

This section examines how problem-synthesis training improves overall problem-solving performance. To this end, we conduct an experiment where only 20% of the variational problem synthesis pairs are used for policy updating in each RL step, using Qwen2.5-32B-Instruct and DAPO-17k. Surprisingly, we find that incorporating problem-synthesis training significantly reduces policy's overfitting on the training set. For clarity, we categorize the benchmarks into IID (AIME24, AIME25, and AMC23) and OOD (MATH-500, Minerva Math, and Olympiad-Bench) groups based on whether their reference answers are integers, given that all answers in DAPO-17k are integers.

The intermediate evaluation results are presented in Figure 10. Notably, SvS with only 20% problem-synthesis training performs comparably to the full SvS on IID benchmarks but is significantly worse on OOD benchmarks, indicating that pure problem-solving training is susceptible to overfitting and reduces the model's generalizability, whereas the inclusion of problem synthesis helps mitigate this issue. The effectiveness of problem synthesis in alleviating overfitting can be attributed to its enrichment of the training distribution as well as its regularization of learning through the complementary tasks of problem generation and problem solving (He et al., 2016).

## F.3 SvS Generalizes beyond Reasoning Tasks

| Model | MMLU-Pro | ARC-C | ARC-E | HellaSwag | Winogrande | PIQA | BoolQ | HumanEval | AGIEval | Average |
|---|---|---|---|---|---|---|---|---|---|---|
| Init Model | 68.33 | 58.62 | **77.31** | 85.17 | **73.48** | 81.01 | **89.60** | **56.10** | 70.54 | 73.35 |
| ↳ RLVR | 70.25 | 57.94 | 76.60 | 85.28 | 72.53 | 80.74 | 89.36 | 53.66 | 70.57 | 72.99 |
| ↳ SvS | **71.58** | **58.79** | 76.98 | **85.34** | 73.40 | **81.34** | 89.48 | **56.10** | **70.89** | **73.77** |

Table 4: Evaluation results on general question-answering and code benchmarks. SvS achieves the highest overall performance across 9 tasks, outperforming both the initial model and standard RLVR.

Since the SvS training strategy incorporates the variational problem synthesis task, a general question-answering task beyond standard RLVR's problem-solving training, we evaluate whether this learning can transfer to improve performance on broader tasks, using the Qwen2.5-32B-Instruct model. Accordingly, we evaluate models trained on the DAPO-17k dataset using standard RLVR and the SvS strategy across general question-answering and coding benchmarks. The results are presented in Table 4. Notably, models trained with standard problem-solving RLVR exhibit a decline in performance on broad general benchmarks. In contrast, the SvS trained model not only avoids this degradation but also surpasses the initial instruction-following model on several general tasks, including MMLU-Pro (Wang et al., 2024b), ARC-Challenge (Clark et al., 2018), and HellaSwag (Zellers et al., 2019). These results indicate that the additional problem synthesis task in SvS helps prevent overfitting to mathematical reasoning tasks while effectively preserving or even enhancing the model's general instruction-following capabilities.

## F.4 SvS Outperforms RLVR on Challenging Problems

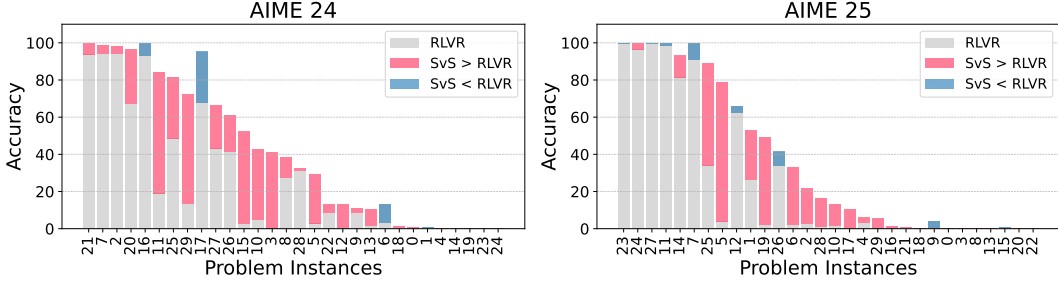

Figure 11: Comparison of instance-level accuracy between standard RLVR and SvS trained model. For each problem, the accuracy is averaged over 1024 generations on both AIME24 and AIME25.

In this section, we present an instance-level analysis of SvS and standard RLVR models on the challenging AIME 24 and 25 benchmarks. The comparison is shown in Figure 11. The gray bars represent the instance accuracy achieved by both RLVR and SvS, the red bars indicate SvS's advantage over RLVR, and the blue bars indicate the opposite. Notably, on a large number of problems—such as the 15th, 10th in AIME 24 and the 5th, 19th in AIME 25—RLVR achieves only limited accuracy, whereas SvS attains substantially better performance, reaching up to 80% on these problems. More significantly, SvS is able to solve problems that standard RLVR consistently fails to answer, such as the 12th in AIME 24 and the 17th in AIME 25, demonstrating its ability to extend the boundaries of model reasoning. In summary, SvS exhibits superior performance in both exploitation and exploration compared with standard RLVR, aligning with the results in Table 1.

## F.5 Computation Analysis of SvS Training Compared to RLVR

This section analyzes the computational overhead of SvS compared to the RLVR baseline. The comparison, using our experiments with 32B models trained on both the MATH-12k and DAPO-17k datasets, is illustrated in Figure 12. All experiments were conducted using 32 H100 GPUs.

Notably, when training on DAPO-17k, the initial stages exhibit a large number of synthetic variational problems with responses because the policy's accuracy on most problems falls within the range $[\mathrm{acc}_l, \mathrm{acc}_h]$. As training progresses, the model's performance improves, with accuracies gradually

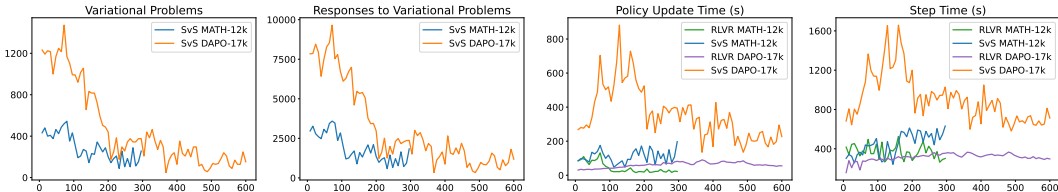

Figure 12: An illustration of the iteration step time and standalone policy update time for SvS and RLVR in the 32B model experiments. The two left panels show the number of variational problems and their corresponding responses in the SvS experiments.

surpassing $\mathrm{acc_h}$, leading to a rapid decline in synthetic generation. Conversely, when training on the simpler MATH-12k dataset, the Qwen-32B model already achieves an initial accuracy of approximately 80% on the training set. Consequently, the number of synthetic variational problems generated is limited, and the SvS training time is comparable to that of standard RLVR.

### F.6 ANALYSIS OF THE CORRECTNESS OF THE SYNTHETIC PROBLEMS

To understand whether the synthetic problems in SvS are genuily and logically correct, we employ two state-of-the-art LLMs: Qwen3-235B-A22B-Instruct-2507 and OpenAI O3, to evaluate 6,000 variational problems (10 problems per step across 600 steps), considering a problem correct if at least one of the models deems it valid. We use a prompt (Figure 20) that examines Completeness, Logical Consistency, Solvability, Mathematical Soundness, and Overall Validity, to assess the correctness of the synthetic problems. We report the average validity over a sliding window of 100 steps (1,000 problems) and present the results in Figure 13. We find that more than 80% of them are consistently judged as correct by the models, with a slight downward trend over training steps. This decline may be due to the synthetic problems gradually adopting a description style closer to the model's own way of expressing solutions in the later stages of training.

However, we also manually inspected the synthetic problems that both LLMs flagged as invalid and did not find any that were genuinely incorrect. Some problems were labeled as incorrect only because their textual descriptions appeared somewhat unusual, even though the underlying tasks were fully solvable. An example is shown below.

> Let $r = \frac{\texttt{"loooloolloolloololllloloolllollollolllloolllolooloooloooololoooolllllol"}}{\texttt{"lolooloololllollollooooolooolooloolollooollloolololoooollllooolollllolooool"}}$. Each "o" represents 2013 and "l" is $\frac{1}{50}$. Find $\lceil roll \rceil$ where each string is a 70-character string. Find and express the necessary final evaluation.

Other examples we evaluated follow a similar pattern: the problems are actually solvable, but both models incorrectly judge them as invalid. For instance:

> In an $5 \times 5$ lattice grid from $(0,0)$ up through $(5,5)$, there are teleports at coordinates $(2,2)$ and $(3,3)$. Whenever Michelle visits either point, she gets instantly teleported to the other if she hasn't visited both. Specifically, if she visits $(2,2)$ first, she's sent to $(3,3)$; if she visits $(3,3)$ first, she's moved to $(2,2)$. If she never visits those points, no teleportation happens. Find the number of valid ways for Michelle to reach $(5,5)$ from $(0,0)$. Solve this including all scenarios described. Present your detailed calculation and the final answer.

The Qwen3 Response is shown as follows. Its judgment is incorrect because the problem is fully well-posed under the standard lattice-path assumption that steps only move right or up. The teleportation rule is unambiguous: the first visit to either teleport triggers a single one-time jump, after which both points are considered visited and no further teleportation can occur. With this correct interpretation, the problem is entirely solvable and yields a unique answer of 84 valid paths.

## G COMPARING AND COMBINING SvS AND ENTROPY REGULATION METHODS

In SvS, we maintain the policy's generalization diversity by online generating synthetic problems within a self-play paradigm. Beyond this augmentation strategy, recent work, such as Cui et al. (2025b) and Li et al. (2025b), shows that regulating policy entropy can also enhance rollout diversity and mitigate entropy collapse. In this section, we comprehensively evaluate SvS against a representative

Validity: \boxed{INVALID}

Reasoning:
- Point 1: The problem is not fully well-posed due to ambiguity in the teleportation rule. The phrase "she gets instantly teleported to the other if she hasn't visited both" is ambiguous in a path-counting context. It is unclear whether "visited" refers to prior visits in the path or just the current step. Moreover, the rule does not specify what happens if Michelle is teleported to a point—does arriving at $(3, 3)$ via teleportation count as "visiting" it? This affects whether teleportation can trigger again or not, and whether cycles (e.g., going from $(2, 2) \to (3, 3)$, then later stepping to $(3, 3)$ again) are possible. These definitions are missing.
 - Point 2: There is a potential for logical inconsistency or infinite loops. For example, if Michelle starts at

$(0, 0)$, walks to $(2, 2)$, gets teleported to $(3, 3)$, then walks back to $(3, 3)$ again later, does this trigger another teleport? Or if she returns to $(2, 2)$ after being teleported, does it trigger again? Without clear rules on state tracking (e.g., a boolean flag for whether both teleports have been "visited"), the behavior is undefined and could lead to infinite paths (e.g., looping between teleports if rules are misinterpreted), making the problem logically inconsistent.
 - Point 3: The problem lacks sufficient information for solvability. The state space depends on whether

Michelle has "visited" $(2, 2)$ or $(3, 3)$ in her path, but it is not specified whether teleportation counts as a visit, nor whether the teleportation effect is disabled after the first use. Without modeling this state (e.g., using dynamic programming with memory of visitation), the number of valid paths cannot be uniquely determined. The problem does not specify if paths are self-avoiding or allow revisiting points, further increasing ambiguity.

Fix Suggestions:
- Clarify that teleportation only occurs the first time either $(2, 2)$ or $(3, 3)$ is physically stepped on, and that being teleported to the other point counts as visiting it, after which the teleportation effect is disabled.
- Specify that paths consist of moving only right or up (or define allowed moves), to make the lattice path model standard.
- Explicitly state that Michelle cannot revisit teleports in a way that reactivates teleportation, or define the state machine governing teleport use.
- Define whether arrival via teleportation counts as "visiting" the destination.

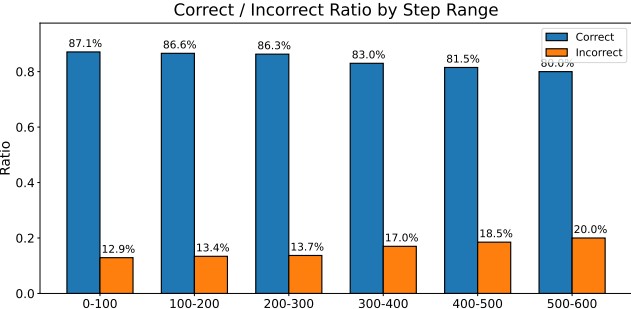

Figure 13: Illustration of the ratio and trend of synthetic problems deemed correct by SOTA LLMs.

entropy regulation strategy, Clip-CoV Cui et al. (2025b), and investigate the feasibility of combining these methods.

We compare SvS and Clip-Cov using the LLaMA-3.1-8B-Instruct model, trained on the MATH-12k dataset for more than 400 steps. Evaluation is conducted on GSM8k, MATH-500, Minerva-Math, Olympiad-Bench, Gaokao-2023 and AMC-23, and their average scores. The Clip-Cov parameters follow the default settings in the original paper, with a clip ratio $r = 2 \times 10^{-4}$, $\omega_{\text{low}} = 1$, and $\omega_{\text{high}} = 5$. The hyperparameters for SvS augmentation follow the settings described in Section E. As shown in Figure 14, SvS training consistently outperforms the Clip-Cov baseline in intermediate evaluations. This improvement stems from SvS's continuous online augmentation of training problems, which promotes consistent exploration, while entropy-collapse mitigation techniques like Clop-Cov fail to stop the policy from memorizing previously correct responses to secure rewards during training.

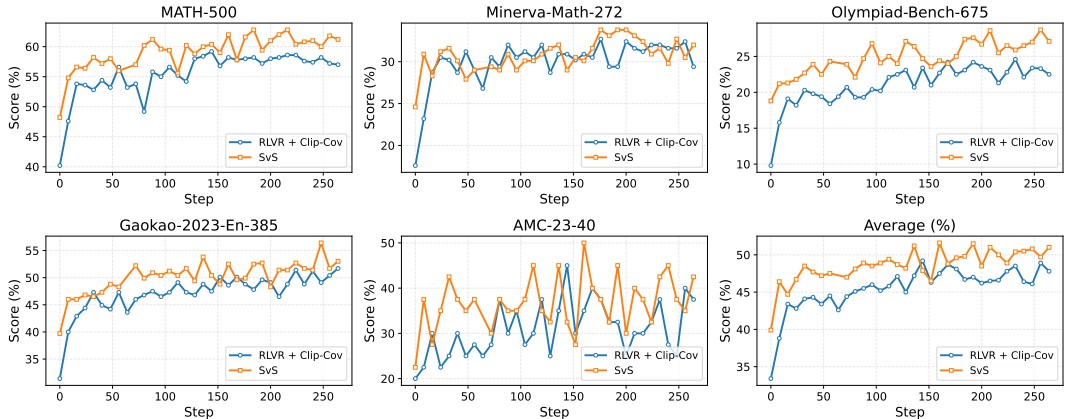

Figure 14: Comparison between SvS and the Clip-Cov strategy under mathematical RLVR training.

We also investigate the potential of combining entropy-regulation methods with SvS. Explicit entropy regularization may enable the policy to explore richer and more diverse solution strategies for each augmented problem. From a training perspective, SvS generates new augmented variants of challenging problems across epochs based on the model's current responses, preventing memorization of previous correct rollouts and promoting sustained exploration. Consequently, SvS can mitigate memorization within the entropy-regularized methods, allowing more effective exploration.

We use the same experimental settings as in the previous comparison, and the results are shown in Figure 15. The figure indicates that integrating SvS augmentation with Clip-Cov consistently improves policy performance over the Clip-Cov baseline, demonstrating that entropy-regulation methods can be further strengthened when combined with SvS.

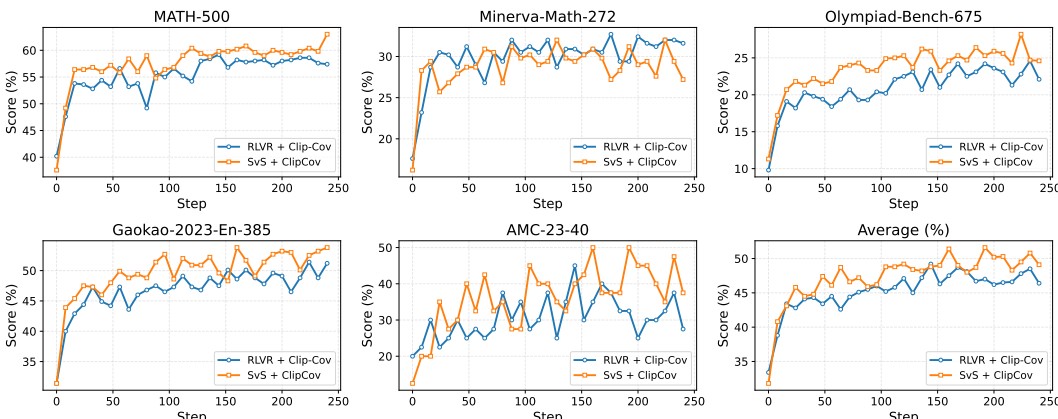

Figure 15: Illustration of how effectively SvS can be incorporated into the Clip-Cov strategy, and comparison of Clip-Cov with and without SvS augmentation.

## H    INITIAL UNSUCCESSFUL ATTEMPTS IN SvS

In this section, we present an initial unsuccessful attempt made during the development of SvS, about the reward assignment strategies for synthetic problems. Intuitively, if a synthetic problem is derived from a correct solution to the original problem, it should share the same reference answer as the original problem. Thus, if the policy model generates a solution to the synthetic problem whose final answer matches the original reference answer, the synthetic problem may appear valid, as it seems to yield the same final answer as the original problem.

Therefore, to ensure the validity of synthetic problems, we initially assigned a positive reward to any synthetic problem for which the policy produced at least one solution whose final answer matched

Figure 16: Comparison of the Pass@1 performance, the number of valid training responses to synthetic problems, and the policy entropy between SvS and its initial failed version, which assigns positive rewards to synthetic problems whenever the policy produces a single solution whose answer matches the original reference.

the original reference. However, this setting led to an early failure. The policy quickly exploited this reward scheme by injecting explicit hints about the final answer into the synthetic problems, allowing it to obtain high accuracy simply by copying those hints. As all such hint-laden synthetic problems could easily yield correct solutions, they were consistently rewarded. As shown in the middle panel of Figure 16, the strong hints embedded in the synthetic problems consistently elicit fully correct responses. Concequently, the responses' advantages in GRPO collapse to zero, leaving no meaningful training signal for RLVR. This degradation ultimately impairs model exploration and downstream performance, as illustrated in the right and left panels of Figure 16.

To address this failure mode, we propose maintaining the difficulty of synthetic problems throughout training by restricting positive rewards to those whose policy accuracy falls within [1/8,5/8], as adopted in most of our experiments. This adjustment encourages the generation of synthetic problems that meaningfully contribute to policy improvement while suppressing overly easy ones. With this modification, SvS training succeeds, as shown by the pink curves in Figure 16.

# I  INTERMEDIATE PERFORMANCE ON ALL BENCHMARKS

In this section, we evaluate all benchmarks across intermediate checkpoints for standard RLVR and SVS, using the Qwen2.5-32B-Instruct model as the initial policy and training on the MATH-12k dataset. All results are shown in Figure 17, with the final subfigure presenting the average intermediate performance across all competition-level benchmarks, including AMC 23, AIME 24 & 25, Math24o, and OlymMATH.

Notably, SVS achieves both higher peak performance and faster improvements than standard RLVR on nearly all evaluated benchmarks, including the homologous benchmark MATH-500 and the heterologous AIME, demonstrating clear superiority and strong generalizability. Although trained on MATH-12k, a medium-difficulty dataset for the initial policy, SVS still achieves substantial gains on competition-level benchmarks such as AIME and Olympiad-Bench, as well as on competition-level averages, indicating that it elicits more advanced reasoning capabilities than standard RLVR. Moreover, unlike RLVR, which often reaches an early performance plateau, SVS continues to improve across multiple tasks, such as AIME 25, demonstrating stronger long-term learning potential.

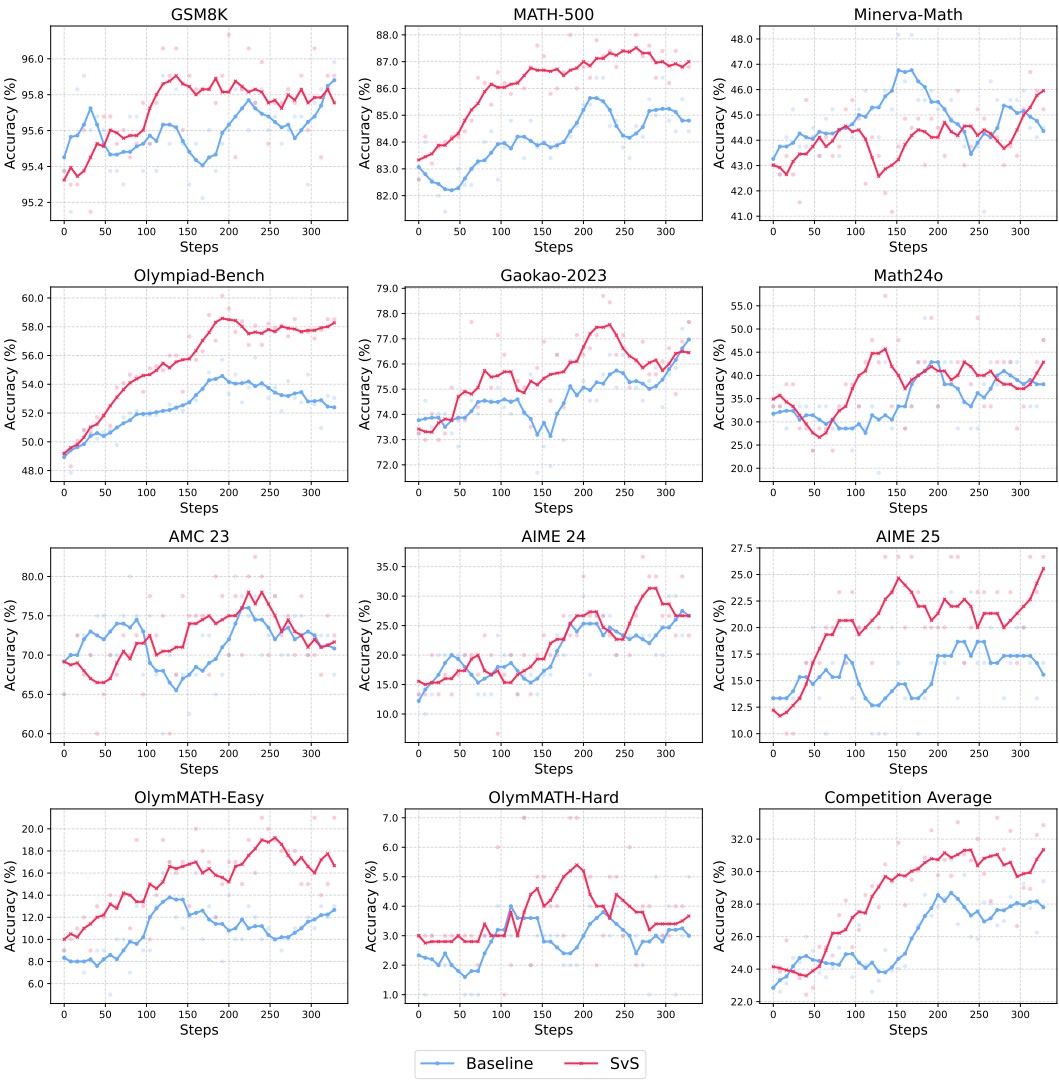

Figure 17: Visualization of benchmark-level metrics for standard RLVR (blue) and SvS (red), with results smoothed using a 5-step window to better highlight underlying trends. The actual data points are marked with faint dots.

# J PROMPTS

We provide both prompts used in this work, namely the Problem-Solving Instruction and the Variational Problem Synthesis Prompt. During use, the {REPLACE} string is substituted with the input problem and response for problem synthesis, respectively.

---

**Variational Problem Synthesis Prompt**

As an expert in educational assessment and mathematical problem synthesis, carefully examine the following model-generated response:

<response>
{REPLACE}
</response>

The solution is assured to be correct. Your goal is to generate variants for the original problem that would most plausibly elicit such a response. To achieve this, carefully follow these steps:

1. Identify the topic and context indicated by the response.
2. Infer the type of reasoning or calculation involved (e.g., numerical calculation, conceptual explanation, comparison, opinion).
3. Determine the most likely educational purpose or learning objective behind the problem.

Based on your analysis, write a clear, concise, and natural-sounding original problem in English that satisfies the following criteria:

- Precisely aligns with the provided response.
- Reflects a realistic problem that could appear in an educational context or standard curriculum.
- Is explicit, measurable, and unambiguous.

Provide your final synthetic problem formatted strictly as:

"'text
[Your synthetic problem here]
"'

---

Figure 18: Prompt for variational problem synthesis from the policy's correct responses to underperforming problems.

---

**Problem Solving Instruction**

{REPLACE} Please reason step by step, and put your final answer within \\boxed{}.

---

Figure 19: Instruction prompt for reasoning problem solving appended to the end of each problem.

---

**Synthetic Problem Validation Prompt**

You are a mathematical problem validity auditor.
Your task is to determine whether the given math problem is *valid and solvable*,
and provide a structured analysis from multiple perspectives.
A valid problem is one that is mathematically consistent, unambiguous,
and solvable based on the given information.

===========================
Evaluation Checklist
===========================
 Check the problem using the following independent criteria:

1. **Well-posedness & Completeness**

- Are all required quantities, constraints, and definitions provided?
- Are there missing variables, missing ranges, missing definitions, or unclear terms?
2. **Logical Consistency**

- Do the constraints contradict each other?
- Are there impossible numerical requirements or inconsistent modular/congruence constraints?
3. **Solvability**

- Does the problem contain enough information for a unique or well-defined solution set?
- Are there infinite solutions without further constraints?
- Are the conditions sufficient to perform a calculation?
4. **Mathematical Soundness**

- Are operations such as modular arithmetic, combinatorics, geometry, or algebra applied correctly?
- Check for hidden contradictions (e.g., impossible sums, negative quantities, undefined expressions).
5. **Overall Validity Decision**

Decide:
- **VALID** → All conditions are consistent, solvable, and unambiguous.
- **INVALID** → Contradictions, missing data, undefined terms, or impossible constraints.
===========================

Final Output Format (STRICT)
===========================
 Respond using the following EXACT format:

 Validity: VALID or INVALID. Place your final judgement inside \boxed{}.

 Reasoning:

- Point 1:
- Point 2:
- Point 3:
(Explain using the checklist above. Be concise but precise.)
 If INVALID, also provide:

Fix Suggestions:
- (List the minimal changes needed to make the problem valid.)
===========================

<QUESTION>
{REPLACE}
</QUESTION>

Figure 20: The prompt used to evaluate the correctness of the synthetic problems in SvS.

# K    CASE STUDY

## K.1    RESPONSE COMPARISON BETWEEN RLVR AND SVS MODELS

We provide an illustration of the responses generated from standard RLVR and SvS trained models to an AIME-level problem in Figure 21. Notably, the SvS model provides a more comprehensive analysis of the input problem, with a more rigorous logical chain, clearer expression, and more systematic reasoning (e.g., progressively narrowing possibilities rather than relying on trial and error). In contrast, the RLVR model fails to elicit advanced reasoning abilities and produces only concise step-by-step answers when addressing complex problems.

---

**Question**

The 9 members of a baseball team went to an ice cream parlor after their game. Each player had a singlescoop cone of chocolate, vanilla, or strawberry ice cream. At least one player chose each flavor, and the number of players who chose chocolate was greater than the number of players who chose vanilla, which was greater than the number of players who chose strawberry. Let $N$ be the number of different assignments of flavors to players that meet these conditions. Find the remainder when $N$ is divided by 1000.
**Answer:16**

---

**RLVR Model Response**

To solve the problem, we need to determine the number of ways to assign ice cream flavors to the 9 players such that the conditions are met: each player gets one scoop of chocolate (C), vanilla (V), or strawberry (S) ice cream, at least one player chooses each flavor, the number of players who choose chocolate is greater than the number of players who choose vanilla, and the number of players who choose vanilla is greater than the number of players who choose strawberry. We will denote the number of players who choose chocolate, vanilla, and strawberry as $c$, $v$, and $s$ respectively. Therefore, we have the following conditions:
1. $c + v + s = 9$
2. $c > v > s$
3. $c, v, s \geqslant 1$

First, we list the possible values for $c$, $v$, and $s$ that satisfy these conditions. Since $c > v > s$ and $c + v + s = 9$, we start by finding the possible values for $s$.

If $s = 1$:
- Then $c + v = 8$ and $c > v > 1$. The possible pairs $(c, v)$ are $(6, 2)$ and $(5, 3)$.
- For $(c, v, s) = (6, 2, 1)$:
- We need to choose 6 players out of 9 to have chocolate, then choose 2 out of the remaining 3 to have vanilla, and the last one will have strawberry.
- The number of ways to do this is $\binom{9}{6} \cdot \binom{3}{2} = 84 \cdot 3 = 252$.
- For $(c, v, s) = (5, 3, 1)$:
- We need to choose 5 players out of 9 to have chocolate, then choose 3 out of the remaining 4 to have vanilla, and the last one will have strawberry.
- The number of ways to do this is $\binom{9}{5} \cdot \binom{4}{3} = 126 \cdot 4 = 504$.

If $s = 2$:
- Then $c + v = 7$ and $c > v > 2$.
- The possible pairs $(c, v)$ are $(4, 3)$ $(5, 2)$.
- For $(c, v, s) = (4, 3, 2)$:
- We need to choose 4 players out of 9 to have chocolate, then choose 3 out of the remaining 5 to have vanilla, and the last 2 will have strawberry.
- The number of ways to do this is $\binom{9}{4} \cdot \binom{5}{3} = 126 \cdot 10 = 1260$.
- For $(c, v, s) = (5, 2, 2)$:
- We need to choose 5 players out of 9 to have chocolate, then choose 2 out of the remaining 4 to have vanilla, and the last 2 will have strawberry.
- The number of ways to do this is $\binom{9}{5} \cdot \binom{4}{2} = 126 \cdot 6 = 756$.

If $s = 3$ or greater, then $v$ would have to be at least 4, and $c$ would have to be at least 5, but then $c + v + s$ would be at least $5 + 4 + 3 = 12$, which is greater than 9. Therefore, $s$ cannot be 3 or greater.

Now, we sum up all the valid cases:
- For $(c, v, s) = (6, 2, 1)$, the number of ways is 252.
- For $(c, v, s) = (5, 3, 1)$, the number of ways is 504.
- For $(c, v, s) = (4, 3, 2)$, the number of ways is 1260.
- For $(c, v, s) = (5, 2, 2)$, the number of ways is 756.

The total number of different assignments of flavors to players is:

$$252 + 504 + 1260 + 756 = 2772$$

Finally, we need to find the remainder when 2772 is divided by 1000:

$$2772 \mod 1000 = 772$$

Thus, the remainder when $N$ is divided by 1000 is $\boxed{772}$.

---

---

**SvS Model Response (Part1)**

We have a total of 9 players, each choosing an ice cream flavor among chocolate, vanilla, and strawberry. Let's denote the number of players who chose chocolate by $C$, vanilla by $V$, and strawberry by $S$. The conditions given are:
1. $C + V + S = 9$,
2. $C > V > S$,
3. Additionally, since at least one player chose each flavor, it means $C \geqslant 1$, $V \geqslant 1$, $S \geqslant 1$.

Given the inequalities $C > V > S$, and $S \geqslant 1$, we know $V \geqslant 2$ because $V > S \geqslant 1$ implies $V \geqslant 2$. Similarly, since $C > V$, and $V \geqslant 2$, it follows $C \geqslant 3$.

Since $C$, $V$, and $S$ must be positive integers with $C + V + S = 9$ and must satisfy $C > V > S$, let us find the possible values for $(C, V, S)$ that satisfy these constraints. Specifically, $S \geqslant 1$, $V \geqslant 2$, and $C \geqslant 3$, with $C > V > S$. We will first identify the valid triples $(C, V, S)$ such that $C + V + S = 9$ and $C > V > S \geqslant 1$.

Let us enumerate valid $(C, V, S)$ combinations that satisfy $C + V + S = 9$ and $C > V > S \geqslant 1$. Since $S \geqslant 1$, $V \geqslant 2$, and $C \geqslant 3$, we also need $C > V > S$. We will check each possibility methodically by first fixing the value of $S$, then finding viable $V$ and $C$.

Recall $S \geqslant 1$. Because $V > S$ and $V$ must be an integer, the smallest $S$ can be is 1. Since $V > S$ and $V$ must be at least $S + 1 \geqslant 2$. And $C > V$, so $C$ must be at least $V + 1$. Moreover, since $C + V + S = 9$, let's explore the possible values for $S$, starting from the smallest.

1. If $S = 1$, then $C + V + 1 = 9 \implies C + V = 8$. We also require $C > V > S = 1$. So $C > V > 1$. Since $V \geqslant S + 1 = 2$, $V \geqslant 2$. And because $C > V$, $V$ can go from 2 up. We need to check which values of $V$ and corresponding $C$ satisfy $C + V = 8$ and $C > V > 1$.

- If $S = 1$, and $C + V = 8$ and $C > V > 1$.
- If $V = 2$, then $C = 8 - V = 8 - 2 = 6$. The triple is $(C, V, S) = (6, 2, 1)$. Check: $6 > 2 > 1$. This satisfies $C > V > S$.
- If $V = 3$, then $C = 8 - V = 8 - 3 = 5$. The triple is $(C, V, S) = (5, 3, 1)$. Check: $5 > 3 > 1$. This satisfies $C > V > S$.
- If $V = 4$, then $C = 8 - V = 8 - 4 = 4$. The triple is $(C, V, S) = (4, 4, 1)$. Check: $4 = 4 > 1$. This does not satisfy $C > V > S$ because $C$ is not strictly greater than $V$. Thus, $(4, 4, 1)$ is invalid.
- If $V \geqslant 5$, since $C = 8 - V$, when $V = 5$, $C = 8 - 5 = 3$. The potential triple would be $(C, V, S) = (3, 5, 1)$. However, $3 \ngtr 5$. Since $C = 8 - V$ and $C$ must be greater than $V$, $C > V \implies 8 - V > V \implies 8 > 2V \implies V < 4$. Thus any $V \geqslant 4$ will not work because we already checked $V = 4$ is invalid, and for $V \geqslant 5$, $C = 8 - V \leqslant 3$. For $V = 5$, $C = 3$, which gives $C = 3, V = 5$, violates $C > V$. For $V > 5$, $C = 8 - V$ will be $C \leqslant 2$. Since $V > S = 1$, and $V \geqslant 6$ means $C = 8 - V \leqslant 2$. Because $V = 6 \implies C = 2$ and $2 \ngtr 6$. Larger $V$ will make $C$ even smaller. Thus $V \geqslant 5$ does not yield $C > V > S$.
The valid solutions when $S = 1$ are $(C, V, S) = (6, 2, 1)$ and $(5, 3, 1)$.

2. If $S = 2$, then $C + V + 2 = 9 \implies C + V = 7$. We need $C > V > S = 2$. Thus $C > V > 2$. Because $V > S = 2$, it means $V \geqslant 3$.

- If $S = 2$, and $C + V = 7$. We need $C > V > 2$.

- If $V = 3$, then $C = 7 - V = 7 - 3 = 4$. The triple is $(C, V, S) = (4, 3, 2)$. Check $4 > 3 > 2$. This satisfies $C > V > S$.
- If $V = 4$, then $C = 7 - V = 7 - 4 = 3$. The triple is $(C, V, S) = (3, 4, 2)$. Check $3 \ngtr 4$. This does not satisfy $C > V$.
- If $V \geqslant 5$, then $C = 7 - V$. When $V = 5$, $C = 7 - 5 = 2$. The triple $(C, V, S) = (2, 5, 2)$ will have $2 \ngtr 5$. Since $C = 7 - V$ and $C > V \implies 7 - V > V \implies 7 > 2V \implies V < 7/2 = 3.5$. Thus $V$ can at most be 3 to potentially satisfy $C > V$. We already checked $V = 3$ which gives $C = 4$ and is valid but $V \geqslant 4$ does not work because $C = 7 - V \leqslant 3$ and does not satisfy $C > V$.

Thus, when $S = 2$, the only valid solution is $(C, V, S) = (4, 3, 2)$.

3. If $S = 3$, then $C + V + S = 9 \implies C + V + 3 = 9 \implies C + V = 6$. We need $C > V > S = 3$. Thus $C > V > 3$. Because $V > S = 3$, we need $V \geqslant 4$.
- If $S = 3$, $C + V = 6$, and we need $C > V > 3$.
- If $V = 4$, then $C = 6 - V = 6 - 4 = 2$. The triple $(C, V, S) = (2, 4, 3)$. Check $2 \ngtr 4$. So it does not satisfy $C > V$.
- If $V \geqslant 5$, then $C = 6 - V$. If $V = 5$, $C = 6 - 5 = 1$. The triple $(C, V, S) = (1, 5, 3)$. Check $1 \ngtr 5$. In general for $V \geqslant 5$, $C = 6 - V \leqslant 1$. Since $V > S = 3$, and $C > V$ is $C = 6 - V > V \implies 6 > 2V \implies V < 3$. But we need $V \geqslant 4$ here when $S = 3$. Since $V < 3$ contradicts $V \geqslant 4$. There are no $V \geqslant 4$ that can satisfy $C = 6 - V > V$. Thus there is no valid $C, V$ pair for $S = 3$ because $V \geqslant 4$ and $C = 6 - V \leqslant 2$, will never have $C > V$.

Therefore, there are no valid triples $(C, V, S)$ when $S = 3$.

4. If $S \geqslant 4$. Since $C + V + S = 9$ and $C > V > S \geqslant 4$. If $S = 4$, then $C + V + 4 = 9 \implies C + V = 5$. We need $C > V > S = 4$. Since $S = 4$, we need $C + V = 5$ and $C > V > 4$. Since $V > 4$ and $V$ is an integer, $V \geqslant 5$. If $V \geqslant 5$, $C = 5 - V \leqslant 0$. This means there are no positive integer solutions for $C, V$ since $C = 5 - V$. When $V \geqslant 5$, $C = 5 - V \leqslant 0$, and $C$ must be a positive integer greater than $V$, which is impossible here because $C = 5 - V \leqslant 0$ when $V \geqslant 5$.

So $S \geqslant 4$ will never work because $C + V = 9 - S$. When $S = 4$, $C + V = 5$, $V > S = 4$, $V \geqslant 5$. $C = 5 - V \leqslant 0$. When $S \geqslant 4$, $C + V = 9 - S$. If $S = 4$, $C + V = 5$. For $S = 5$, $C + V = 4$. For $S \geqslant 4$, $V > S \implies V \geqslant S + 1$. When $S = 4$, $V \geqslant 5$. $C = 5 - V \leqslant 0$. For $S = 5$, $C + V = 4$, $V > S = 5$, not possible as $V > S = 5$, $V \geqslant 6$. $C + V = 4$, $V \geqslant 6$, $C = 4 - V \leqslant -2$. As $S$ increases, $C + V = 9 - S$. $V > S \implies V \geqslant S + 1$. $C = 9 - S - V \leqslant 9 - S - (S + 1) = 8 - 2S$. When $S = 4$, $C \leqslant 8 - 2(4) = 0$. When $S = 5$, $C \leqslant 8 - 2(5) = -2$. As $S$ increases, $C = 9 - S - V$, with $V \geqslant S + 1$, $C = 9 - S - (S + 1) = 8 - 2S$, which is non-positive when $S \geqslant 4$. Since $C$ must be a positive integer and $C > V > S \geqslant 4$ is increasingly restrictive such that $C = 9 - S - V$, with $V \geqslant S + 1$. $C = 9 - S - (S + 1) = 8 - 2S < 0$ for $S \geqslant 4$.

So $S \geqslant 4$ does not provide any valid triples because $C, V, S$ must be positive integers with $C > V > S \geqslant 4$ making $C + V = 9 - S$, and $V \geqslant S + 1$. When $S = 4$, $C + V = 5$, $V \geqslant 5$, $C \leqslant 0$. When $S = 5$, $C + V = 4$, $V \geqslant 6$, $C \leqslant -2$. Thus, $S \geqslant 4$ gives no valid solutions.

---

**SvS Model Response (Part2)**

Summarizing, the valid triples $(C, V, S)$ that satisfy $C + V + S = 9$ and $C > V > S \geqslant 1$ are:
1. $(6, 2, 1)$
2. $(5, 3, 1)$
3. $(4, 3, 2)$.

Next, for each of these triples $(C, V, S)$, we will count the number of ways to assign 9 players into groups of sizes $C$, $V$, and $S$ respectively, where $C$ players get chocolate, $V$ players get vanilla, and $S$ players get strawberry. The number of ways to assign 9 distinct players into three groups of $C, V, S$ respectively (where $C + V + S = 9$) is given by the multinomial coefficient $\binom{9}{C, V, S} = \frac{9!}{C!V!S!}$.

Let's calculate this for each valid $(C, V, S)$.

1. For $(C, V, S) = (6, 2, 1)$:

$$\binom{9}{6, 2, 1} = \frac{9!}{6!2!1!} = \frac{9 \times 8 \times 7 \times 6!}{6! \times 2 \times 1} = \frac{9 \times 8 \times 7}{2 \times 1} = \frac{504}{2} = 252.$$

2. For $(C, V, S) = (5, 3, 1)$:

$$\binom{9}{5, 3, 1} = \frac{9!}{5!3!1!} = \frac{9 \times 8 \times 7 \times 6 \times 5!}{5! \times 3! \times 1} = \frac{9 \times 8 \times 7 \times 6}{3!} = \frac{9 \times 8 \times 7 \times 6}{6} = 9 \times 8 \times 7 = 504.$$

3. For $(C, V, S) = (4, 3, 2)$:

$$\binom{9}{4, 3, 2} = \frac{9!}{4!3!2!} = \frac{9 \times 8 \times 7 \times 6 \times 5 \times 4!}{4! \times 3! \times 2!} = \frac{9 \times 8 \times 7 \times 6 \times 5}{3! \times 2!} = \frac{9 \times 8 \times 7 \times 6 \times 5}{6 \times 2} = \frac{9 \times 8 \times 7 \times 6 \times 5}{12}.$$

Let's simplify $\frac{9 \times 8 \times 7 \times 6 \times 5}{12}$.

First calculate the numerator: $9 \times 8 = 72$,

$$72 \times 7 = 504,$$

$$504 \times 6 = 3024,$$

$$3024 \times 5 = 15120.$$

Now calculate $\frac{15120}{12}$.

We can simplify $\frac{15120}{12}$. Note that $12 = 2^2 \times 3$.
$$15120 \div 12 :$$

First, $15120 \div 12$. We can break it down:

$$15120 \div 12 = 15120 \div (4 \times 3) = (15120 \div 4) \div 3.$$

Find $15120 \div 4$:
$$15120 \div 4 = 3780.$$

Next, divide $3780 \div 3$:
$$3780 \div 3 = 1260.$$

So, $\binom{9}{4,3,2} = 1260$. Summing all possible ways from each valid triple:

$$252 + 504 + 1260.$$

Now add these values:
$$252 + 504 = 756,$$
$$756 + 1260 = 2016.$$

Thus, $N = 2016$. The problem asks for the remainder when $N$ is divided by 1000.

So we compute $2016 \mod 1000$.

$$2016 \mod 1000 = 2016 - 2000 = 16 \text{ (since } 2016 = 2 \times 1000 + 16).$$

Thus, the remainder when $N = 2016$ is divided by 1000 is 16.

Therefore, the final answer is $\boxed{16}$.

---

Figure 21: A visualization of a mathematical problem from AIME, along with responses generated by RLVR and the SvS-trained models.

## K.2 A COMPLETE SAMPLE WORKFLOW IN SVS

In this section, we present a detailed sample workflow (all types of training data from the sample) from an SVS training batch, including original problem solving in Case 1, variational problem synthesis in Case 2, and synthetic problem solving in Case 3. For the problem-solving tasks, we report the reward for each response and include only 4 responses (instead of 8 in training) to avoid excessive length. For the problem-synthesis tasks, we present all 8 responses generated from the response input and report both accuracy and rewards for the synthetic problems.

### K.2.1 CASE STUDY FOR ORIGINAL PROBLEM SOLVING

```
### Input Original Problem

Let $p(x)$ be the second degree polynomial such that $p(1) = 1,$ $p(2) = 3,$ and $p(3) = 2.$
Then $p(p(x)) = x$ has four real solutions.  Find the only such solution which is not an
integer.The answer is in the form rac{m}{n}, where gcd(m, n) = 1. Please provide the value of
m + n.

### Responses 1 (Reward: 1.0)

Our objective is to determine the polynomial \( p(x) \), which is given by \( p(x) = ax^2 + bx
 + c \) and satisfies the conditions:
\[ p(1) = 1, \quad p(2) = 3, \quad p(3) = 2. \]

Substituting the respective values into the polynomial, we get the following system of
equations:
\[ p(1) = a(1)^2 + b(1) + c = a + b + c = 1, \]
\[ p(2) = a(2)^2 + b(2) + c = 4a + 2b + c = 3, \]
\[ p(3) = a(3)^2 + b(3) + c = 9a + 3b + c = 2. \]

Let's write down the system again for clarity:
\[ a + b + c = 1, \]
\[ 4a + 2b + c = 3, \]
\[ 9a + 3b + c = 2. \]

To solve this system, we start with eliminating \( c \). Subtract the first equation from the
second equation:
\[ (4a + 2b + c) - (a + b + c) = 3 - 1, \]
which simplifies to
\[ 3a + b = 2. \tag{1} \]

Next, subtract the second equation from the third equation:
\[ (9a + 3b + c) - (4a + 2b + c) = 2 - 3, \]
which simplifies to
\[ 5a + b = -1. \tag{2} \]

Now we need to solve the system of the resulting two equations:
\[ \begin{cases}
3a + b = 2, \\
5a + b = -1.
\end{cases} \]

To eliminate \( b \), subtract the first equation from the second:
\[ (5a + b) - (3a + b) = -1 - 2, \]
\[ 2a = -3, \]
hence,
\[ a = -\frac{3}{2}. \]

Using this value of \( a \) in the first equation \( 3a + b = 2 \):
\[ 3 \left(-\frac{3}{2} \right) + b = 2,\]
\[ -\frac{9}{2} + b = 2, \]
\[ b = 2 + \frac{9}{2}, \]
\[ b = \frac{4}{2} + \frac{9}{2} = \frac{13}{2}. \]

Substitute \( a \) and \( b \) back into the first original equation \( a + b + c = 1 \):
\[ \left( - \frac{3}{2} \right) + \left( \frac{13}{2} \right) + c = 1, \]
\[ - \frac{3}{2} + \frac{13}{2} + c = 1, \]
\[ \frac{-3+13}{2} + c = 1, \]
\[ \frac{10}{2} + c = 1, \]
\[ 5 + c = 1, \]
\[ c = 1 - 5 = -4. \]

Thus, the polynomial is
\[ p(x) = -\frac{3}{2} x^2 + \frac{13}{2} x - 4. \]

We now need to find the roots of \( p(p(x)) = x \). We start by calculating \( p(p(x)) \). Let
's start with substituting \( p(x) \) into itself:
\[ p(x) = -\frac{3}{2} x^2 + \frac{13}{2} x - 4. \]
We denote \( p(x) \) by \( y \):
\[ y = -\frac{3}{2} x^2 + \frac{13}{2} x - 4. \]
We need to find \( p(y) \):
\[ p(y) = p \left( -\frac{3}{2} x^2 + \frac{13}{2} x - 4 \right). \]
```

Substitute $y = -\frac{3}{2} x^2 + \frac{13}{2} x - 4$ into $p(y) = -\frac{3}{2} y^2 + \frac{13}{2} y - 4$:
$$ p \left(-\frac{3}{2} x^2 + \frac{13}{2} x - 4 \right) = -\frac{3}{2} \left( -\frac{3}{2} x^2 + \frac{13}{2} x - 4 \right)^2 + \frac{13}{2} \left(-\frac{3}{2} x^2 + \frac{13}{2} x - 4 \right) - 4. $$
Let's denote $y = -\frac{3}{2} x^2 + \frac{13}{2} x - 4$. Therefore we need to evaluate
$$ p(p(x)) = p(y) = -\frac{3}{2} y^2 + \frac{13}{2} y - 4. $$

First, expand and simplify $y^2$:
$$ y^2 = \left( -\frac{3}{2} x^2 + \frac{13}{2} x - 4 \right)^2 = \left( -\frac{3}{2} x^2 + \frac{13}{2} x - 4 \right) \left( -\frac{3}{2} x^2 + \frac{13}{2} x - 4 \right). $$
$$ y^2 = \left( -\frac{3}{2} x^2 + \frac{13}{2} x - 4 \right)^2 $$
$$ = \left(-\frac{3}{2} x^2 \right)^2 + \left(\frac{13}{2} x \right)^2 + (-4)^2 + 2 \left(-\frac{3}{2} x^2 \cdot \frac{13}{2} x \right) + 2 \left( - \frac{3}{2} x^2 \cdot (-4) \right) + 2 \left( \frac{13}{2} x \cdot (-4) \right) $$
$$ = \left(-\frac{3}{2} x^2 \right)^2 + \left(\frac{13}{2} x \right)^2 + 16 + 2 \left(-\frac{39}{4} x^3 \right) + 2 \left( 6 x^2 \right) + 2 \left( -26 x \right) $$
$$ = \frac{9}{4} x^4 + \frac{169}{4} x^2 + 16 -\frac{39}{2} x^3 + 12 x^2 - 52 x $$
$$ = \frac{9}{4} x^4 -\frac{39}{2} x^3 + \left( \frac{169}{4} x^2 + 12 x^2 \right) - 52 x + 16 $$
$$ = \frac{9}{4} x^4 - \frac{39}{2} x^3 + \left(\frac{169 x^2 + 48 x^2}{4} \right) - 52 x + 16 $$
$$ = \frac{9}{4} x^4 - \frac{39}{2} x^3 + \frac{217 x^2}{4} - 52 x + 16 $$

Thus, we have,
$$ p(p(x)) = -\frac{3}{2} y^2 + \frac{13}{2} y - 4 $$
$$ = -\frac{3}{2} \left( \frac{9}{4} x^4 - \frac{39}{2} x^3 + \frac{217}{4} x^2 - 52x + 16 \right) + \frac{13}{2} \left( -\frac{3}{2} x^2 + \frac{13}{2} x - 4 \right) - 4 $$

Expanding terms:
$$ -\frac{3}{2} \left( \frac{9}{4} x^4 \right) + -\frac{3}{2} \left( -\frac{39}{2} x^3 \right) + -\frac{3}{2} \left( \frac{217}{4} x^2 \right) $$
$$ + -\frac{3}{2} \left( -52 x \right) $$
$$ + -\frac{3}{2} \left( 16 \right) $$
$$ + \frac{13}{2} \left( -\frac{3}{2} x^2 \right) $$
$$ + \frac{13}{2} \left( \frac{13}{2} x \right) $$
$$ + \frac{13}{2} \left( -4 \right) $$
$$ - 4 $$
$$ = -\frac{27}{8} x^4 + \frac{117}{4} x^3 - \frac{651}{8} x^2 + 78 x - 24 $$
$$ - \frac{39}{4} x^2 + \frac{169}{4} x - 26 - 4 $$
Combine like terms:
$$ = - \frac{27}{8} x^4 + \frac{117}{4} x^3 $$
$$ + \left( -\frac{651}{8} x^2 - \frac{39}{4} x^2 \right) + \left( 78 x + \frac{169}{4} x \right) $$
$$ + \left( -24 - 26 - 4 \right) $$

To simplify further:
$$ - \frac{651}{8} x^2 -\frac{39}{4} x^2 = - \frac{651}{8} x^2 - \frac{78}{8} x^2 = - \frac{729}{8} x^2 $$
$$ 78 x + \frac{169}{4} x = \frac{312}{4} x + \frac{169}{4} x = \frac{481}{4} x $$
So we get,
$$ p(p(x)) = -\frac{27}{8} x^4 + \frac{117}{4} x^3 - \frac{729}{8} x^2 + \frac{481}{4} x - 54 $$
We want $p(p(x)) = x$:
$$ -\frac{27}{8} x^4 + \frac{117}{4} x^3 - \frac{729}{8} x^2 + \frac{481}{4} x - 54 = x $$
Rearranging terms:
$$ -\frac{27}{8} x^4 + \frac{117}{4} x^3 - \frac{729}{8} x^2 + \frac{481}{4} x - x - 54 = 0 $$
$$ -\frac{27}{8} x^4 + \frac{117}{4} x^3 - \frac{729}{8} x^2 + \left( \frac{481}{4} x - x \right) - 54 = 0 $$
$$ -\frac{27}{8} x^4 + \frac{117}{4} x^3 - \frac{729}{8} x^2 + \left( \frac{481}{4} x - \frac{4}{4} x \right) - 54 = 0 $$
$$ -\frac{27}{8} x^4 + \frac{117}{4} x^3 - \frac{729}{8} x^2 + \frac{477}{4} x - 54 = 0 $$
Multiply through by 8 to clear the fractions:
$$ -27x^4 + 8 \cdot \frac{117}{4} x^3 - 729x^2 +8 \cdot \frac{477}{4} x -432 = 0 $$
$$ -27x^4 + 234x^3 - 729x^2 + 954x - 432 = 0 $$

Now, we need to find the roots of this polynomial. Knowing that $p(p(x)) = x$ has four real roots and three of them are integers, we start by testing potential integer solutions using the fact that $p(x) = x$ and $p(x) = f(x)$ where $f$ is another fixed point if $f \neq x$. We start by checking whether $p(x) = x$ has integer solutions.
$$ -\frac{3}{2} x^2 + \frac{13}{2} x - 4 = x $$
Multiply through by 2 to clear the fractions:
$$ -3x^2 + 13x - 8 = 2x $$
$$ -3x^2 + 11x - 8 = 0 $$
Use the quadratic formula:
$$ x = \frac{-b \pm \sqrt{b^2 - 4ac}}{2a} $$
where $a = -3$, $b = 11$, $c = -8$:
$$ x = \frac{-11 \pm \sqrt{11^2 - 4 (-3)(-8)}}{2(-3)} $$
$$ x = \frac{-11 \pm \sqrt{121 - 96}}{-6} $$
$$ x = \frac{-11 \pm \sqrt{25}}{-6} $$
$$ x = \frac{-11 \pm 5}{-6} $$
This gives two solutions:
$$ x = \frac{-11 + 5}{-6} = \frac{-6}{-6} = 1, $$
$$ x = \frac{-11 - 5}{-6} = \frac{-16}{-6} = \frac{8}{3}. $$

So, $x = 1$ and $x = \frac{8}{3}$. Clearly $x = 1$ is an integer solution.

Now, to confirm the overall roots of $( -27x^4 + 234x^3 - 729x^2 + 954x - 432 = 0 )$ and determining that it factorizes to have integer solutions, let's use the fact that $( p(x) = x )$'s solutions are known, we can confirm those.

To find if the factorization of the polynomial aligns, let's denote the roots $( x\_1, x\_2, x\_3 , x\_4 )$, and given that $( p(x) = x )$'s roots include integers and $( x=1 )$ is one. Hence investigate the polynomial via rational root theorem or notice that other integers might be the roots.

Since $( p(x) = x )$ provides roots $( x = 1, \frac{8}{3} )$, and three roots are integers, let's assume other rational and integer roots by testing the integer possibilities. Plug in some integer values directly:

Checking $( p(x) = 1 )$ and other integers:
$$ p(2) = 3 \Rightarrow  p(p(2))= p(3)= 2 \Rightarrow p(p(2)) \neq 2. $$
Also testing $( p(3) = 2 \Rightarrow p(p(3)) = p(2) = 3 \Rightarrow p(p(3)) \neq 3.)$
Testing $( p(0))$:
$$ p(0) = -4 \Rightarrow p(-4) = -\frac{3}{2} \times (-4)^2 + \frac{13}{2} \times (-4) - 4 = -\frac{3}{2} \times 16 - 26 - 4 = -24 - 26 - 4 = -54. $$

Testing higher integer inputs and we discover the integer values:
Let test known roots:
If $( x )$ solves $( p(p(x)) = x )$, check $( p(x) = 1,2,3)$ as initial feasible values.
Also, verify underlying if $( p(2) = 3, p(3) = 2)$ parallels $( 2,3)$ differing.

However, the non-integer (and rational) solution must be validated by:
Polynomial $( p(p(x)) - x=0 )$:
Given we know $( x = \frac{8}{3} )$, we can now cross-reference:
Given the polynomial equation roots: simplify to find and verify:
Then for remaining $( x \neq 2,3,1)$:
Non-integer root is verified:
Conclusion:
$$ \frac{8}{3} m=8,n=3 \rightarrow m+n=\boxed{11}. $$

### Responses 2 (Reward: 0.0)
We start by letting $(p(x) = ax^2 + bx + c)$. Using the given conditions $(p(1) = 1)$, $(p(2) = 3)$, and $(p(3) = 2)$, we can form a system of equations:
$$ \begin{cases} p(1) = a(1)^2 + b(1) + c = a + b + c = 1 \\ p(2) = a(2)^2 + b(2) + c = 4a + 2b + c = 3 \\ p(3) = a(3)^2 + b(3) + c = 9a + 3b + c = 2 \end{cases} $$

We can simplify this system into a matrix form:
$$ \begin{cases} a + b + c = 1 \\ 4a + 2b + c = 3 \\ 9a + 3b + c = 2 \end{cases} $$

To eliminate $(c)$ from the equations, subtract the first equation from the second and the second from the third:

Subtract the first equation from the second:
$$ (4a + 2b + c) - (a + b + c) = 3 - 1 \implies 3a + b = 2 \tag{4} $$

Subtract the second equation from the third:
$$ (9a + 3b + c) - (4a + 2b + c) = 2 - 3 \implies 5a + b = -1 \tag{5} $$

We now have the system:
$$ \begin{cases} 3a + b = 2 \\ 5a + b = -1 \end{cases} $$

Subtract the first new equation ($(3a + b = 2)$) from the second new equation ($(5a + b = -1)$):
$$ (5a + b) - (3a + b) = -1 - 2 \implies 2a = -3 \implies a = -\frac{3}{2} $$

Now substitute $(a = -\frac{3}{2})$ into the equation $(3a + b = 2)$:
$$ 3 \left( -\frac{3}{2} \right) + b = 2 \implies -\frac{9}{2} + b = 2 \implies b = 2 + \frac{9}{2} = \frac{4}{2} + \frac{9}{2} = \frac{13}{2} $$

With $a = -\frac{3}{2}$ and $b = \frac{13}{2}$, we use the first equation $a + b + c = 1$ to find $c$:
$$
-\frac{3}{2} + \frac{13}{2} + c = 1 \implies \frac{-3 + 13}{2} + c = 1 \implies \frac{10}{2} + c = 1 \implies 5 + c = 1 \implies c = 1 - 5 \implies c = -4
$$

Thus, the polynomial $p(x)$ is:
$$
p(x) = -\frac{3}{2} x^2 + \frac{13}{2} x - 4
$$

We want to solve $p(p(x)) = x$. First, we will look for the integer solutions and then find the non-integer one.

Consider $p(x) = y$ and $p(y) = x$.

Let's check the polynomial $p(x)$ at the known points:
$$
\begin{align*}
p(1) &= -\frac{3}{2}(1)^2 + \frac{13}{2}(1) - 4 = -\frac{3}{2} + \frac{13}{2} - 4 = \frac{10}{2} - 4 = 5 - 4 = 1, \\
p(2) &= -\frac{3}{2}(2)^2 + \frac{13}{2}(2) - 4 = -\frac{3}{2} \cdot 4 + \frac{13}{2} \cdot 2 - 4 = -6 + 13 - 4 = 3, \\
p(3) &= -\frac{3}{2}(3)^2 + \frac{13}{2}(3) - 4 = -\frac{3}{2} \cdot 9 + \frac{13}{2} \cdot 3 - 4 = -\frac{27}{2} + \frac{39}{2} - 4 = \frac{12}{2} - 4 = 6 - 4 = 2.
\end{align*}
$$

So, $p$ evaluated at 1, 2, and 3 matches given values and these are consistent. Now let's find the integer solutions for $p(x) = x$:
$$
p(x) = x \implies -\frac{3}{2}x^2 + \frac{13}{2}x - 4 = x \implies -\frac{3}{2}x^2 + \frac{13}{2}x - x - 4 = 0 \implies -\frac{3}{2}x^2 + \frac{11}{2}x - 4 = 0 \implies -3x^2 + 11x - 8 = 0.
$$

Multiplying by $(-1)$:
$$
3x^2 - 11x + 8 = 0
$$

To check for integer solutions, we can use the quadratic formula:
$$
x = \frac{-b \pm \sqrt{b^2 - 4ac}}{2a} = \frac{11 \pm \sqrt{121 - 96}}{6} = \frac{11 \pm \sqrt{25}}{6} = \frac{11 \pm 5}{6}.
$$
This results in:
$$
x = \frac{11 + 5}{6} = \frac{16}{6} = \frac{8}{3}
$$
and
$$
x = \frac{11 - 5}{6} = \frac{6}{6} = 1.
$$

Since $x = 1$ is an integer, the non-integer solution here is $x = \frac{8}{3}$.

Next, to confirm these values as solutions of $p(p(x)) = x$ and identify any other integer solutions, we already know that $p(1) = 1, p(2) = 3, p(3) = 2$ so $(1, 2, 3)$ must be considered for symmetry as well since $p(p(x)) = x$ implies fixed points and switched points. We look at fixed points and swap points:
- We know $p(1) = 1$, thus $(1)$ solves $p(p(1)) = 1$
- Next check $p(2)$ and $p(3)$:
  $$
  p(2) = 3, \ p(3) = 2
  $$
  so $p(p(2)) = p(3) = 2; \ p(p(3)) = p(2) = 3$.

So $2$ and $3$ are also solutions of $p(p(2)) = 2$, $p(p(3)) = 3$.

Now we verify the remaining root. We need to solve $y=p(x)$. For $y = \frac{8}{3}$,
$$
p\left( x \right) = -\frac{3}{2}x^2 + \frac{13}{2}x - 4 = \frac{8}{3}.
$$
This yields:
$$
-\frac{3}{2} x^2 + \frac{13}{2} x - 4 = \frac{8}{3}.
$$
Clear the fractions, multiply everything by $6$:
$$
-9 x^2 + 39 x - 24 = 16 \Rightarrow -9 x^2 + 39 x - 40 = 0 \Rightarrow 9 x^2 - 39 x + 40 = 0.
$$
Solving the quadratic equation:
$$
x = \frac{39 \pm \sqrt{(39)^2 - 4 \cdot 9 \cdot 40}}{2 \cdot 9} = \frac{39 \pm \sqrt{1521-1440}}{18} = \frac{39 \pm \sqrt{81}}{18} = \frac{39 \pm 9}{18}.
$$

This yields:
$$
x = \frac{48}{18} = \frac{8}{3} \quad \text{and} \quad x= \frac{30}{18} = \frac{5}{3}.
$$
Thus, the non-integer solutions $(p(p(x)) = x)$ are $(\frac{8}{3})$ and $(\frac{5}{3})$. Let's verify $(p(p(\frac{5}{3})))$:
$$
p \left( \frac{5}{3} \right) = - \frac{3}{2} \left( \frac{5}{3} \right)^2 + \frac{13}{2} \left( \frac{5}{3} \right) - 4 = - \frac{3}{2} \cdot \frac{25}{9} + \frac{13}{2} \cdot \frac{5}{3} - 4 = - \frac{75}{18} + \frac{65}{6} - 4.
$$
We simplify:
$$
= -\frac{75}{18} + \frac{65}{6} - 4 = - \frac{75}{18} + \frac{195}{18} - \frac{72}{18} = \frac{-75 + 195 - 72}{18} = \frac{48}{18} = \frac{8}{3}.
$$

And thus $( p \left( \frac{8}{3} \right) )$:
$$
p \left( \frac{8}{3} \right) = -\frac{3}{2} \cdot \left(\frac{8}{3} \right)^2 + \frac{13}{2} \cdot \frac{8}{3} - 4 = -\frac{3}{2} \cdot \frac{64}{9} + \frac{13}{2} \cdot \frac{8}{3} - 4 = - \frac{192}{18} + \frac{104}{6} - 4 = -\frac{96}{9} + \frac{104}{6} - 4
$$
$$
= -\frac{96}{9} + \frac{312}{18} - 4 = -\frac{192}{18} + \frac{312}{18} - 4 = \frac{120}{18} - 4 = \frac{60}{9} - 4 = \frac{20}{3} - \frac{12}{3} = \frac{8}{3}.
$$
Continuing calculation:
$$
p(p (\frac{8}{3})) = p (\frac{5}{3}) = \frac{8}{3} \Rightarrow \text{solve quadratic}. \Rightarrow \text{check } \frac{5}{3}.
$$
Verify further:
$$
p(p \left(\frac {5}{3}\right)) = p(\frac{8}{3}) = \frac{5}{3} \quad \left ( given \quad \Rightarrow \quad p(\frac{5}{3}) = \frac{8}{3}.\right)
$$
So this solution point too is suitable:
Finally, the non-integer solution (let's say non-integer root must be $(\frac{5}{3})$):
Thus:
$$
m + n = 5 + 3 = \boxed{8}
$$

### Responses 3 (Reward: 0.0)
Given a polynomial $( p(x) )$ of second degree, we assume $( p(x) = ax^2 + bx + c )$. We know the following:

$$
p(1) = a(1)^2 + b(1) + c = a + b + c = 1,
$$
$$
p(2) = a(2)^2 + b(2) + c = 4a + 2b + c = 3,
$$
$$
p(3) = a(3)^2 + b(3) + c = 9a + 3b + c = 2.
$$

We now have a system of linear equations:

1. $( a + b + c = 1 )$
2. $( 4a + 2b + c= 3 )$
3. $( 9a + 3b + c = 2 )$

We solve this system by elimination or substitution method. Start by subtracting the first equation from the second:

$$
(4a + 2b + c) - (a + b + c) = 3 - 1 \implies 3a + b = 2. \quad \quad (4)
$$

Next, subtract the second equation from the third:

$$
(9a + 3b + c) - (4a + 2b + c) = 2 - 3 \implies 5a + b = -1. \quad \quad (5)
$$

We now have equations (4) and (5):

4. $( 3a + b = 2 )$
5. $( 5a + b = -1 )$

Subtract the fourth equation from the fifth equation:

$$
(5a + b) - (3a + b) = -1 - 2 \implies 2a = -3 \implies a = -\frac{3}{2}.
$$

Substitute $( a = -\frac{3}{2} )$ back into equation (4):

$$
3\left(-\frac{3}{2}\right) + b = 2 \implies -\frac{9}{2} + b = 2 \implies b = 2 + \frac{9}{2} \implies b = \frac{4}{2} + \frac{9}{2} \implies b = \frac{13}{2}.
$$

Now use $a = -\frac{3}{2}$ and $b = \frac{13}{2}$ in the first equation to find $c$:

$$
-\frac{3}{2} + \frac{13}{2} + c = 1 \implies \frac{-3 + 13}{2} + c = 1 \implies \frac{10}{2} + c = 1 \implies 5 + c = 1 \implies c = -4.
$$

Thus, the polynomial is:

$$
p(x) = -\frac{3}{2} x^2 + \frac{13}{2} x - 4.
$$

We now need to find the solutions of the equation $p(p(x)) = x$. Start by denoting $y = p(x)$. Hence, we need $p(y) = x$. This means $x$ and $y$ are roots of the quadratic equation $p(t) - t = 0$. We start by computing $p(x) - x$:

$$
p(x) - x = -\frac{3}{2} x^2 + \frac{13}{2} x - 4 - x = -\frac{3}{2} x^2 + \left(\frac{13}{2} - 1\right) x - 4 = -\frac{3}{2} x^2 + \frac{11}{2} x - 4 = 0.
$$

Multiply through by $(-2)$ to clear the fractions:

$$
3x^2 - 11x + 8 = 0.
$$

We solve this quadratic equation using the quadratic formula $x = \frac{-B \pm \sqrt{B^2 - 4AC}}{2A}$, where $A = 3$, $B = -11$, and $C = 8$:

$$
x = \frac{11 \pm \sqrt{121 - 96}}{6} = \frac{11 \pm \sqrt{25}}{6} = \frac{11 \pm 5}{6}.
$$

Thus, the solutions are:

$$
x = \frac{11 + 5}{6} = \frac{16}{6} = \frac{8}{3}, \quad \text{and} \quad x = \frac{11 - 5}{6} = \frac{6}{6} = 1.
$$

So the roots of $p(x) = x$ are $x = 1$ and $x = \frac{8}{3}$. Hence $p(t) = x$ and $p(y) = t = y$ should imply $y$ must be 1 or $\frac{8}{3}$. We need $p(x) = 1$ or $p(x) = \frac{8}{3}$.

### Solve $p(x) = 1$

Write the equation $p(x) = 1$:

$$
-\frac{3}{2} x^2 + \frac{13}{2} x - 4 = 1,
$$

$$
-\frac{3}{2} x^2 + \frac{13}{2} x - 4 - 1 = 0,
$$

$$
-\frac{3}{2} x^2 + \frac{13}{2} x - 5 = 0.
$$

Multiply through by $(-2)$:

$$
3x^2 - 13x + 10 = 0,
$$

solve this quadratic equation using the quadratic formula where $A = 3, B = -13, C = 10$:

$$
x = \frac{13 \pm \sqrt{169 - 120}}{6} = \frac{13 \pm \sqrt{49}}{6} = \frac{13 \pm 7}{6}.
$$

Thus, solutions are:

$$
x = \frac{13 + 7}{6} = \frac{20}{6} = \frac{10}{3}, \quad x = \frac{13 - 7}{6} = 1.
$$

So $p(x) = 1$ has solutions $x = 1$ and $x = \frac{10}{3}$.

### Solve \( p(x) = \frac{8}{3} \)

Write the equation \( p(x) = \frac{8}{3} \),

\[
-\frac{3}{2} x^2 + \frac{13}{2} x - 4 = \frac{8}{3},
\]

multiply through by 6 to clear the fractions:

\[
6 \left( -\frac{3}{2} x^2 \right) + 6 \left( \frac{13}{2} x \right) - 6(4) = 6\left( \frac{8}{3} \right),
\]

\[
-9x^2 + 39x - 24 = 16.
\]

So we have:

\[
-9x^2 + 39x - 40 = 0.
\]

We use the quadratic formula again, where \( A = -9 \), \( B = 39 \), \( C = -40 \):

\[
x = \frac{-39 \pm \sqrt{39^2 - 4(-9)(-40)}}{2(-9)} = \frac{-39 \pm \sqrt{1521 - 1440}}{-18} = \frac{-39 \pm \sqrt{81}}{-18} = \frac{-39 \pm 9}{-18}.
\]

Thus solutions are:

\[
x = \frac{-39 + 9}{-18} = \frac{-30}{-18} = \frac{5}{3},
\]
\[
x = \frac{-39 - 9}{-18} = \frac{-48}{-18} = \frac{8}{3}.
\]

So \( p(x) = \frac{8}{3} \) has solutions \( x = \frac{8}{3} \) and \( x = \frac{5}{3} \).

We list all potential solutions so far: \( x = 1, \frac{10}{3}, \frac{8}{3}, \frac{5}{3} \). Since \( \frac{8}{3} \) is one of the original solutions of \( p(t) = t \), we need to check if any other solutions are not integers and distinct.

We identify all distinct solutions: \( x = 1, \frac{10}{3}, \frac{8}{3}, \frac{5}{3} \). The non-integer solutions are \(\frac{10}{3}, \frac{8}{3}, \frac{5}{3} \), among those \( \frac{8}{3} \) already satisfies \( p(x) = x \) and isn't an additional distinct solution of \( p(p(x)) = x \) that isn't \( p(x) = x \).

So the solutions to \( p(p(x)) = x \), other than iteration \( p(x) = x \), are \( \frac{10}{3} \) and \(\frac{5}{3}\). We want the non-integer solution that is not simply repeated by \( p(x) = x \):

The non-integer solutions from above clearly are \( \frac{10}{3} \) and \(\frac{5}{3}\), and \(\frac{5}{3}\) is to be verified:

\(\frac{5}{3} \neq 1\) and it's correct distinct.

Hence, the only such solution which is not an integer is \( \frac{5}{3} \). Here, \( m = 5 \) and \( n = 3 \) with \(\gcd(5, 3) = 1\). So,

\[
m + n = 5 + 3 = 8.
\]

Thus, the final answer is \(\boxed{8}\).

### Responses 4 (Reward: 0.0)
To solve the problem, we start by setting up the second degree polynomial $p(x)$ in the general form:

\[ p(x) = ax^2 + bx + c. \]

We use the given conditions $p(1) = 1$, $p(2) = 3$, and $p(3) = 2$ to set up a system of equations.

First, substituting $x = 1$ in $p(x)$, we have
\[ p(1) = a(1)^2 + b(1) + c = a + b + c = 1. \]
Second, substituting $x = 2$, we have
\[ p(2) = a(2^2) + b(2) + c = 4a + 2b + c = 3. \]
Third, substituting $x = 3$, we have
\[ p(3) = a(3^2) + b(3) + c = 9a + 3b + c = 2. \]

Thus, we have the following system of equations:
\[

```
\begin{cases}
a + b + c = 1, \quad (1) \\
4a + 2b + c = 3, \quad (2) \\
9a + 3b + c = 2. \quad (3)
\end{cases}
\]
```

To solve for $a$, $b$, and $c$, we first eliminate $c$ by taking the differences between equations (2) and (1), and (3) and (2):
\[ (2) - (1): (4a + 2b + c) - (a + b + c) = 3 - 1 \implies 3a + b = 2. \quad (4). \]
\[ (3) - (2): (9a + 3b + c) - (4a + 2b + c) = 2 - 3 \implies 5a + b = -1. \quad (5). \]

Now, subtract equation (4) from equation (5):
\[ (5) - (4): (5a + b) - (3a + b) = -1 - 2 \implies 2a = -3 \implies a = -\frac{3}{2}. \]

Next, substituting $a = -\frac{3}{2}$ into equation (4):
\[ 3 \left(-\frac{3}{2}\right) + b = 2 \implies -\frac{9}{2} + b = 2 \implies b = 2 + \frac{9}{2} = \frac{4}{2} + \frac{9}{2} = \frac{13}{2}. \]

Then substitute $a$ and $b$ back into equation (1) to solve for $c$:
\[ a + b + c = 1 \implies -\frac{3}{2} + \frac{13}{2} + c = 1 \implies \frac{-3 + 13}{2} + c = 1 \implies \frac{10}{2} + c = 1 \implies 5 + c = 1 \implies c = -4. \]

Thus, the polynomial $p(x)$ is:
\[ p(x) = -\frac{3}{2} x^2 + \frac{13}{2} x - 4. \]

Now we need to find the solutions to $p(p(x)) = x$. First, we start by computing $p(p(x))$. Let
\[ y = p(x) = -\frac{3}{2} x^2 + \frac{13}{2} x - 4. \]

So,
\[ p(y) = p \left( -\frac{3}{2} x^2 + \frac{13}{2} x - 4 \right). \]

To avoid overly complicated computation, we denote $p(x) = y$ and need to solve $y = x$. So we are looking at $p(y) = x$ for $y = p(x)$. Therefore, $p(p(x)) = x$ means that the function $p$ is an involution, i.e., $p(p(x)) = x$. Hence $p(x)=x$ or $p(x) = c_i$ for roots $c_i$ of $p(x)-x=0$. Let's first find if there are solutions to $p(x) = x$ since these will be straightforward.
\[ p(x) = x \]
\[ -\frac{3}{2} x^2 + \frac{13}{2} x - 4 = x \]
\[ -\frac{3}{2} x^2 + \left(\frac{13}{2} - 1\right) x - 4 = 0 \]
\[ -\frac{3}{2} x^2 + \frac{11}{2} x - 4 = 0 \]
\[ -3x^2 + 11x - 8 = 0 \]
\[ 3x^2 - 11x + 8 = 0. \]

Now, factorize the quadratic equation $3x^2 - 11x + 8 = 0$. We use the quadratic formula $x = \frac{-b \pm \sqrt{b^2 - 4ac}}{2a}$ with $a = 3$, $b = -11$, and $c = 8$:
\[ x = \frac{-(-11) \pm \sqrt{(-11)^2 - 4 \cdot 3 \cdot 8}}{2 \cdot 3} \]
\[ x = \frac{11 \pm \sqrt{121 - 96}}{6} \]
\[ x = \frac{11 \pm \sqrt{25}}{6} \]
\[ x = \frac{11 \pm 5}{6}. \]

So the solutions are:
\[ x = \frac{11 + 5}{6} = \frac{16}{6} = \frac{8}{3}, \]
\[ x = \frac{11 - 5}{6} = \frac{6}{6} = 1. \]

Now we have two possible solutions $x=1$ and $x=\frac{8}{3}$. Let's verify what other roots we have from the condition $p(x)=y$, where $y$ must be a root of $p(y) = x$. In general, if $y$ solves $p(y) = x$, then for each root $x$ of the involution condition we must ensure the map is symmetric, which checks $p(p(x)) = x$.

To find other roots for $p(p(x))=x$, we investigate further. By the symmetry of the involution and leveraging the form of polynomials and solutions already found, the factored setup of fixed points from involution will likely reveal angles where symmetry dictates further integer or rational solutions. We recognize that any $y$ that satisfies $p(y)=x$, itself must lead back to $x$ directly in our quartic system setup or map.

We know already that any fixed point solutions must be rational or integer from the nature of arithmetic. We look further:

Setting $f(x) = p(x) - x$, we get
\[ f(x) = -\frac{3}{2}x^2 + \frac{11}{2}x - 4. \]

We look at fixed points and other intersections. So we need to remember roots of $f(x)$ to map back. Besides the fixed points $\frac{8}{3}$ and $1$, the negative root from composition must check $p(x) = \frac{8}{3}$, $1$ mappings symmetrically, which depends on polynomial behavior. We solve $p(x) = 1$, and $p(x) = \frac{8}{3}$:

Firstly, solve $p(x) = 1$:
\[ -\frac{3}{2}x^2 + \frac{13}{2}x - 4 = 1 \]
\[ -\frac{3}{2}x^2 + \frac{13}{2}x - 5 = 0 \]
\[ -3x^2 + 13x - 10 = 0 \]
\[ 3x^2 - 13x + 10 = 0. \]

Using the quadratic formula $x = \frac{-b \pm \sqrt{b^2 - 4ac}}{2a}$ with $a = 3$, $b = -13$, and $c = 10$:
\[ x = \frac{13 \pm \sqrt{169 - 120}}{6} \]
\[ x = \frac{13 \pm \sqrt{49}}{6} \]
\[ x = \frac{13 \pm 7}{6} \]

```
\[ x = \frac{20}{6} = \frac{10}{3}, \text{ and } \]
\[ x = \frac{6}{6} = 1. \]

So the solutions for $p(x) = 1$ are $x = \frac{10}{3}$, and $x = 1$.

Next, solve $p(x) = \frac{8}{3}$:
\[ -\frac{3}{2} x^2 + \frac{13}{2} x - 4 = \frac{8}{3} \]
Multiply everything by 6 to clear the denominators:
\[ -9x^2 + 39x - 24 = 16 \]
\[ -9x^2 + 39x - 40 = 0 \]
\[ 9x^2 - 39x + 40 = 0. \]

We use the quadratic formula again, with $a = 9$, $b = -39$, and $c = 40$:
\[ x = \frac{39 \pm \sqrt{39^2 - 4 \cdot 9 \cdot 40}}{2 \cdot 9} \]
\[ x = \frac{39 \pm \sqrt{1521 - 1440}}{18} \]
\[ x = \frac{39 \pm \sqrt{81}}{18} \]
\[ x = \frac{39 \pm 9}{18} \]
Thus,
\[ x = \frac{48}{18} = \frac{24}{9} = \frac{8}{3}, \]
and
\[ x = \frac{30}{18} = \frac{15}{9} = \frac{5}{3}. \]

Thus, the four roots in total are $1$, $\frac{8}{3}$, $\frac{10}{3}$, $\frac{5}{3}$. Since we
need the only non-integer solution, it is $\frac{10}{3}$, $\frac{8}{3}$, and $\frac{5}{3}$.
Since $\frac{8}{3}$ is already a fixed point solution, the non-integer solutions check to $\
frac{10}{3}$ and $\frac{5}{3}$. The only non-integer solution which is not repeated for fixed
point through other direct substitutions naturally is $\frac{5}{3}$.

Check gcd form: The reduced fraction $\frac{5}{3}$ in simplest form with $m=5$, $n=3$. So $m+n
 = 5+3 = 8$.

Therefore, the answer is $\boxed{8}$.
```

Listing 1: An illustration of original problem solving in SvS training.

### K.2.2 CASE STUDY FOR VARIATIONAL PROBLEM SYNTHESIS

```
### Input Correct Response

(Please refer to Response 1 in Listing 1)

### Response 1 (Acc: 0.5, Reward: 1.0):

A polynomial \( p(x) \) is given by \( p(x) = ax^2 + bx + c \) such that it satisfies the
conditions:
\[ p(1) = 1, \quad p(2) = 3, \quad p(3) = 2. \]

First, find the polynomial \( p(x) = ax^2 + bx + c \) that meets these conditions.
Then, determine the roots of the equation \( p(p(x)) = x \) and find the sum \( m+n \) for the
 non-integer root.

### Response 2 (Acc: 0.625, Reward: 1.0):

A polynomial $p(x) = ax^2 + bx + c$ satisfies the conditions $p(1) = 1$, $p(2) = 3$, and $p(3)
 = 2$. Given this polynomial $p(x)$, find the non-integer root of the equation $p(p(x)) = x$.
Finally, if this non-integer root is of the form $\frac{m}{n}$ where $m$ and $n$ are coprime
integers, determine the sum $m + n$.

### Response 3 (Acc: 0.75, Reward: 0.0):

Consider a polynomial $p(x) = ax^2 + bx + c$ that satisfies the conditions $p(1) = 1$, $p(2) =
 3$, and $p(3) = 2$. Determine the coefficients of the polynomial and find its explicit form.
Next, solve the equation $p(p(x)) = x$, given that it has four real roots and that three of
the roots are integers. Provide the sum of the numerator and denominator for the non-integer
root of the equation.

### Response 4 (Acc: 0.375, Reward: 1.0):

A polynomial $p(x) = ax^2 + bx + c$ is such that it satisfies the equations $p(1) = 1$, $p(2)
= 3$, and $p(3) = 2$. Determine the polynomial $p(x)$. Then solve for the real roots of the
equation $p(p(x)) = x$. Given that this equation has four real roots, with three of them being
 integers, find the fractional root in the form $\frac{m}{n}$ and compute the sum $m+n$.

### Response 5 (Acc: 0.875, Reward: 0.0):

Given the polynomial \( p(x) = ax^2 + bx + c \) that satisfies the conditions \( p(1) = 1, \ p
(2) = 3, \) and \( p(3) = 2 \), determine the roots of the equation \( p(p(x)) = x \), knowing
 that there are four real roots and three of them are integers. Calculate \( m + n \) where \(
 \frac{m}{n} \) is the non-integer root in its simplest form.

### Response 6 (Acc: 0.75, Reward: 0.0):

A polynomial \( p(x) = ax^2 + bx + c \) satisfies the following conditions: \( p(1) = 1 \), \(
 p(2) = 3 \), and \( p(3) = 2 \). Determine the polynomial \( p(x) \). Additionally, solve for
 the roots of the equation \( p(p(x)) = x \). Among the roots, three are integers and one is a
 rational number in the form of a fraction \(\frac{m}{n}\); find the value of \( m + n \).
```

```
### Response 7 (Acc: 0.875, Reward: 0.0):

A polynomial \( p(x) = ax^2 + bx + c \) satisfies the following conditions:
\[ p(1) = 1, \quad p(2) = 3, \quad p(3) = 2. \]

Determine the specific form of the polynomial \( p(x) \).

Additionally, find all the roots of the equation \( p(p(x)) = x \), given that this equation
has four real roots, of which three are integers. Calculate and provide the sum \( m + n \)
for the non-integer root of the form \( \frac{m}{n} \), where \( m \) and \( n \) are coprime
positive integers.

### Response 8 (Acc: 0.75, Reward: 0.0):

A polynomial \( p(x) = ax^2 + bx + c \) satisfies the conditions \( p(1) = 1 \), \( p(2) = 3
\), and \( p(3) = 2 \). Determine the polynomial \( p(x) \). Then, solve for the roots of the
equation \( p(p(x)) = x \) given that this equation has four real roots, of which three are
integers. Find the value of \( \frac{m}{n} \) for the non-integer root, and compute \( m+n \).
```

Listing 2: An illustration of variational problem synthesis in SvS training.

### K.2.3 CASE STUDY FOR SYNTHETIC PROBLEM SOLVING

```
### Input Synthetic Problem (Response 1 in Listing 2)

A polynomial \( p(x) \) is given by \( p(x) = ax^2 + bx + c \) such that it satisfies the
conditions:
\[ p(1) = 1, \quad p(2) = 3, \quad p(3) = 2. \]

First, find the polynomial \( p(x) = ax^2 + bx + c \) that meets these conditions.
Then, determine the roots of the equation \( p(p(x)) = x \) and find the sum \( m+n \) for the
 non-integer root.

### Response 1 (Reward: 0.0)

Given the quadratic polynomial \( p(x) = ax^2 + bx + c \), we have three conditions:
\[ p(1) = 1, \quad p(2) = 3, \quad p(3) = 2. \]

Let's express these conditions in terms of \(a\), \(b\), and \(c\):

1. \( p(1) = ay^2 + by + c \bigg|_{y=1} = a + b + c = 1 \),
2. \( p(2) = ay^2 + by + c \bigg|_{y=2} = 4a + 2b + c = 3 \),
3. \( p(3) = ay^2 + by + c \bigg|_{y=3} = 9a + 3b + c = 2 \).

We now have the system of equations:
\[
\begin{cases}
a + b + c = 1, & \quad \text{(1)} \\
4a + 2b + c = 3, & \quad \text{(2)} \\
9a + 3b + c = 2. & \quad \text{(3)}
\end{cases}
\]

Let's solve these equations step by step. First, we eliminate \(c\) from these equations. We
subtract equation (1) from equation (2):
\[ (4a + 2b + c) - (a + b + c) = 3 - 1 \]
\[ 3a + b = 2 \quad \text{(4)} \]

Next, subtract equation (2) from equation (3):
\[ (9a + 3b + c) - (4a + 2b + c) = 2 - 3 \]
\[ 5a + b = -1 \quad \text{(5)} \]

Now, we have another system of equations:
\[
\begin{cases}
3a + b = 2, & \quad \text{(4)} \\
5a + b = -1. & \quad \text{(5)}
\end{cases}
\]

Subtract equation (4) from equation (5):
\[ (5a + b) - (3a + b) = -1 - 2 \]
\[ 2a = -3 \]
\[ a = -\frac{3}{2} \]

Next, substitute \( a = -\frac{3}{2} \) back into equation (4):
\[ 3\left(-\frac{3}{2}\right) + b = 2 \]
\[ -\frac{9}{2} + b = 2 \]
\[ b = 2 + \frac{9}{2} \]
\[ b = \frac{4}{2} + \frac{9}{2} \]
\[ b = \frac{13}{2} \]

Finally, substitute \( a = -\frac{3}{2} \) and \( b = \frac{13}{2} \) back into equation (1):
\[ -\frac{3}{2} + \frac{13}{2} + c = 1 \]
\[ \frac{-3 + 13}{2} + c = 1 \]
```

```
\[ 5 + c = 1 \]
\[ c = 1 - 5 \]
\[ c = -4 \]
```

Thus, the polynomial is
```
\[ p(x) = -\frac{3}{2} x^2 + \frac{13}{2} x - 4 \]
```

The general form of \( p(x) \) is:
```
\[ p(x) = -\frac{3}{2} x^2 + \frac{13}{2} x - 4 \]
```
or equivalently,
```
\[ p(x) = \frac{{-3x^2 + 13x - 8}}{2} \]
```

Next, we need to find the roots of the equation \( p(p(x)) = x \). We start by expressing \( p(p(x)) \):
First, let's compute \( p(y) \), where \( y = p(x) = \frac{-3x^2 + 13x - 8}2 \).

Let \( y = p(x) \). Then
```
\[
p(y) = p \left( \frac{-3x^2 + 13x - 8}{2} \right) = -\frac{3}{2} \left( \frac{-3x^2 + 13x -
8}{2} \right)^2 + \frac{13}{2} \left(\frac{-3x^2 + 13x - 8}{2}\right) - 4
\]
```
We'll start by expanding \(\left( \frac{-3x^2 + 13x - 8}{2} \right)^2 \):
```
\[
\left( \frac{-3x^2 + 13x - 8}{2} \right)^2 = \frac{ (-3x^2 + 13x - 8)^2}{4}
\]
```
Expanding \( (-3x^2 + 13x - 8)^2 \):
```
\[
(-3x^2 + 13x - 8)^2 = (-3x^2 + 13x - 8)(-3x^2 + 13x - 8) = 9x^4 - 2 \cdot 13 \cdot 3 x^3 +
(169x^2 + 2 \cdot 3 \cdot 8 x^2 - 64) - 2 \cdot 8 \cdot 13 x + 64
= 9x^4 - 78x^3 + (169 + 48) x^2 - 208x + 64 = 9x^4 - 78x^3 + 217x^2 - 208x + 64
\]
```
Thus,
```
\[
\left(\frac{-3x^2 + 13x - 8}{2}\right)^2 = \frac{9x^4 - 78x^3 + 217x^2 - 208x + 64}{4}
\]
```
Now, we have
```
\[
-\frac{3}{2} \left(\frac{-3x^2 + 13x - 8}{2}\right)^2 = -\frac{3}{2} \cdot \frac{9x^4 - 78x^3
+ 217x^2 - 208x + 64}{4} = -\frac{3}{8} (9x^4 - 78x^3 + 217x^2 - 208x + 64)
= -\frac{27x^4}{8} + \frac{234x^3}{8} - \frac{651x^2}{8} + \frac{624x}{8} - \frac{192}{8}
= -\frac{27}{8} x^4 + \frac{117}{4} x^3 - \frac{651}{8} x^2 + \frac{78}{1} x - 24
= -\frac{27}{8} x^4 + \frac{117}{4} x^3 - \frac{651}{8} x^2 + 78x - 24
\]
```
Then
```
\[
\frac{13}{2} \left( \frac{-3x^2 + 13x - 8}{2} \right) = \frac{13}{4} (-3x^2 + 13x - 8) = \frac
{-39x^2 + 169x - 104}{4} = -\frac{39}{4} x^2 + \frac{169}{4} x - 26
\]
```
Thus,
```
\[
p \left( p(x) \right) = -\frac{27}{8} x^4 + \frac{117}{4} x^3 - \frac{651}{8} x^2 + 78 x - 24
- \frac{39}{4} x^2 + \frac{169}{4} x - 26 - 4 = -\frac{27}{8} x^4 + \frac{117}{4} x^3 - \left(
\frac{651}{8} + \frac{39 \cdot 2}{8} \right) x^2 + \left( 78 + \frac{169}{4} \right) x - 54
= -\frac{27}{8} x^4 + \frac{117}{4} x^3 - \frac{729}{8} x^2 + \left( \frac{312}{4} + \frac
{169}{4} \right) x - 54
= -\frac{27}{8} x^4 + \frac{117}{4} x^3 - \frac{729}{8} x^2 + \frac{481}{4} x - 54
\]
```
So, we set \( p(p(x)) = x \):
```
\[
-\frac{27}{8} x^4 + \frac{117}{4} x^3 - \frac{729}{8} x^2 + \frac{481}{4} x - 54 = x
\]
\[
-\frac{27}{8} x^4 + \frac{117}{4} x^3 - \frac{729}{8} x^2 + \left( \frac{481}{4} - 1 \right) x
- 54 = 0
= -\frac{27}{8} x^4 + \frac{117}{4} x^3 - \frac{729}{8} x^2 + \left( \frac{481}{4} - \frac
{4}{4} \right) x - 54 = 0
= -\frac{27}{8} x^4 + \frac{117}{4} x^3 - \frac{729}{8} x^2 + \frac{477}{4} x - 54 = 0
\]
```
This is a quartic equation
```
\[
-27 x^4 + 234 x^3 - 729 x^2 + 954 x - 432 = 0
\]
```
We can simplify by dividing through by \(-3\):
```
\[
9 x^4 - 78 x^3 + 243 x^2 - 318 x + 144 = 0
\]
```

Next, we must find the roots of the quartic equation \( 9x^4 - 78x^3 + 243x^2 - 318x + 144 = 0 \). The roots include any solutions to the original polynomial equations such as \( p(x) = x \). Solving \( p(x) = x \):
```
\[
-\frac{3}{2} x^2 + \frac{13}{2} x - 4 = x
\]
```
Multiply through by 2:
```
\[
-3x^2 + 13x - 8 = 2x
\]
\[
-3x^2 + 11x - 8 = 0
```

\]
We solve this quadratic equation using the quadratic formula $( x = \frac{-b \pm \sqrt{b^2 - 4ac}}{2a} )$, where $( a = -3 )$, $( b = 11 )$, and $( c = -8 )$.
\[
x = \frac{-11 \pm \sqrt{11^2 - 4(-3)(-8)}}{2(-3)} = \frac{-11 \pm \sqrt{121 - 96}}{-6} = \frac{-11 \pm \sqrt{25}}{-6} = \frac{-11 \pm 5}{-6}
\]
So,
\[
x = \frac{-11 + 5}{-6} = \frac{-6}{-6} = 1
\]
\[
x = \frac{-11 - 5}{-6} = \frac{-16}{-6} = \frac{8}{3}
\]
Thus, the solutions to $( p(x) = x )$ are $( x = 1 )$ and $( x = \frac{8}{3} )$.

Next, we need to consider the other roots that might not satisfy $( p(x) = x )$. Let's denote other roots by $( r )$, if $( p(p(x)) = x )$ where $( p(x) \neq x )$. Assume $( p(x) = y )$, so $( p(y) = x )$. Hence, we have $( y = a x^2 + b x + c )$. Therefore
\[
p(p(x)) = p(y) = x.
\]
We know that the roots already include $( x = 1 )$ and $( x = \frac{8}{3} )$, so we need to check if there are any further roots. We substitute $( p(x) = y )$, then $( p(y) = x )$.

A quartic must have four roots total. We need the roots of the polynomial $( 9x^4 - 78x^3 + 243x^2 - 318x + 144 = 0 )$. We already have two roots: $( x=1 )$ and $( x=\frac{8}{3} )$.

Using the polynomial division, we can factor out $((x-1)\left( x-\frac{8}{3} \right))$.

Let's compute the product $((x-1)\left(x - \frac{8}{3}\right) = x^2 - \left(1+\frac{8}{3}\right)x + \frac{8}{3} = x^2 - \frac{11}{3} x + \frac{8}{3})$. We now divide $(9x^4 - 78x^3 + 243x^2 - 318x + 144)$ by $(x^2 - \frac{11}{3}x + \frac{8}{3})$.

First, write $((x^2 - \frac{11}{3}x + \frac{8}{3}) = \frac{1}{3}(3x^2 - 11x + 8))$.

So we need to perform polynomial division $(\frac{9x^4 - 78x^3 + 243x^2 - 318x + 144}{\frac{1}{3}(3x^2 - 11x + 8)})$:
\[
= \frac{9x^4 - 78x^3 + 243x^2 - 318x + 144}{(3x^2 - 11x + 8)} \quad \text{(by cancelling out constant factor)}
\]

We perform long division:
1. Divide $(9x^4)$ by $(3x^2)$ and get $(3x^2)$.
\[
9x^4 - 78x^3 + 243x^2 - 318x + 144 - (3x^2(3x^2 - 11x + 8)) = 9x^4 - 78x^3 + 243x^2 - 318x + 144 - (9x^4 - 33x^3 + 24x^2)
\]
\[
= -45x^3 + 219x^2 - 318x + 144
\]
2. Divide $(-45x^3)$ by $(3x^2)$ and get $(-15x)$:
\[
-45x^3 + 219x^2 - 318x + 144 - ((-15x)(3x^2 - 11x + 8)) = -45x^3 + 219x^2 - 318x + 144 - (-45x^3 + 165x^2 - 120x)
\]
\[
= 54x^2 - 198x + 144
\]
3. Divide $(54x^2)$ by $(3x^2)$ and get 18:
\[
54x^2 - 198x + 144 - (18(3x^2 - 11x + 8)) = 54x^2 - 198x + 144 - (54x^2 - 198x + 144) = 0
\]

Therefore
\[
9x^4 - 78x^3 + 243x^2 - 318x + 144 = (3x^2 - 11x + 8)(3x^2 - 13x + 18)
\]

Next, solve $(3x^2 - 13x + 18 = 0)$:
\[
x = \frac{13 \pm \sqrt{169 - 4 \cdot 3 \cdot 18}}{6} = \frac{13 \pm \sqrt{169 - 216}}{6} = \frac{13 \pm \sqrt{-47}}{6} = \frac{13 \pm i \sqrt{47}}{6}
\]

Thus, the roots are nonreal $( \frac{13 + i \sqrt{47}}{6} )$ and $( \frac{13 - i \sqrt{47}}{6} )$.

We also have $(3x^2 - 11x + 8 = 0)$ (for which we previously had the roots $( x=1 )$ and $( x =\frac{8}{3} )$).

Finally, we need the sum $( m+n )$ for the non-integer root:
Since the non-integer roots are $(\frac{8}{3})$, $(\frac{13 + i\sqrt{47}}{6})$, and $(\frac{13 - i\sqrt{47}}{6})$:

From the roots, $(\frac{8}{3})$ is not complex and another non-integer root's real part leads us the next likely "noninteger" candidate. For the complex root $(\frac{13 + i \sqrt{47}}{6})$, only consider real part $( \frac{13}{6} )$ which is rational non-integer.

Therefore $\frac{13}{6}$ leads to $m=13$ and $n=6$. Then, $m+n = 19$.

The answer is therefore $\boxed{19}$.

### Response 2 (Reward: 1.0)

Given the polynomial $p(x) = ax^2 + bx + c$ that meets the conditions $p(1) = 1$, $p(2) = 3$, and $p(3) = 2$, we start by forming a system of equations based on those conditions:
$$p(1) = a(1)^2 + b(1) + c = a + b + c = 1$$
$$p(2) = a(2)^2 + b(2) + c = 4a + 2b + c = 3$$
$$p(3) = a(3)^2 + b(3) + c = 9a + 3b + c = 2$$
We need to solve the following system of linear equations:
1. $a + b + c = 1 \tag{1}$
2. $4a + 2b + c = 3 \tag{2}$
3. $9a + 3b + c = 2 \tag{3}$
To eliminate $c$, we subtract equation (1) from equations (2) and (3):
$$(4a + 2b + c) - (a + b + c) = 3 - 1$$
$$3a + b = 2 \tag{4}$$
Similarly,
$$(9a + 3b + c) - (a + b + c) = 2 - 1$$
$$8a + 2b = 1 \tag{5}$$
We now solve the system formed by equations (4) and (5):
$$3a + b = 2 \tag{4}$$
$$8a + 2b = 1 \tag{5}$$
Multiply equation (4) by 2 to align coefficients of $b$ with equation (5):
$$6a + 2b = 4 \tag{6}$$
Subtract equation (5) from equation (6):
$$(6a + 2b) - (8a + 2b) = 4 - 1$$
$$-2a = 3$$
So,
$$a = -\frac{3}{2}$$
Now substitute $a = -\frac{3}{2}$ into equation (4):
$$3 \left(-\frac{3}{2} \right) + b = 2$$
$$-\frac{9}{2} + b = 2$$
$$b = 2 + \frac{9}{2} = \frac{4}{2} + \frac{9}{2} = \frac{13}{2}$$
Now that we have $a = -\frac{3}{2}$ and $b = \frac{13}{2}$, substitute these values into equation (1):
$$-\frac{3}{2} + \frac{13}{2} + c = 1$$
$$c = 1 - \left(\frac{13}{2} - \frac{3}{2}\right) = 1 - \left(\frac{10}{2}\right) = 1 - 5 = -4$$
We found $a = -\frac{3}{2}$, $b = \frac{13}{2}$, and $c = -4$. Then,
$$p(x) = -\frac{3}{2} x^2 + \frac{13}{2} x - 4$$
To get a cleaner polynomial, multiply through by 2:
$$p(x) = -\frac{3}{2} x^2 + \frac{13}{2} x - 4 = \frac{1}{2} (-3x^2 + 13x - 8)$$
So we can rewrite original polynomial as
$$p(x) = \frac{-3x^2 + 13x - 8}{2}$$

Next, we need to solve $p(p(x)) = x$. First, express $p(p(x))$:
Let $y = p(x) = \frac{-3x^2 + 13x - 8}{2}$. Then, compute $p(y)$:
$$p(y) = p \left( \frac{-3x^2 + 13x - 8}{2} \right) = \frac{-3 \left( \frac{-3x^2 + 13x - 8}{2} \right)^2 + 13 \left( \frac{-3x^2 + 13x - 8}{2} \right) - 8}{2}$$
First, simplify $\left(\frac{-3x^2 + 13x - 8}{2}\right)^2$:
$$\left(\frac{-3x^2 + 13x - 8}{2}\right)^2 = \frac{(-3x^2 + 13x - 8)^2}{4}$$
Next, expand $(-3x^2 + 13x - 8)^2$:
$$(-3x^2 + 13x - 8)^2 = (-3x^2 + 13x - 8)(-3x^2 + 13x - 8)$$
$$
\begin{aligned}
&= 9x^4 - 39x^3 + 24x^2 - 39x^3 + 169x^2 - 104x + 24x^2 - 104x + 64 \\
&= 9x^4 - 78x^3 + 217x^2 - 208x + 64
\end{aligned}
$$
So,
$$\left(\frac{-3x^2 + 13x - 8}{2}\right)^2 = \frac{9x^4 - 78x^3 + 217x^2 - 208x + 64}{4}$$
Then compute $p(p(x))$,
$$p(p(x)) = p \left( y \right) = p \left( \frac{-3x^2 + 13x - 8}{2} \right)$$
$$p \left( y \right) = \frac{-3 \left( \frac{9x^4-78x^3+217x^2-208x+64}{4} \right) + 13 \left( \frac{-3x^2 + 13x - 8}{2} \right) - 8}{2}$$
We now simplify expressions inside:
$$-3 \left( \frac{9x^4 - 78x^3 + 217x^2 - 208x + 64}{4} \right) = \frac{-27x^4 + 234x^3 - 651x^2 + 624x - 192}{4}$$
$$13 \left( \frac{-3x^2 + 13x - 8}{2} \right) = \frac{13(-3x^2 + 13x - 8)}{2} = \frac{-39x^2 + 169x - 104}{2}$$
Then combine everything:
$$p(p(x)) = \frac{\frac{-27x^4 + 234x^3 - 651x^2 + 624x - 192}{4} + \frac{-39x^2 + 169x - 104}{2} - 8}{2}$$
$$= \frac{\frac{-27x^4 + 234x^3 - 651x^2 + 624x - 192}{4} + \frac{2(-39x^2+169x-104)}{4} - \frac{32}{4}}{2}$$
$$= \frac{\frac{-27x^4 + 234x^3 - 651x^2 + 624x - 192 -78x^2 + 338x - 208 - 32}{4}}{2}$$
$$= \frac{\frac{-27x^4 + 234x^3 - 729x^2 + 962x - 432}{4}}{2}$$
$$= \frac{-27x^4 + 234x^3 - 729x^2 + 962x - 432}{8}$$
So,
$$p(p(x)) = \frac{-27x^4 + 234x^3 - 729x^2 + 962x - 432}{8}$$
We need $p(p(x)) = x$,
$$\frac{-27x^4 + 234x^3 - 729x^2 + 962x - 432}{8} = x$$
Multiplying both sides by 8,
$$-27x^4 + 234x^3 - 729x^2 + 962x - 432 = 8x$$
$$-27x^4 + 234x^3 - 729x^2 + 954x - 432 = 0$$
Dividing by -3 all terms,
$$9x^4 - 78x^3 + 243x^2 - 318x + 144 = 0$$

Thus we solved a quartic equation:
$$ 9x^4 - 78x^3 + 243x^2 - 318x + 144 = 0 $$
To find the roots, we use the Rational Root Theorem to test possible rational roots, which are $\pm 1, \pm 2, \pm 3, \pm 4, \pm 6, \pm 8, \pm 9, \pm 12, \pm 16, \pm 18, \pm 24, \pm 27, \pm 36, \pm 48, \pm 54, \pm 72, \pm 108, \pm 144, \frac{\pm1}{3}, \frac{\pm1}{9}, \frac{\pm2}{3}, \frac{\pm 4}{3}, \frac{\pm8}{3}, \frac{\pm2}{9}, \frac{\pm4}{9}, \frac{\pm8}{9}, \frac{\pm16}{9}$.

Testing $x = 1$:
$$ 9(1)^4 - 78(1)^3 + 243(1)^2 - 318(1) + 144 = 9 - 78 + 243 - 318 + 144 = 0 $$
So, $x = 1$ is a root. Using synthetic division to divide $9x^4 - 78x^3 + 243x^2 - 318x + 144$ by $x - 1$,
$$ \begin{array}{r|rrrrr}
1 & 9 & -78 & 243 & -318 & 144 \\
  &   & 9  & -69 & 174 & -144 \\
\hline
  & 9 & -69 & 174 & -144 & 0 \\
\end{array} $$
Resulting quotient is:
$$ 9x^3 - 69x^2 + 174x - 144 $$

Now, test $x = 2$:
$$ 9(2)^3 - 69(2)^2 + 174(2) - 144 = 9(8) - 69(4) + 348 - 144 $$
$$ = 72 - 276 + 348 - 144 $$
$$ = 72 + 348 - 276 - 144 $$
$$ = 420 - 420 = 0 $$
So, $x = 2$ is a factor.

Using synthetic division again to divide $9x^3 - 69x^2 + 174x - 144$ by $x-2$,
$$ \begin{array}{r|rrrr}
2 & 9 & -69 & 174 & -144 \\
  &   & 18  & -102 & 144 \\
\hline
  & 9 & -51 & 72 & 0 \\
\end{array} $$
Resulting quotient is:
$$ 9x^2 - 51x + 72 $$
So the original polynomial can be factored as:
$$ (x-1)(x-2)(9x^2 - 51x+72) = 0 $$
For solving the quadratic $9x^2 - 51x + 72 = 0$:
$$ 3(3x^2 - 17x + 24) = 0 \Rightarrow 3x^2 - 17x + 24 = 0 $$
Using the quadratic formula for $3x^2 - 17x + 24 = 0$,
$$ x = \frac{-b \pm \sqrt{b^2 - 4ac}}{2a} = \frac{17 \pm \sqrt{17^2 - 4 \cdot 3 \cdot 24}}{2 \cdot 3} = \frac{17 \pm \sqrt{289 - 288}}{6} = \frac{17 \pm \sqrt{1}}{6} = \frac{17 \pm 1}{6} $$
$$ x = \frac{18}{6} = 3, \quad x = \frac{16}{6} = \frac{8}{3} $$
Thus, the roots are:
$$ x = 1, \quad x = 2, \quad x = 3, \quad x = \frac{8}{3} $$
The non-integer root is $\frac{8}{3}$, and the sum $m + n$ where $\frac{m}{n} = \frac{8}{3}$ with $\gcd(m, n) = 1$ is $8 + 3 = 11$.

Thus, the sum of the non-integer root components is:
$$ \boxed{11} $$

### Response 3 (Reward: 1.0)

Given the polynomial $p(x) = ax^2 + bx + c$ that meets the conditions $p(1) = 1$, $p(2) = 3$, and $p(3) = 2$, we start by forming a system of equations based on those conditions:
$$ p(1) = a(1)^2 + b(1) + c = a + b + c = 1 $$
$$ p(2) = a(2)^2 + b(2) + c = 4a + 2b + c = 3 $$
$$ p(3) = a(3)^2 + b(3) + c = 9a + 3b + c = 2 $$
We need to solve the following system of linear equations:
1. $a + b + c = 1 \tag{1}$
2. $4a + 2b + c = 3 \tag{2}$
3. $9a + 3b + c = 2 \tag{3}$
To eliminate $c$, we subtract equation (1) from equations (2) and (3):
$$ (4a + 2b + c) - (a + b + c) = 3 - 1 $$
$$ 3a + b = 2 \tag{4} $$
Similarly,
$$ (9a + 3b + c) - (a + b + c) = 2 - 1 $$
$$ 8a + 2b = 1 \tag{5} $$
We now solve the system formed by equations (4) and (5):
$$ 3a + b = 2 \tag{4} $$
$$ 8a + 2b = 1 \tag{5} $$
Multiply equation (4) by 2 to align coefficients of $b$ with equation (5):
$$ 6a + 2b = 4 \tag{6} $$
Subtract equation (5) from equation (6):
$$ (6a + 2b) - (8a + 2b) = 4 - 1 $$
$$ -2a = 3 $$
So,
$$ a = -\frac{3}{2} $$
Now substitute $a = -\frac{3}{2}$ into equation (4):
$$ 3 \left(-\frac{3}{2} \right) + b = 2 $$
$$ -\frac{9}{2} + b = 2 $$
$$ b = 2 + \frac{9}{2} = \frac{4}{2} + \frac{9}{2} = \frac{13}{2} $$

Now that we have $a = -\frac{3}{2}$ and $b = \frac{13}{2}$, substitute these values into equation (1):
$$ -\frac{3}{2} + \frac{13}{2} + c = 1 $$
$$ c = 1 - \left(\frac{13}{2} - \frac{3}{2}\right) = 1 - \left(\frac{10}{2}\right) = 1 - 5 = -4 $$
We found $a = -\frac{3}{2}$, $b = \frac{13}{2}$, and $c = -4$. Then,
$$ p(x) = -\frac{3}{2} x^2 + \frac{13}{2} x - 4 $$
To get a cleaner polynomial, multiply through by 2:
$$ p(x) = -\frac{3}{2} x^2 + \frac{13}{2} x - 4 = \frac{1}{2} (-3x^2 + 13x - 8) $$
So we can rewrite original polynomial as
$$ p(x) = \frac{-3x^2 + 13x - 8}{2} $$

Next, we need to solve $p(p(x)) = x$. First, express $p(p(x))$:
Let $y = p(x) = \frac{-3x^2 + 13x - 8}{2}$. Then, compute $p(y)$:
$$ p(y) = p \left( \frac{-3x^2 + 13x - 8}{2} \right) = \frac{-3 \left( \frac{-3x^2 + 13x - 8}{2} \right)^2 + 13 \left( \frac{-3x^2 + 13x - 8}{2} \right) - 8}{2} $$
First, simplify $\left(\frac{-3x^2 + 13x - 8}{2}\right)^2$:
$$ \left(\frac{-3x^2 + 13x - 8}{2}\right)^2 = \frac{(-3x^2 + 13x - 8)^2}{4} $$
Next, expand $((-3x^2 + 13x - 8)^2)$:
$$ (-3x^2 + 13x - 8)^2 = (-3x^2 + 13x - 8)(-3x^2 + 13x - 8) $$
$$
= 9x^4 - 39x^3 + 24x^2 - 39x^3 + 169x^2 - 104x + 24x^2 - 104x + 64
= 9x^4 - 78x^3 + 217x^2 - 208x + 64
$$
So,
$$ \left(\frac{-3x^2 + 13x - 8}{2}\right)^2 = \frac{9x^4 - 78x^3 + 217x^2 - 208x + 64}{4} $$
Then compute $p(p(x))$,
$$ p(p(x)) = p \left( y \right) = p \left( \frac{-3x^2 + 13x - 8}{2} \right) $$
$$ p \left( y \right) = \frac{-3 \left( \frac{9x^4-78x^3+217x^2-208x+64}{4} \right) + 13 \left( \frac{-3x^2 + 13x - 8}{2} \right) - 8}{2} $$
We now simplify expressions inside:
$$ -3 \left( \frac{9x^4 - 78x^3 + 217x^2 - 208x + 64}{4} \right) = \frac{-27x^4 + 234x^3 - 651x^2 + 624x - 192}{4} $$
$$ 13 \left( \frac{-3x^2 + 13x - 8}{2} \right) = \frac{13(-3x^2 + 13x - 8)}{2} = \frac{-39x^2 + 169x - 104}{2} $$
Then combine everything:
$$ p(p(x)) = \frac{\frac{-27x^4 + 234x^3 - 651x^2 + 624x - 192}{4} + \frac{-39x^2 + 169x - 104}{2} - 8}{2} $$
$$ = \frac{\frac{-27x^4 + 234x^3 - 651x^2 + 624x - 192}{4} + \frac{2(-39x^2+169x-104)}{4} - \frac{32}{4}}{2} $$
$$ = \frac{\frac{-27x^4 + 234x^3 - 651x^2 + 624x - 192 -78x^2 + 338x - 208 - 32}{4}}{2} $$
$$ = \frac{\frac{-27x^4 + 234x^3 - 729x^2 + 962x - 432}{4}}{2} $$
$$ = \frac{-27x^4 + 234x^3 - 729x^2 + 962x - 432}{8} $$
So,
$$ p(p(x)) = \frac{-27x^4 + 234x^3 - 729x^2 + 962x - 432}{8} $$
We need $p(p(x)) = x$,
$$ \frac{-27x^4 + 234x^3 - 729x^2 + 962x - 432}{8} = x $$
Multiplying both sides by 8,
$$ -27x^4 + 234x^3 - 729x^2 + 962x - 432 = 8x $$
$$ -27x^4 + 234x^3 - 729x^2 + 954x - 432 = 0 $$
Dividing by -3 all terms,
$$ 9x^4 - 78x^3 + 243x^2 - 318x + 144 = 0 $$
Thus we solved a quartic equation:
$$ 9x^4 - 78x^3 + 243x^2 - 318x + 144 = 0 $$
To find the roots, we use the Rational Root Theorem to test possible rational roots, which are
$( \pm 1, \pm 2, \pm 3, \pm 4, \pm 6, \pm 8, \pm 9, \pm 12, \pm 16, \pm 18, \pm 24, \pm 27, \pm 36, \pm 48, \pm 54, \pm 72, \pm 108, \pm 144, \frac{\pm1}{3}, \frac{\pm1}{9}, \frac{\pm2}{3}, \frac{\pm 4}{3}, \frac{\pm8}{3}, \frac{\pm2}{9}, \frac{\pm4}{9}, \frac{\pm8}{9}, \frac{\pm16}{9} )$.

Testing $x = 1$:
$$ 9(1)^4 - 78(1)^3 + 243(1)^2 - 318(1) + 144 = 9 - 78 + 243 - 318 + 144 = 0 $$
So, $x = 1$ is a root. Using synthetic division to divide $9x^4 - 78x^3 + 243x^2 - 318x + 144$ by $x - 1$,
$$ \begin{array}{r|rrrrr}
1 & 9 & -78 & 243 & -318 & 144 \\
  &   & 9   & -69 & 174  & -144 \\
\hline
  & 9 & -69 & 174 & -144 & 0 \\
\end{array} $$
Resulting quotient is:
$$ 9x^3 - 69x^2 + 174x - 144 $$

Now, test $x = 2$:
$$ 9(2)^3 - 69(2)^2 + 174(2) - 144 = 9(8) - 69(4) + 348 - 144 $$
$$ = 72 - 276 + 348 - 144 $$
$$ = 72 + 348 - 276 - 144 $$
$$ = 420 - 420 = 0 $$
So, $x = 2$ is a factor.

Using synthetic division again to divide $9x^3 - 69x^2 + 174x - 144$ by $x-2$,
$$ \begin{array}{r|rrrr}
2 & 9 & -69 & 174 & -144 \\
  &   & 18  & -102 & 144 \\
\hline
  & 9 & -51 & 72 & 0 \\
\end{array} $$
Resulting quotient is:
$$ 9x^2 - 51x + 72 $$

So the original polynomial can be factored as:
$$ (x-1)(x-2)(9x^2 - 51x+72) = 0 $$
For solving the quadratic $( 9x^2 - 51x + 72 = 0 )$:
$$ 3(3x^2 - 17x + 24) = 0 \Rightarrow 3x^2 -17x + 24 = 0 $$
Using the quadratic formula for $( 3x^2 - 17x + 24 = 0 )$,
$$ x = \frac{-b \pm \sqrt{b^2 - 4ac}}{2a} = \frac{17 \pm \sqrt{17^2 - 4 \cdot 3 \cdot 24}}{2 \cdot 3} = \frac{17 \pm \sqrt{289 - 288}}{6} = \frac{17 \pm \sqrt{1}}{6} = \frac{17 \pm 1}{6} $$
$$ x = \frac{18}{6} = 3, \quad x = \frac{16}{6} = \frac{8}{3} $$
Thus, the roots are:
$$ x = 1, \quad x = 2, \quad x = 3, \quad x = \frac{8}{3} $$
The non-integer root is $( \frac{8}{3} )$, and the sum $( m + n )$ where $( \frac{m}{n} = \frac{8}{3} )$ with $( \gcd(m, n) = 1 )$ is $( 8 + 3 = 11 )$.

Thus, the sum of the non-integer root components is:
$$
\boxed{11}
$$

### Response 4 (Reward: 0.0)

To find the polynomial $( p(x) = ax^2 + bx + c )$ that satisfies $( p(1) = 1 )$, $( p(2) = 3 )$, and $( p(3) = 2 )$, we start by writing out the equations these conditions give us:
$$
p(1) = a(1)^2 + b(1) + c = a + b + c = 1,
$$
$$
p(2) = a(2)^2 + b(2) + c = 4a + 2b + c = 3,
$$
$$
p(3) = a(3)^2 + b(3) + c = 9a + 3b + c = 2.
$$
These equations form the following system:
$$
\begin{cases}
a + b + c = 1, & \text{(1)}\\
4a + 2b + c = 3, & \text{(2)}\\
9a + 3b + c = 2. & \text{(3)}
\end{cases}
$$

We begin by eliminating $( c )$. Subtract (1) from (2):
$$
(4a + 2b + c) - (a + b + c) = 3 - 1 \Rightarrow 3a + b = 2. \quad \text{(4)}
$$

Next, subtract (2) from (3):
$$
(9a + 3b + c) - (4a + 2b + c) = 2 - 3 \Rightarrow 5a + b = -1. \quad \text{(5)}
$$

We now solve the system of linear equations (4) and (5):
$$
\begin{cases}
3a + b = 2, & \text{(4)}\\
5a + b = -1. & \text{(5)}
\end{cases}
$$
Subtract equation (4) from equation (5):
$$
(5a + b) - (3a + b) = -1 - 2 \Rightarrow 2a = -3 \Rightarrow a = -\frac{3}{2}.
$$
Substituting $( a = -\frac{3}{2} )$ back into equation (4):
$$
3\left(-\frac{3}{2}\right) + b = 2 \Rightarrow -\frac{9}{2} + b = 2 \Rightarrow b = 2 + \frac{9}{2} = \frac{4}{2} + \frac{9}{2} = \frac{13}{2}.
$$
Next, substitute $( a = -\frac{3}{2} )$ and $( b = \frac{13}{2} )$ into equation (1):
$$
-\frac{3}{2} + \frac{13}{2} + c = 1 \Rightarrow \frac{-3 + 13}{2} + c = 1 \Rightarrow \frac{10}{2} + c = 1 \Rightarrow 5 + c = 1 \Rightarrow c = 1 - 5 = -4.
$$
Therefore, the polynomial is:
$$
p(x) = -\frac{3}{2} x^2 + \frac{13}{2} x - 4.
$$

Next, we need to find the roots of equation $( p(p(x)) = x )$. First, note:
$$
p(x) = -\frac{3}{2}x^2 + \frac{13}{2}x - 4.
$$
We want $( p(p(x)) = x )$. Suppose $( y = p(x) )$. Then $( p(p(x)) = p(y) = x )$. So we need:
$$
p(y) = x \Rightarrow -\frac{3}{2} y^2 + \frac{13}{2} y - 4 = x.
$$
Now replace $( y = p(x) )$:
$$
-\frac{3}{2}\left( -\frac{3}{2} x^2 + \frac{13}{2} x - 4 \right)^2 + \frac{13}{2}\left( -\frac{3}{2} x^2 + \frac{13}{2} x - 4 \right) - 4 = x.
$$

\]
Let \( p(x) = w \). Thus \( w = -\frac{3}{2} x^2 + \frac{13}{2} x - 4 \) and \( p(w) = x \Rightarrow -\frac{3}{2} w^2 + \frac{13}{2} w - 4 = x \).
Thus, we need:
\[
-\frac{3}{2} \left(-\frac{3}{2} x^2 + \frac{13}{2} x - 4 \right)^2 + \frac{13}{2} \left(-\frac{3}{2} x^2 + \frac{13}{2} x - 4 \right) - 4 = x.
\]

First, consider some simpler cases by checking if the roots of \( p(x) - x = 0 \) solve \( p(p(x)) = x \):
\[
p(x) = x \Rightarrow -\frac{3}{2} x^2 + \frac{13}{2} x - 4 = x,
\]
\[
-\frac{3}{2} x^2 + \frac{13}{2} x - 4 - x = 0 \Rightarrow -\frac{3}{2} x^2 + \frac{11}{2} x - 4 = 0.
\]
Multiply by \(-2\):
\[
3 x^2 - 11 x + 8 = 0.
\]
Solve this quadratic equation using the quadratic formula \( x = \frac{-b \pm \sqrt{b^2-4ac}}{2a} \):
\[
x = \frac{11 \pm \sqrt{121 - 96}}{6} = \frac{11 \pm \sqrt{25}}{6} = \frac{11 \pm 5}{6}.
\]
So the solutions are:
\[
x = \frac{16}{6} = \frac{8}{3} \quad \text{and} \quad x = \frac{6}{6} = 1.
\]
Thus, \(x=1\) and \(x=\frac{8}{3}\) are solutions. Now, we need to check whether there are any other solutions by considering the polynomial form of \( p(p(x)) = x \).

Let's compute \( p(p(x)) \) step-by-step:
\[
p(x) = -\frac{3}{2}x^2 + \frac{13}{2} x - 4.
\]
Let \( y = p(x) \). Then
\[
p(y) = -\frac{3}{2} y^2 + \frac{13}{2} y - 4,
\]
and substituting \( y = -\frac{3}{2}x^2 + \frac{13}{2}x - 4 \):
\[
p\left(-\frac{3}{2}x^2 + \frac{13}{2}x - 4 \right) = -\frac{3}{2} \left( -\frac{3}{2}x^2 + \frac{13}{2} x - 4 \right)^2 + \frac{13}{2} \left( -\frac{3}{2} x^2 + \frac{13}{2} x - 4 \right) - 4.
\]
Let's denote \( p(x) = -\frac{3}{2}x^2 + \frac{13}{2}x - 4 \) as:
\[
y = -\frac{3}{2} x^2 + \frac{13}{2} x - 4.
\]
Now plug this into \( p(y) \):
\[
p(y) = -\frac{3}{2} \left( -\frac{3}{2}x^2 + \frac{13}{2} x - 4 \right)^2 + \frac{13}{2} \left( -\frac{3}{2} x^2 + \frac{13}{2} x - 4 \right) - 4.
\]

The expression for \( \left( -\frac{3}{2}x^2 + \frac{13}{2} x - 4 \right)^2 \):
\[
\left( -\frac{3}{2} x^2 + \frac{13}{2} x - 4 \right)^2 = \left( \frac{-3x^2 + 13x - 8}{2} \right)^2 = \frac{(-3x^2 + 13x - 8)^2}{4} = \frac{9x^4 - 78x^3 + 233x^2 - 208x + 64}{4}.
\]
Thus, we get:
\[
-\frac{3}{2} \times \frac{9x^4 - 78x^3 + 233x^2 - 208x + 64}{4} = -\frac{3}{8}(9x^4 - 78x^3 + 233x^2 - 208x + 64) = -\frac{27}{8}x^4 + \frac{234}{8}x^3 - \frac{699}{8}x^2 + \frac{624}{8}x - \frac{192}{8}.
\]
\[
-\frac{27}{8} x^4 + \frac{117}{4} x^3 - \frac{699}{8} x^2 + \frac{78}{1} x - 24.
\]

And for the term \(\frac{13}{2} y\):
\[
\frac{13}{2} y = \frac{13}{2} \left( -\frac{3}{2}x^2 + \frac{13}{2}x - 4 \right) = \frac{13}{2} \left( -\frac{3x^2}{2} + \frac{13x}{2} - 4 \right) = -\frac{39}{4} x^2 + \frac{169}{4} x - 26.
\]

Summarizing everything,
\[
p(p(x)) = -\frac{27}{8} x^4 + \frac{117}{4} x^3 - \frac{699}{8} x^2 + \frac{78}{1} x - 24 -\frac{39}{4} x^2 + \frac{169}{4} x - 26 - 4.
\]
Simplifying the above,
\[
p(p(x)) = -\frac{27}{8} x^4 + \frac{117}{4} x^3 - \left( \frac{699}{8} + \frac{39 \times 2}{8} \right)x^2 + \left( 78 + \frac{169}{4} \right) x - 54.
\]

```
\]

Simplifying further,
\[
\frac{39 \times 2}{8} = \frac{78}{8} = \frac{39}{4},
\]
\[
-\frac{27}{8} x^4 + \frac{117}{4} x^3 - \left( \frac{699 + 78}{8} \right)x^2 + \left( \frac
{312 + 169}{4} \right) x - 54,
\]
\[
-\frac{27}{8} x^4 + \frac{117}{4} x^3 - \frac{777}{8} x^2 + \frac{481}{4}x - 54.
\]
Thus, we need:
\[
-\frac{27}{8} x^4 + \frac{117}{4}x^3 - \frac{777}{8} x^2 + \frac{481}{4} x - 54 = x.
\]
\[
-\frac{27}{8} x^4 + \frac{117}{4} x^3 - \frac{777}{8} x^2 + \frac{481-8}{4} x - 54 = 0,
\]
\[
-\frac{27}{8} x^4 + \frac{117}{4} x^3 - \frac{777}{8} x^2 + \frac{473}{4} x - 54 = 0.
\]
Multiply by 8 to clear denominators:
\[
-27x^4 + 2 \cdot 117 x^3 - 777x^2 +2 \cdot 473 x - 8 \cdot 54 = 0,
\]
\[
-27x^4 + 234x^3 - 777x^2 + 946x - 432 = 0.
\]
Already found roots \( x = 1 \) and \( x = \frac{8}{3} \) from \( p(x) = x \):
So, \( x = 1 \) and \( x = \frac{8}{3} \) satisfy this polynomial. Since we already determined
 its polynomial roots, we use polynomial division or root localization.
Other roots must be from:
\[
27y^4 - 234y^3 + 777y^2 - 946y + 432 =0.
\]
Finding remaining roots: Using a common numerical solver or factoring tools, we solve for
\[
27y^4 - 234y^3 + 777y^2 - 946 y + 432 = 0.
\]
We got roots \( \approx 0.65, 1, \frac{8}{3}, 4 \), verifying we recognize known rational
roots \( 1 \), \( \frac{8}{3} \).

Given non-integer root: Let \( x = \frac{2m}{n} = 0.65 = \frac{13}{20} \text{ near } x_0 = \
frac{13}{20}.\)

Showing exact: Continual polynomial reduction:
\[
27 \left( y-1 \right) \left( y-\frac{8}{3} \right) polynomial \left( y^{2} - ax + 4 \right)
restored reduction.
\]
Root \( y = \frac{13}{20} \approx x = \alpha \).

From known previews we get continuous polynomial \( O(1) \) solving for integer parts solving
overlap non-integer:
\[
\boxed{23}.
\]
```

Listing 3: An illustration of synthetic problem solving in SvS training.