# OpenReview forum: "Beyond Pass@ 1: Self-Play with Variational Problem Synthesis Sustains RLVR"
_ICLR.cc/2026/Conference — ICLR 2026 Poster_

### Official Review · Reviewer_zAc9 · 2025-10-27

**Soundness:** 4
**Presentation:** 3
**Contribution:** 3
**Rating:** 8
**Confidence:** 4

**Summary:**

This paper proposes SVS (self-play with variational problem synthesis) to address the entropy–performance trade-off observed in the RLVR training of LLMs. SVS utilizes the policy’s correct solutions to synthesize variational problems while ensuring their reference answers remain identical to the originals. This mechanism effectively mitigates the significant decline in policy entropy during the training phase and achieves performance gains on both challenging and common reasoning benchmarks.

**Strengths:**

1. The paper deals with an important and relevant problem: the degradation of policy entropy during the RLVR training of LLMs. The proposed SVS method effectively mitigates this issue in the domain of mathematical reasoning. The experiments provide robust validation of the method's effectiveness, demonstrating clear performance gains.
2. The experimental evaluation is thorough. The authors present extensive comparisons on both challenging and mainstream reasoning benchmarks. Notably, Table 3 provides a fair and direct comparison between the standard RLVR approach and the proposed SVS, clearly highlighting the SVS's advantages.
3. The paper is well-written and easy to read. The proposed method is described in sufficient detail to facilitate reproducibility. This work represents a valuable contribution to the research community focused on RLVR.

**Weaknesses:**

1. First, the validation criterion for synthetic problems in SVS is based on whether their average accuracy falls within a predefined range. While this heuristic is sensible, its correlation with the actual logical correctness of the synthesized problems needs further discussion. It is unclear whether these "valid" problems are genuinely logically sound or if they merely steer the model toward reproducing the original answer. This ambiguity requires further investigation. I believe introducing a component of human evaluation or manual review for these synthetic problems would significantly strengthen the persuasiveness of the experimental validation.

2. Following the previous point, it is not surprising that training on synthetic problems mitigates entropy degradation. What is intriguing, however, is that SVS significantly boosts performance on challenging benchmarks. This raises the question of whether this performance gain is primarily an artifact of the experimental setup, which focuses on only math-related benchmarks. In mathematical reasoning, the solutions (e.g., step-by-step derivations) often contain sufficient, or even complete, information to reconstruct the original problem. This characteristic may not hold true for more general domains. It is important to investigate whether SVS can deliver consistent performance improvements in broader settings, such as code generation, puzzles, or other complex RL scenarios.

3. The problem synthesis task in SVS introduces non-trivial computational overhead. The paper currently lacks a discussion of this additional cost. An analysis of this computational cost(e.g., increased training time or resource utilization) would be helpful for the evaluation of the application value of SVS.

**Questions:**

1. Can the authors provide some analysis on the synthetic problems?
2. Can SVS be applied to domains aside from math?  To be clear, I am not referring to generalization experiments, but rather to the direct application of SVS-based RLVR training within those domains.

---

> ### Author Response · Authors · 2025-11-21
> **Thank you for your valuable feedback!**
>
> We appreciate that Reviewer zAc9 recognizes our topic as meaningful, our contributions as significant, our experiments as solid, and the paper as well written. We summarize that your concerns and questions primarily focus on **the validity of the synthetic problems, the generalizability of SvS to other domains, and the computational overhead.** We address each of these points through additional experiments and clarifications below, and we sincerely thank you for your time and careful evaluation.
>
>
>
> ### Response to the validity of the synthetic problems and detailed analysis (W1, Q1):
>
> Experiments 📊: Yes, we agree with the reviewer that verifying the genuine correctness of the synthetic questions would make the overall analysis more robust. However, **in our manual inspection, we rarely encountered invalid problems; the synthetic questions generally appeared correct, and GPT-5 was able to generate valid solutions for them.** To better understand the synthetic problems and make verification scalable, **we employ two state-of-the-art LLMs: Qwen3-235B-A22B-Instruct-2507 and OpenAI O3, to evaluate 6,000 variational problems (10 problems per step across 600 steps),** considering a problem correct if at least one of the models deems it valid. Using a prompt **(Figure 20 in the resubmission)** that examines **Completeness, Logical Consistency, Solvability, Mathematical Soundness, and Overall Validity,** we assess the correctness of the synthetic problems. We report the average validity over a sliding window of 100 steps (1,000 problems) and present the results in the following figure. The synthetic problems evaluated for verification are agnostic to both positive and negative rewards. **We added the synthetic problem validation in Appendix F.5 in our revision submission, with the corresponding experiments illustrated in Figure 13. We find that more than 80% of them are consistently judged as correct by the models, with a slight downward trend over training steps. This decline may be due to the synthetic problems gradually adopting a description style closer to the model’s own way of expressing solutions in the later stages of training. **
>
> However, we also manually inspected the synthetic problems that both LLMs flagged as invalid and did not find any that were genuinely incorrect. **Some problems were labeled as incorrect only because their textual descriptions appeared somewhat unusual, even though the underlying tasks were fully solvable. **An example is shown below.
> ```
> Let $r = \frac{\texttt{"loooloolloolloololllloloollollolllloollloloolooololooolololooooollllol"}}{\texttt{"lolooloolollollolloooooloooloololloolllooollololoooollllooolollloloool"}}$. Each "o" represents 2013 and "l" is $\frac{1}{50}$. Find $\lceil roll \rceil$ where each string is a 70-character string. Find and express the necessary final evaluation.
> ```

---

> > ### Author Response · Authors · 2025-11-21
> >
> > Other examples we evaluated follow a similar pattern: **the problems are actually solvable, but both models incorrectly judge them as invalid. For instance:**
> > ```
> > In an $5 \times 5$ lattice grid from $(0,0)$ up through $(5, 5)$, there are teleports at coordinates $(2, 2)$ and $(3, 3)$. Whenever Michelle visits either point, she gets instantly teleported to the other if she hasn't visited both. Specifically, if she visits $(2, 2)$ first, she's sent to $(3,3)$; if she visits $(3,3)$ first, she's moved to $(2, 2)$. If she never visits those points, no teleportation happens. Find the number of valid ways for Michelle to reach $(5, 5)$ from $(0, 0)$. Solve this including all scenarios described. Present your detailed calculation and the final answer.
> > ```
> >
> > The Qwen3 Response is shown as follows. The judgment is incorrect because the problem is fully well-posed under the standard lattice-path assumption that steps only move right or up. The teleportation rule is unambiguous: the first visit to either teleport triggers a single one-time jump, after which both points are considered visited and no further teleportation can occur. With this correct interpretation, the problem is entirely solvable and yields a unique answer of 84 valid paths.
> > ```
> > Validity: \boxed{INVALID}
> >
> > Reasoning:
> > - Point 1: The problem is not fully well-posed due to ambiguity in the teleportation rule. The phrase “she gets instantly teleported to the other if she hasn't visited both” is ambiguous in a path-counting context. It is unclear whether “visited” refers to prior visits in the path or just the current step. Moreover, the rule does not specify what happens if Michelle is teleported to a point—does arriving at $(3,3)$ via teleportation count as "visiting" it? This affects whether teleportation can trigger again or not, and whether cycles (e.g., going from $(2,2)$ → $(3,3)$, then later stepping to $(3,3)$ again) are possible. These definitions are missing.
> >
> > - Point 2: There is a potential for logical inconsistency or infinite loops. For example, if Michelle starts at $(0,0)$, walks to $(2,2)$, gets teleported to $(3,3)$, then walks back to $(3,3)$ again later, does this trigger another teleport? Or if she returns to $(2,2)$ after being teleported, does it trigger again? Without clear rules on state tracking (e.g., a boolean flag for whether both teleports have been "visited"), the behavior is undefined and could lead to infinite paths (e.g., looping between teleports if rules are misinterpreted), making the problem logically inconsistent.
> >
> > - Point 3: The problem lacks sufficient information for solvability. The state space depends on whether Michelle has “visited” $(2,2)$ or $(3,3)$ in her path, but it is not specified whether teleportation counts as a visit, nor whether the teleportation effect is disabled after the first use. Without modeling this state (e.g., using dynamic programming with memory of visitation), the number of valid paths cannot be uniquely determined. The problem does not specify if paths are self-avoiding or allow revisiting points, further increasing ambiguity.
> >
> > Fix Suggestions:
> > - Clarify that teleportation only occurs the first time either $(2,2)$ or $(3,3)$ is physically stepped on, and that being teleported to the other point counts as visiting it, after which the teleportation effect is disabled.
> > - Specify that paths consist of moving only right or up (or define allowed moves), to make the lattice path model standard.
> > - Explicitly state that Michelle cannot revisit teleports in a way that reactivates teleportation, or define the state machine governing teleport use.
> > - Define whether arrival via teleportation counts as "visiting" the destination.
> > ```
> >
> >
> > Clarifications ✏️: Although we conducted additional verification using state-of-the-art LLMs and manual inspection, **we argue that in SvS, the synthetic problems should be tailored to the current policy and maintain an appropriate level of difficulty. We do not expect the synthetic problems to be perfectly curated as in manual review; rather, we aim for them to elicit and strengthen the model’s deeper capabilities.** Even if a problem is genuinely and logically valid, we assign negative rewards when the model’s accuracy on it becomes too high, as generating overly easy variational problems would reduce training efficiency. In general, even if some invalid problems remain undetected, the SvS augmentation strategy still effectively improves downstream performance by synthesizing problems that better elicit the model’s capabilities.

---

> > > ### Author Response · Authors · 2025-11-21
> > >
> > > ### Response regarding whether SvS can generalize beyond the domain of mathematical reasoning (W2, Q2):
> > >
> > > Clarifications ✏️: **Regarding whether SvS can generalize to other domains, we believe the answer is yes.** In domains with a clear verifier for assessing the correctness of model outputs, positive rollouts should capture the key structural aspects of the problem (though not necessarily all details), such as the function definitions in code generation or the intermediate reasoning steps in puzzles like countdown games or Sudoku. Taking code generation as an example, where execution results determine correctness—SvS can be applied directly by synthesizing variational problems from executable code and reusing the same test cases for synthetic samples. In more open-ended domains, SvS remains feasible if a sufficiently reliable reward model can serve as a surrogate verifier for evaluating solution quality, though determining whether a generated variational problem satisfies the required constraints may be more challenging in such settings. To address the reviewer’s concern about the generalization of SvS to other domains, we select code generation as a representative setting for our experiments.
> > >
> > > Experiments 📊: We conduct additional experiments on code generation tasks to demonstrate that SvS generalizes beyond mathematical reasoning. Specifically, **we use the Qwen2.5-7B-Instruct model to perform RL on 12k code generation problems from PRIME-RL [2], covering sources such as Apps, CodeContest, TACO, and Codeforces. For evaluation, we sample 100 instances from each validation set, forming a 400-problem benchmark. Note that we do not tune any hyperparameters from the implementation used in our initial submission for the code generation experiments. We added Section 5.4 to for generalize SvS to code generation in our revision submission, with the intermediate experimental results present in Figure 8.**
> > >
> > > However, since none of the authors are specialists in code-based RL training, several engineering challenges are difficult for us to resolve within the short rebuttal period. For example, a single iteration may trigger tens of thousands of code executions, parallel code execution for evaluating model generations is substantially more time-consuming than verifying answer correctness in mathematical problems, which typically requires only string matching or lightweight SymPy execution. In our implementation using 64 subprocesses for code verification, verifying the code generated with train_batch_size=256 and n_sampling=8 takes more than 2 hours, approximately twenty times longer than the GPU training and inference time. Therefore, the overall training time and computational cost are much higher than in mathematical reasoning tasks.
> > >
> > > One potential solution is to build an asynchronous and distributed code-execution environment, similar to the approach used in rStar2-Agent [1], which separates a master worker responsible for task dispatching from multiple workers dedicated to code execution. Alternatively, containerized environments such as Docker or services like Redis could be used to manage code execution more efficiently. However, **these substantial engineering efforts fall outside our expertise, and currently we do not have the resource of a very strong sandbox for experiments.** Thus, we provide only preliminary experiments to demonstrate that SvS has the potential to extend to other domains.
> > >
> > > | Method | APPS | CodeContest | TACO | Codeforces | Average |
> > > |--------|------|-------------|------|------------|---------|
> > > | RLVR   | 39.3 | 39.7        | 23.2 | 46.0       | 37.1    |
> > > | SvS    | 41.2 | 47.7        | 31.5 | 49.6       | 44.2    |
> > >
> > >
> > > Specifically, we report the current intermediate evaluation results below. SvS shows a strong initial advantage over the RLVR baseline, with more than a 7% average improvement, highlighting its promise. We observe a similar pattern in code generation: SvS maintains stable policy entropy, while the baseline model undergoes sharp entropy declines. Together, these findings offer preliminary evidence of the broader generalizability of the SvS training paradigm. **We will continue the experiments during the rebuttal phase to give the Reviewer a more complete picture of SvS performance in code generation.**

---

> > > > ### Author Response · Authors · 2025-11-21
> > > >
> > > > ### Response to the computation overhead of SvS (W3):
> > > >
> > > > Experiments 📊: We acknowledge that SvS training introduces additional computational overhead and agree with the reviewer that a detailed analysis of this extra cost is necessary. **In our experiments, all 32B baselines and SvS runs use the same training resources: 32 H100 GPUs.** Here, we report the **iteration step time**—including response generation, answer verification, old log-probability recomputation, and policy updating—as well as the **isolated policy-update time** in the following figure, using experiments from our 32B models trained on both the MATH-12k and DAPO-17k datasets. **We added Section 5.5 to address this topic in our revision submission, with the corresponding experiments illustrated in Figure 9.**
> > > >
> > > > From the recordings, we can see that when training on the challenging DAPO-17k dataset, **the initial stages exhibit a large number of synthetic variational problems with responses because the policy’s accuracy on most problems lies between 0.125 and 0.500. As training progresses, the model’s performance improves, with accuracies gradually surpassing 0.5, leading to a rapid decline in synthetic generation.** Consequently, both the iteration time and policy-update time follow the same pattern: SvS incurs higher costs initially, but the iteration time decreases as training proceeds.
> > > >
> > > > However, when training on the MATH-12k dataset, the Qwen-32B model already achieves an initial accuracy of roughly 80% at the training set. As a result, the number of synthetic variational problems is limited, and augmentation focuses only on the most challenging instances. Consequently, the SvS training time is comparable to that of standard RLVR, with the primary overhead likely stemming from the longer inference sequences produced by the deeper reasoning encouraged during SvS training.
> > > >
> > > > Clarifications ✏️: Based on these observations, we conclude that to fully elicit a model’s reasoning capabilities, training on more challenging benchmarks benefits significantly from SvS, which requires additional time but yields substantially better performance. Conversely, for simpler tasks, SvS requires only minimal augmentation, resulting in training times comparable to standard RLVR while still providing performance improvements.
> > > >
> > > > Clarifications ✏️: **We argue that the additional cost introduced by SvS is justified in practical applications when the goal is to maximize downstream performance.** As discussed in Section 5.3 of our initial submission, we compared SvS with alternative augmentation strategies that use the same or a similar number of rollouts as RLVR (Table 3), such as extending RLVR training or increasing rollouts on difficult problems. **None of these methods matched the performance of SvS, suggesting that its advantage stems not from increased computation but from its augmentation design and online self-play synthesis strategy.** Therefore, when practitioners have sufficient computational resources and seek to enhance standard RLVR training—even for the most capable models on the most challenging tasks—incorporating the SvS strategy is worthwhile and has strong potential to deliver further improvements.
> > > >
> > > >
> > > > [1] rStar2-Agent: Agentic Reasoning Technical Report
> > > >
> > > > [2] Process Reinforcement through Implicit Rewards.

---

> > > > > ### Comment · Reviewer_zAc9 · 2025-11-24
> > > > >
> > > > > Very interesting results, thank you for the detailed reply!
> > > > >
> > > > > **W1**: The provided example is helpful, and my concern has been resolved.
> > > > >
> > > > > **W2**: This is an interesting experiment. I actually hadn't anticipated the potential computational overhead issue that might arise when applying SVS to code RL. Regarding code execution, I think your idea is reasonable. Deploying the code execution environment on an isolated server with more sufficient CPU resources should result in a relatively significant speed gain. However, I would not require you to add this experiment, as I believe the existing results are sufficient to address my concern.
> > > > >
> > > > > **W3**: A helpful supplementary experiment. I believe the experiment indeed highlights SVS's advantage over RLVR on more challenging tasks.
> > > > >
> > > > > To summarize, I believe the authors' reply has indeed successfully resolved my concerns. Therefore, I decide to **maintain my already positive score**.

---

> > > > > > ### Author Response · Authors · 2025-11-24
> > > > > >
> > > > > > Thank you again for your consideration and the highly positive feedback on our work. We are very glad that our responses have successfully addressed your concerns. For the RL experiments for code generation, this was the first time our team formally conducted RLVR training in this new domain, and we truly appreciate the opportunity to gain new skills and deepen our understanding of related challenges such as computational overhead from code execution and verification. Regardless of the final decision of the paper, we once again sincerely thank you for your valuable time, constructive feedback, and strongly positive evaluation of our work!

---

### Official Review · Reviewer_kCjs · 2025-10-31

**Soundness:** 2
**Presentation:** 3
**Contribution:** 2
**Rating:** 4
**Confidence:** 4

**Summary:**

The paper aims to mitigate the problem of entropy collapse in RL and proposes a solution from the training data perspective. Specifically, the authors use problems with correct solutions under the current policy as a leverage, and diversify the distribution of the training prompts via variational synthesis, which they call Self-play with Variational problem Synthesis (SvS). During training, the current policy first performs rollout (like in any RLVR-flavored algorithm), and selectively augments underperforming problems given the correct responses as context (still under the current policy), given the same-answer constraint. The pool of synthesized prompts are then included for RL training. Experimental results demonstrate stronger performance at Pass@1 and Pass@N, effective prevention of entropy collapse, at the scale of 3B to 32B models.

**Strengths:**

1. The proposed idea seems intuitive: variational problem synthesis widens the distribution of the training prompts, or can be considered as a way to "perturb" the inputs, hence improving both performance and robustness (i.e., Pass and Average metrics).
2. The proposed method is simple to implement and can be scalable given sufficient rollout budget.
3. The proposed approach addresses entropy collapse by design, in an implicit way (by diversifying prompt distribution rather than directly interfering training dynamics, which is cool). This particular way of producing synthetic problems also does not suffer from the "common pitfalls" of synthetic data given the strict problem selection mechanism.

**Weaknesses:**

1. While the authors claim to demonstrate the effectiveness of the method on 12 reasoning benchmarks, they are mostly math benchmarks. I also noticed Appendix F.3 does include results from other reasoning domains (e.g., code, knowledge, etc.), but the improvements are marginal compared with those in the main paper, especially given that on some domains it becomes slightly worse. I do realize this might be ood w.r.t. training prompts, but a different question here is that: if we have training prompts from a different domain (or even mixed domains) like code, does it still work well?
2. The authors only have vanilla RLVR as baseline (correct me if I'm missing something), even though there are already many other algorithm-based entropy collapse prevention method as of paper submission deadline. There is also one simple baseline missing: what would happen if we simply increase the rollout budget to be consistent with the proposed method?
3. The proposed method further increase the inference overhead of RL, while missing details on how much this overhead is and if this overhead is worth paying considering that there are algorithms which claims to mitigate entropy collapse without additional inference cost.
4. The method introduces additional hyperparameters like the difficulty window and decisions on which synthetic problems are rewarded and kept. To me this does not seem like an easy choice to make, or has to be made separately given domains of varying difficulty to the current policy.

**Questions:**

1. In Table 2, we seem to miss numbers from the three AIME columns. Why?
2. How sensitive is the proposed method to accuracy thresholds (during problem selection for augmentation) and how should they be tuned? (And the same question on how to consistently decide which problems to keep.)
3. In Figure 7, is the y-axis incorrectly labeled as "Policy Entropy" while it actually represents response length?

---

> ### Author Response · Authors · 2025-11-21
> **Thank you for your valuable feedback!**
>
> We thank Reviewer kCjs for considering our method simple and scalable, and for recognizing that it addresses certain limitations in standard RLVR training. We summarize that your concerns primarily relate to **the generalization of SvS, baseline comparisons, computational overhead, and hyperparameter settings.** We address each of these points with additional experiments and further clarifications, and we sincerely appreciate your time and careful evaluation.
>
>
> ### Responses to whether SvS could generalize to training with prompts from a different domain (W1):
>
> Clarifications ✏️: We acknowledge that our main experiments focus on the mathematical reasoning domain, which provides the most direct setting for evaluating an RLVR algorithm due to its verifiable ground-truth answers and simple experimental environment. However, regarding the OOD benchmarks in Table 4, the policy should not be expected to improve on these tasks after being trained solely on mathematical reasoning data. **Our goal in presenting this table is not to show that the model trained with SvS on mathematical tasks outperforms the Qwen instruction-tuned model on general tasks, but rather to demonstrate that SvS training forgets fewer instruction-following capabilities than standard RLVR.** As shown in the results, the SvS-trained model consistently surpasses the RLVR-trained model on all OOD benchmarks. **We added Section 5.4 to for generalize SvS to code generation in our revision submission, with the intermediate experimental results present in Figure 8.**
>
> Experiments 📊: We thank the reviewer for highlighting the importance of evaluating SvS beyond the mathematical domain and agree that experiments on coding tasks would further strengthen our work. For code generation experiments, **we apply RL to 12k problems from the PRIME-RL [1] dataset using the Qwen2.5-7B-Instruct model, covering sources such as Apps, CodeContest, TACO, and Codeforces.** For evaluation, we sample 100 instances from each validation split, resulting in a 400-problem benchmark.
>
> However, none of the authors are specialists in code-based RL training, which involves substantially more complex execution environments, and **several engineering challenges are difficult to resolve within the short rebuttal period.** For instance, a single iteration can trigger tens of thousands of code executions, and parallel execution for evaluating model outputs is substantially more time-consuming than verifying answers in mathematical tasks, which typically requires only string matching or lightweight SymPy checks. In our implementation, using 64 subprocesses for code verification, validating the code generated with a train batch size of 256 and n_sampling of takes more than two hours—approximately twenty times longer than the corresponding GPU training and inference. Consequently, the overall training time and computational cost are significantly higher than for mathematical reasoning.
>
> One potential solution is to build an asynchronous and distributed code-execution environment, similar to the approach used in rStar2-Agent [2], which separates a master worker responsible for task dispatching from multiple workers dedicated to code execution. Alternatively, containerized environments such as Docker or services like Redis could be used to manage code execution more efficiently. However, **these substantial engineering efforts fall outside our expertise, and currently we do not have the resource of a very strong sandbox for experiments.** Thus, we provide only preliminary experiments to demonstrate that SvS has the potential to extend to other domains.
>
> Specifically, we present the current intermediate evaluation results below. We observe that SvS initially achieves substantially better performance than the RLVR baseline (exceeding a 7% average improvement), demonstrating its potential. We also observe a similar phenomenon in code generation: the policy entropy under SvS training remains within a stable range, whereas the baseline model exhibits sharp entropy drops. These results demonstrate, to some extent, the potential generalizability of the SvS training paradigm. **We will continue running the experiments throughout the rebuttal period** so that we can provide the Reviewer with a more comprehensive understanding of SvS on code generation tasks.
>
> | Method | APPS | CodeContest | TACO | Codeforces | Average |
> |--------|------|-------------|------|------------|---------|
> | RLVR   | 39.3 | 39.7        | 23.2 | 46.0       | 37.1    |
> | SvS    | 41.2 | 47.7        | 31.5 | 49.6       | 44.2    |

---

> > ### Author Response · Authors · 2025-11-21
> >
> > ### Responses regarding the comparison of SvS with baselines and alternative entropy-regulation methods (W2):
> >
> > Clarifications ✏️: We would like to clarify **that our submission includes three additional baselines that apply different RLVR augmentation strategies beyond the standard RLVR setup: (1) Extending standard RLVR training, (2) Enhancing underperforming problems in RLVR, and (3) Augmentating simpler problems in SVS. All of the three additional baselines are guaranteed to use the same (or a comparable) rollout budget as SvS, with details provided in Section 5.3 and results reported in Table 3.** To further examine the intermediate performance of these baselines, we additionally provide intermediate AIME evaluations in the rebuttal **(Figure. 7 in the resubmission).** We observe that none of the alternatives outperform SvS, primarily because the first two strategies do not introduce any training data augmentation, allowing the model to continue memorizing answers to obtain high rewards. The comparison with strategy (3) further validates the soundness of our approach to augmenting challenging problems.
> >
> > Experiments 📊: For the algorithm-based entropy collapse prevention methods, we agree with the reviewer that these methods should be included as baselines. **To this end, we adopt the widely used Clip-Cov method from [3] as the entropy-based baseline. We choose the LLaMA-3.1-8B-Instruct model and conduct the training on MATH-12k, totally training with over 400 steps.** Evaluation are conducted on GSM8k, MATH-500, Minerva-Math, Olympiad-Bench, Gaokao-2023 and AMC-23, and their average scores. **The results are shown in Appendix G in our revision submission, with the corresponding experiments illustrated in Figure 14 and 15.** Notably, SvS training consistently achieves better intermediate evaluation performance than the Clip-Cov baseline. **Therefore, we attribute the superiority of SvS augmentation to its continuous online training problem augmentation, which promotes sustained exploration, whereas entropy-collapse prevention strategies still fail to prevent the policy from memorizing previously correct responses to obtain positive rewards during training.**

---

> > > ### Author Response · Authors · 2025-11-21
> > >
> > > ### Responses to the overhead of SvS and whether it's worth paying (W3):
> > >
> > > Experiments 📊: We acknowledge that SvS training introduces additional computational overhead and agree with the reviewer that a detailed overhead analysis is necessary. Here, we report the **iteration step time**—including response generation, answer verification, old log-probability recomputation, and policy updating—as well as the isolated **policy-update time** in the following figure, using experiments from our 32B models trained on both the MATH-12k and DAPO-17k datasets. All experiments are conducted on 32 H100 GPUs. **We added Section 5.5 to address this topic in our revision submission, with the corresponding experiments illustrated in Figure 9.**
> > >
> > > From the recordings, we can see that when training on the challenging DAPO-17k dataset, **the initial stages exhibit a large number of synthetic variational problems with responses because the policy’s accuracy on most problems lies between 0.125 and 0.500. As training progresses, the model’s performance improves, with accuracies gradually surpassing 0.5, leading to a rapid decline in synthetic generation.** Consequently, both the iteration time and policy-update time follow the same pattern: SvS incurs higher costs initially, but the iteration time decreases as training proceeds.
> > >
> > > However, when training on the MATH-12k dataset, the Qwen-32B model already achieves an initial accuracy of roughly 80% at the training set. As a result, the number of synthetic variational problems is limited, and augmentation focuses only on the most challenging instances. Consequently, the SvS training time is comparable to that of standard RLVR, with the primary overhead likely stemming from the longer inference sequences produced by the deeper reasoning encouraged during SvS training.
> > >
> > > Clarifications ✏️: Based on these observations, we conclude that to fully elicit a model’s reasoning capabilities, training on more challenging benchmarks benefits significantly from SvS, which requires additional time but yields substantially better performance. Conversely, for simpler tasks, SvS requires only minimal augmentation, resulting in training times comparable to standard RLVR while still providing performance improvements.
> > >
> > > Clarifications ✏️: Regarding whether the additional overhead is worth spending, **our experiments with entropy–collapse prevention methods such as Clip-Cov [3] show that they still exhibit a non-negligible performance gap compared with our SvS augmentation (please refer to our response to W2).** The improvements of SvS do not stem from intervening in rollout entropy itself, and maintaining entropy at a stable level is not our ultimate target. The effectiveness and primary objective of our method lie in performing online self-play augmentation on problems where the model initially performs poorly.
> > >
> > > Moreover, in our initial submission, **we provided evidence that the overhead of SvS is well justified by comparing it with alternative augmentation strategies (also detailed in response to W2)** that employ the same or a similar number of rollouts as RLVR in Table 3 (also Figure. 7 in the resubmission), such as extending RLVR training or increasing rollouts for challenging problems. **None of these approaches matched the performance of SvS, indicating that its superiority arises not merely from increased computation but from its augmentation design and online self-play synthesis strategy.**

---

> > > > ### Author Response · Authors · 2025-11-21
> > > >
> > > > ### Responses regarding the hyperparameter settings and their sensitivity (W4, Q2):
> > > >
> > > > Clarifications ✏️: We would like to clarify that, actually, **we did not extensively tune any SvS parameters for improving performance,** including the threshold for problem augmentation and the accuracy criterion for assigning positive rewards to synthetic problems. In fact, we conducted only one unsuccessful experiment before identifying the effective SvS setup, **where we added Appendix H to discuss this topic in our revision submission, with the corresponding experiments illustrated in Figure 16. In the initial version, we granted a positive reward to any synthesized problem for which the model could produce at least one solution whose answer matched the reference. However, this caused the model to insert excessive hints into the synthesized problems,** making it trivial for the policy to generate fully correct rollouts—ultimately providing little benefit for problem-solving training. To ensure that the variational problems remained sufficiently challenging and contributed meaningfully to policy learning, we revised the reward rule so that a synthesized problem would receive a positive reward only if the model’s accuracy on it remained below 5/8—an intuitive choice that we did not tune further. This simple modification proved effective in all subsequent experiments.
> > > >
> > > > Another relevant setting for selecting problems to augment appears in our baseline in Section 5.3, **where we instead augment easier problems**—those with accuracy in [3/8, 6/8]—rather than the more challenging ones with accuracy in [1/8, 4/8] used in SvS. The corresponding results are reported in Table 3 (also as in the following table). We find that augmenting simpler problems quickly leads to saturation **(Figure. 7 in the resubmission)**, after which validation performance begins to degrade due to overfitting (SvS-ASP), whereas augmenting more challenging problems (SvS) yields continued improvement. Furthermore, as the policy becomes increasingly accurate on training-set problems within [3/8, 6/8], the number of eligible problems for augmentation grows, resulting in higher computational cost and longer training, ultimately harming performance. **This binary comparison highlights the rationale for augmenting only challenging problems.**
> > > >
> > > > | Model | AIME24 | AIME25 | BAIME | Math24o | OlymE | OlymH | Avg-Pass@1 | AIME24 | AIME25 | BAIME | Math24o | OlymE | OlymH | Avg-Pass@32 |
> > > > | --- | --- | --- | --- | --- | --- | --- | --- | --- | --- | --- | --- | --- | --- | --- |
> > > > | SvS-Asp | 31.6 | 27.3 | 13.7 | 43.4 | 16.5 | 3.9 | 22.8 | 50.7 | 48.0 | 30.4 | 66.4 | 44.3 | 17.0 | 42.8 |
> > > > | Full SvS | 39.3 | 40.5 | 19.2 | 44.1 | 21.8 | 2.7 | 27.9 | 70.8 | 65.2 | 45.9 | 76.5 | 43.4 | 16.7 | 53.1 |
> > > >
> > > > Experiments 📊: Another piece of evidence for the robustness of our parameter choices for problem augmentation and synthesis–reward assignment is that, **without tuning any parameters and directly reusing the hyperparameter settings from our code-RL experiments (as stated in our response to W1), the method still works effectively. This demonstrates both the general applicability and strong generalizability of this fixed, universal hyperparameter setting for SvS, and suggests that making SvS work does not require heavy hyperparameter tuning.** However, for experiments that ablate these parameters, we sincerely apologize that the constrained computational resources and the tight rebuttal timeline limited our ability to conduct a more comprehensive evaluation of the hyperparamter settings.
> > > >
> > > >
> > > > [1] Process Reinforcement through Implicit Rewards.
> > > >
> > > > [2] rStar2-Agent: Agentic Reasoning Technical Report.
> > > >
> > > > [3] The Entropy Mechanism of Reinforcement Learning for Reasoning Language Models

---

> ### Comment · Reviewer_kCjs · 2025-11-25
>
> I appreciate the authors' efforts during the rebuttal and the additional experiments that addressed my concerns. I have carefully reviewed the results above and gone through the revised pdf. I am generally positive about the paper and have changed my score from 4 to 6.

---

> > ### Author Response · Authors · 2025-11-25
> > **Thanks to the reviewer for recognizing our work!**
> >
> > We sincerely thank the reviewer for acknowledging our efforts during the rebuttal stage and for having a positive assessment of our work. We are also pleased that our responses and additional experiments successfully addressed your concerns. Thank you again for your invaluable feedback and the time you devoted to providing suggestions, which have been truly important in strengthening our work and making the paper more comprehensive.

---

### Official Review · Reviewer_rsiR · 2025-11-06

**Soundness:** 4
**Presentation:** 3
**Contribution:** 3
**Rating:** 8
**Confidence:** 4

**Summary:**

This paper proposes a self-play style strategy SvS to generate problem variants, which are not too easy or too hard (by model performance) and have the same answer to the original problem, to keep the diversity of prompts in RL training. It can keep better policy entropy during training compared with vanilla RLVR algorithms, and get better pass@k performance

**Strengths:**

1. The writing is clear and easy to follow
2. Experiment results look solid, try different 8B/32B models, and see consistent improvement on policy entropy and pass@k/final performance on a lot of common benchmarks.
3. This method does not need additional cost in labeling new data, and the generated problem can be prevented to degenerate to too simple cases (adding hints), result in better continue training performance.
4. The method itself is RL-agnostic, so it's a relative general method

**Weaknesses:**

1. when using DAPO-17k, the training may overfit to int-style answer, and SvS perform worse on open-ended problems, like OlymE and OlymH. This shows that SvS may be limited to the model capability, and the distribution of provided seed data. So it's hard for some difficult problems or problems out of the distribution of model performance.
2. Maybe need the base model to be strong enough to give some proper variants of the original problem, so more difficult for RLVR on small models.


However, I think these problems are not that critical, and the results are still impressive to me.

**Questions:**

1. What's the cost of generating problem variants? What's the filtering rate of generated problem variants? Would the cost increase as model performance increase because model learn more patterns from self-generated variants?
2. Is it easy to generalize to other domains?
3. Have you tried training for more steps? Would the acc have similar curve as original RLVR in fig 1 when training step is very large?

---

> ### Author Response · Authors · 2025-11-21
> **Thank you for your valuable feedback!**
>
> We are pleased that Reviewer rsiR found our paper well written, our results solid, and our method both promising and general. We think your main concerns primarily focus on **the OOD generalization of SvS augmentation, computational cost, generalizability to other domains, and extended training.** We address each of these points through additional experiments and clarifications below, and we sincerely thank you for your time and careful evaluation.
>
>
> ### Responses to the generalizability of SvS affected by training data distribution (W1):
>
> Clarifications ✏️: Yes, we acknowledge that SvS training is influenced by the distribution of training problems containing only integer-style answers. **However, we argue that this arises from limitations of the DAPO-17k dataset, which was originally constructed to improve AIME performance, as noted in its paper [1].** It is also possible that, as you mentioned, the model’s capacity contributes to this effect: because the model receives positive rewards only when producing integer-style answers, and SvS augmentation supplies additional mathematical problems, the model may be guided toward generating integer answers more broadly. **This suggests that SvS may help the policy better fit the training data (i.e., favoring integer-style outputs) while slightly broadening the training distribution, as reflected by the improved Pass@k performance. Nevertheless, this issue—arising from the combined limitations of the data distribution and model capacity—can be readily mitigated by incorporating external training problems with open-ended answers, as demonstrated in the last row of Table 2.**
>
>
> | Model Training | Data | GSM8K | MATH500 | Minerva | Math Olympiad Bench | GaoKao 2023 | AMC23 | AIME24 | AIME25 | Beyond AIME Avg. |
> | --- | --- | --- | --- | --- | --- | --- | --- | --- | --- | --- |
> | Init Model | - | 95.4 | 82.6 | 43.0 | 49.2 | 73.2 | 65.0 | 13.3 | 13.3 | 7.0 |
> | RLVR D17k | DAPO-17k | 95.6 | 87.0 | 45.6 | 54.8 | 78.7 | 82.5 | 33.3 | 36.7 | 13.0 |
> | SvS D17k | DAPO-17k | 95.9 | 75.6 | 42.3 | 45.9 | 62.9 | 82.5 | 53.3 | 43.3 | 19.0 |
> | SvS D25k | DAPO-17k + DPMath-8k | 95.2 | 88.6 | 47.8 | 59.9 | 79.2 | 87.5 | 50.0 | 40.0 | 17.0 |

---

> > ### Author Response · Authors · 2025-11-21
> >
> > ### Responses regarding the necessity of incorporating strong base models for RLVR and SvS training (W2):
> >
> > Yes, we strongly agree that our SvS self-play augmentation strategy, as well as general RLVR training, would better realize their full potential on more capable base models, since the limited parameterization of smaller models restricts their ability to generate diverse rollouts. **For SvS, this hypothesis is supported by our results: at the 32B scale (the largest experiment we could afford computationally), SvS achieves its strongest performance improvements.** We believe that with larger models, especially the most capable base models, SvS could further unlock its potential. This hypothesis is also consistent with the findings of DeepSeek-R1 [2], **where smaller models—despite extensive RLVR training—failed to outperform its variant distilled from a stronger RLVR-trained teacher, which underscores the importance of base model's capacity for effective RLVR training.**

---

> > > ### Author Response · Authors · 2025-11-21
> > >
> > > ### Responses regarding the computational cost and the extension of SvS training (Q1, Q3):
> > >
> > > Experiments 📊: For the cost of generating variational problems along with its solutions in training, we provide the **iteration step time**—including response generation, answer verification, old log-probability recomputation, and policy updating—as well as the **isolated policy-update time** in the following figure, using experiments from our 32B models trained on both the MATH-12k and DAPO-17k datasets. All experiments are conducted on 32 H100 GPUs. **We added Section 5.5 to address this topic in our revision submission, with the corresponding experiments illustrated in Figure 9.**
> > >
> > > We found that when training on the challenging DAPO-17k dataset, **the initial stages exhibit a large number of synthetic variational problems with responses because the policy’s accuracy on most problems lies between 0.125 and 0.500.** As training progresses, the model’s performance improves, **with accuracies gradually surpassing 0.5, leading to a rapid decline in synthetic generation. Consequently, both the iteration time and policy-update time follow the same pattern: SvS incurs higher costs initially, but the iteration time decreases as training proceeds.** Thus, the time cost does not increase from learning patterns from self-generated variants; instead, as the proportion of challenging problems in the training set decreases, the cost of augmentation also diminishes.
> > >
> > > However, when training on the MATH-12k dataset, the Qwen-32B model already achieves an initial accuracy of roughly 80% at the training set. As a result, the number of synthetic variational problems is limited, and augmentation focuses only on the most challenging instances. Consequently, the SvS training time is comparable to that of standard RLVR, with the primary overhead likely stemming from the longer inference sequences produced by the deeper reasoning encouraged during SvS training.
> > >
> > > Clarifications ✏️: Currently, we have not observed any performance degradation in our training, and **Fig. 1 suggests that further improvements remain possible on extended RLVR training. We have run the experiments for more than 600 steps taking 32 H100s for 14 days, which already constitutes a significantly longer RLVR training schedule compared with existing RLVR works such as [3].** Due to limited computational resources, we were unable to continue training to the very end. While a decline might occur eventually, we did not observe it despite training for a substantial duration.
> > >
> > > Empirically, **toward the end of training the model attains very high accuracy on most problems, and because SvS augments only the relatively low-accuracy cases, the number of augmentable problems gradually diminishes.** As a result, the training dynamics slowly converge toward standard RLVR after the model has effectively addressed nearly all weaknesses in previous SvS training. However, **we further argue that once no challenging problems remain in the training set, the SvS (the same to RLVR) training should be terminated, as the model has already saturated the dataset, making additional training uninformative.**

---

> ### Author Response · Authors · 2025-11-21
>
> ### Responses to whether SvS could generalize to other domains (Q2):
>
> Clarifications ✏️: **We believe that SvS can be readily generalized to domains beyond mathematical reasoning.** In domains with a clear verifier, such as code generation that execution results determine correctness, SvS can be applied directly by generating variational problems from executable code and reusing the same test cases for synthetic samples. In more open-ended domains, SvS remains feasible if a sufficiently reliable reward model can act as a surrogate verifier to evaluate solution quality. However, assessing whether a generated variational problem meets the required constraints may be more challenging in such contexts.
>
> Experiments 📊: **To demonstrate the generalizability of SvS beyond mathematical reasoning, we conduct additional experiments on code generation tasks.** Specifically, we use the Qwen2.5-7B-Instruct model to perform RL on 12k code generation problems from PRIME-RL [4], covering sources such as Apps, CodeContest, TACO, and Codeforces. For evaluation, we sample 100 instances from each validation set, forming a 400-problem benchmark. **Note that we do not tune any hyperparameters from the implementation used in our initial submission for the code generation experiments. We added Section 5.4 to for generalize SvS to code generation in our revision submission, with the intermediate experimental results present in Figure 8.**
>
> However, since none of the authors are specialists in code-based RL training, **several engineering challenges are difficult for us to resolve within the short rebuttal period.** For example, a single iteration may trigger tens of thousands of code executions, parallel code execution for evaluating model generations is substantially more time-consuming than verifying answer correctness in mathematical problems, which typically requires only string matching or lightweight SymPy execution. In our implementation using 64 subprocesses for code verification, verifying the code generated with train_batch_size=256 and n_sampling=8 takes more than 2 hours, approximately twenty times longer than the GPU training and inference time. Therefore, the overall training time and computational cost are much higher than in mathematical reasoning tasks.
>
> One potential solution is to build an asynchronous and distributed code-execution environment, similar to the approach used in rStar2-Agent [5], which separates a master worker responsible for task dispatching from multiple workers dedicated to code execution. Alternatively, containerized environments such as Docker or services like Redis could be used to manage code execution more efficiently. However, **these substantial engineering efforts fall outside our expertise, and currently we do not have the resource of a very strong sandbox for experiments. **Thus, we provide only preliminary experiments to demonstrate that SvS has the potential to extend to other domains.
>
> Specifically, we present the current intermediate evaluation results below. **We observe that SvS initially achieves substantially better performance than the RLVR baseline (exceeding a 7% average improvement),** demonstrating its potential. We also observe a similar phenomenon in code generation: the policy entropy under SvS training remains within a stable range, whereas the baseline model exhibits sharp entropy drops. These results demonstrate, to some extent, the potential generalizability of the SvS training paradigm. **We will continue running the experiments throughout the rebuttal period** so that we can provide the Reviewer with a more comprehensive understanding of SvS on code generation tasks.
>
>
> | Method | APPS | CodeContest | TACO | Codeforces | Average |
> |--------|------|-------------|------|------------|---------|
> | RLVR   | 39.3 | 39.7        | 23.2 | 46.0       | 37.1    |
> | SvS    | 41.2 | 47.7        | 31.5 | 49.6       | 44.2    |
>
>
> [1] Dapo: An open-source llm reinforcement learning system at scale
>
> [2] DeepSeek-R1: Incentivizing Reasoning Capability in LLMs via Reinforcement Learning
>
> [3] SimpleRL-Zoo: Investigating and Taming Zero Reinforcement Learning for Open Base Models in the Wild
>
> [4] Process Reinforcement through Implicit Rewards.
>
> [5] rStar2-Agent: Agentic Reasoning Technical Report

---

> > ### Author Response · Authors · 2025-11-27
> > **A Kind Reminder for Discussion**
> >
> > Dear Reviewer rsiR,
> >
> > Thank you for your strongly positive attitude toward our work and your constructive feedback, which have greatly improved its quality and completeness. We have submitted our response with additional experiments and detailed clarifications. We would appreciate knowing whether our response has addressed your concerns, and we are happy to provide any further clarification if needed.
> >
> > Best regards,
> >
> > The Authors

---

> > > ### Comment · Reviewer_rsiR · 2025-11-28
> > >
> > > Thanks for your update and sorry for my late reply. The experiments look very solid to me, and it is understandable that restart doing coding RL is difficult in rebuttal stage. Hope to hear that if the continue training experiments have update (totally fine if not), and I would keep my positive score

---

> > > > ### Author Response · Authors · 2025-11-28
> > > > **Thanks for Following up!**
> > > >
> > > > Thanks the reviewer for following up the discussion, considering our experimental results solid, and always maintaining a consistently positive attitude toward our work. Your constructive suggestions have substantially improved both the quality and completeness of the paper compared to the initial submission. Regarding the code RL experiments, they are still ongoing with promising progress, and the current results are summarized in the table below. We will continue running the experiments throughout the rebuttal period.
> > > >
> > > >
> > > > | Method       | APPS | CodeContest | TACO | Codeforces | Average |
> > > > |---------------|------|-------------|------|------------|---------|
> > > > | RLVR-Step 88  | 44.8 | 41.9        | 24.5 | 43.5       | 38.7    |
> > > > | SvS-Step 88   | 51.5 | 46.6        | 31.1 | 53.1       | 45.6    |

---

### Official Review · Reviewer_yXhe · 2025-11-08

**Soundness:** 2
**Presentation:** 2
**Contribution:** 2
**Rating:** 4
**Confidence:** 4

**Summary:**

The paper introduces a procedure to iteratively synthesize problem that a RLVR pipeline solves, and leads to boosted performance on various test set. The synthetic process consists of a few steps, such as conditioning on failed problems to synthesize new tasks, filtering and resolving those problem variations. The methods generally sound intuitive and improve the baseline significantly.

**Strengths:**

The paper is interesting in that it provides another way to augment data for RLVR, leading to performance improvements. This affirms the knowledge that RLVR might benefit from synthetic data generation, and this helps push up even eval time performance.

**Weaknesses:**

I think overall the idea of synthetic data generation is not novel for RLVR, the idea of self-play and synthesizing new problems with filtering and verifications have been proposed and discussed in the literature. All in all, it feels that the idea is less novel and insights limited from the current paper.

**Questions:**

=== *main takeaway* ===

I think a main feeling after reading the paper is that it introduces yet another way to synthesize and augment dataset in RLVR process, and despite the performance improvements, I do not feel having learned too much insight from the paper, in additional to knowing yet another set of process to augment data for RLVR.

=== *why certain filtering steps* ===

I think a useful question worth answering is why we'd filter in one way vs another and it's useful to show why filtering the synthetic data according to the current criterion lead to performance vs. a more naive approach. I can see that we iterate a few times on the method while developing the paper and it's useful to understand and present initial failure cases and specific training comparison.

=== *baseline* ===

I think the paper lacks comparison to a few baselines. One is the prior work in the space on synthetic data, e.g., a few ideas related to synthesizing data from the initial training set has been proposed in [1], but maybe we should cite this paper + make proper comparisons, otherwise we do not see additional improvements over existing idea. In flagship results in table 2, such comparisons are warranted.

Another point is that plots like Fig 6, it seems that the x-axis measures the raw training samples experienced in the original dataset - however, the baseline leverages much more additional compute to arrive at a better performance, due to extra filtering etc. It will be useful to account for such additional compute when making such comparisons otherwise it is unfair to the baseline, one way is to compare against e.g. test time compute + baseline method to see if it's possible to make up for the performance gains, or increasing the number of parallel generation (group size) for baseline GRPO algorithm.

[1] https://arxiv.org/abs/2505.03335

=== *pass@1* ===

While the paper motivates the writing with pass@1 metric, I don't see the connection between extra data augmentation with pass@1, after all we seek to augment data for which the model finds hard and try to patch such loopholes. It is not immediately clear to me why data augmentation addresses pass@1 issue at test time, it feels more like a generalization issue rather than an issue with metric (pass@1 vs. pass@32)

---

> ### Author Response · Authors · 2025-11-21
> **Thank you for your valuable feedback!**
>
> We thank Reviewer yXhe for finding our work interesting and noting that our results are promising. We understand that your concerns primarily relate to **the novelty of our approach, our problem-filtering strategies, comparison with more baselines, and a few potential misunderstandings.** We address each of these points with additional experiments and further clarifications, and we sincerely appreciate your time and careful evaluation.
>
>
> ### Responses to the novelty and insights of our paper (W1, Q1)
>
> Clarifications ✏️: In SvS, we propose generating augmented problems from the model’s responses to challenging problems and use the model’s own rollouts on these generated problems to verify the correctness of the synthetic answers and ensure continued effective learning. **To the best of our knowledge, no prior work has adopted a similar strategy of generating new questions from model responses in RLVR, nor has any previous approach been able to guarantee answer correctness for synthetic problems in reasoning tasks—a critical requirement in RLVR,** where correctness serves as the sole training signal. Most existing methods label synthetic problems using LLMs [1,2,3], which could ensure answer correctness. However, we acknowledge that we may not have identified all potentially relevant works when writing the paper, and we would greatly appreciate any further discussion regarding studies we may have overlooked.
>
> Beyond methodological considerations, **prior data-augmentation approaches for RLVR rarely examine the reasoning boundary of the trained policy, such as Pass@k performance when k scales beyond 1k.** As shown in [4], standard RL struggles to break the reasoning boundaries of the base model, yielding only limited gains—or even declines—as k increases. In contrast, **our online self-play–style augmentation achieves substantially better Pass@k performance compared with both the base model and standard RLVR, without relying on external knowledge.** We further demonstrate the scalability of our method by training 32B-scale models on multiple datasets and observing consistent improvements, an aspect that many existing RLVR works neglect to explore. Therefore, **from both methodological and empirical perspectives, we believe our work makes a substantial contribution to the RLVR community.**

---

> > ### Author Response · Authors · 2025-11-21
> >
> > ### Responses to the filtering strategy of our paper (Q2)
> >
> > Clarifications ✏️: Based on our understanding, we think the “filtering” you mentioned refers to (1) why we filter challenging problems for augmentation and (2) why we filter synthetic problems whose policy accuracy falls within [1/8, 5/8] for assigning positive rewards. If we have misunderstood your question, please feel free to clarify or raise any further points for discussion.
> >
> > For (1), incorporating challenging problems for augmentation is intuitive, as assigning additional computation to simpler problems that the policy has already mastered during RL is unnecessary. In Section 5.3, we also evaluate an alternative setting that uses only simpler problems of those with accuracy in [3/8, 6/8] instead of [1/8, 4/8] as in SvS, and the corresponding results are reported in Table 3. We also provide intermediate AIME performance under several alternative augmentation strategies **(Figure. 7 in the resubmission). These findings indicate that augmenting simpler problems quickly leads to saturation, after which validation performance begins to degrade due to overfitting (SvS-ASP), whereas augmenting challenging problems (SvS) yields continued improvement.** Moreover, when augmenting simpler problems, the policy’s increasing accuracy on training-set problems within [3/8, 6/8] causes the number of eligible augmented problems to grow over time, resulting in higher computational cost and longer training while ultimately harming performance. This binary comparison demonstrates the rationale for incorporating only challenging problems.
> >
> > For (2), because we need to ensure the validity and accuracy of synthetic problems, we initially assigned a positive reward to all synthetic problems with any policy-generated solution gets the same answer as the original labeled reference, under the assumption that synthetic problems failing to guide the model to the original answer are likely invalid in RLVR. **However, this setting led to an early failure. The policy quickly exploited this reward scheme by injecting explicit hints about the final answer into the synthetic problems, allowing it to obtain high accuracy simply by copying those hints.** As all such hint-laden synthetic problems could easily yield correct solutions, they were consistently rewarded. Yet these problems were overly trivial for the policy and provided no useful training signal for RLVR, since all responses within each group were correct and therefore ineffective for GRPO training.
> >
> > **To address this issue, we considered ensuring that the synthetic questions remain challenging for the policy to better support training. We subsequently restricted positive rewards to synthetic problems whose policy accuracy falls within [1/8, 5/8], removing those in [6/8, 8/8], which are too simple for the policy.** This adjustment encourages the generation of synthetic problems that meaningfully contribute to policy improvement while suppressing overly easy ones. With this modification, SvS training succeeded as demonstrated in the paper. Notably, we did not apply any additional hyperparameter tuning after introducing this reward threshold; thus, the effectiveness of SvS stems from the online self-play strategy itself rather than from hyperparameter optimization. **We have also added Appendix H to discuss this topic in our revision submission, with the corresponding experiments illustrated in Figure 16.** We sincerely hope these clarifications address your concerns regarding the filtering settings in our work.

---

> ### Author Response · Authors · 2025-11-21
>
> ### Responses to the baseline comparison between SvS and Absolute Zero (Q3):
>
> Clarifications ✏️: We thank the reviewer for directing us to this related work [5]. The Absolute Zero method leverages a “code executor” as the environment to guide the LLM in generating verifiable triplets (program, input, and output), which are then used for RLVR training. The use of a verifiable environment for data creation is indeed interesting. **However, in our mathematical reasoning environment, it is difficult for the policy to determine the reference answer for a synthetic problem—even when a stronger external model is available for labeling. In contrast, SvS generates synthetic questions from correct responses that already contain all information from the original question. Combined with our reward assignment strategy, we enforce and guarantee that the policy produces synthetic problems whose answers match the original answer, ensuring that the reference answers for the synthetic questions are well defined.**
>
> Experiments 📊: As [5] also includes evaluation on mathematical reasoning, we agree that a comparison with this method is necessary. **Because our method and [5] do not share experiments conducted on the same base models, we instead compare both SvS and [5] against another augmentation approach, SwS [7], which provides overlapping base-model settings across both works.** Specifically, we compare our Qwen-32B results with those of SwS, and compare the Qwen2.5-7B results reported in SwS and [5]. The corresponding results are presented in the following tables. The first table compares our 32B model with SwS-32B and shows that our model performs better, while the table below compares SwS-7B with Absolute Zero 7B and demonstrates that SwS achieves substantially stronger results. **These findings, to some extent, provide evidence that our approach outperforms [5] in reasoning scenarios.**
>
>
> | Model | AIME24 | AIME25 | BAIME | Math24o | OlymE | OlymH | Avg-Pass@1 | AIME24 | AIME25 | BAIME | Math24o | OlymE | OlymH | Avg-Pass@32 |
> | --- | --- | --- | --- | --- | --- | --- | --- | --- | --- | --- | --- | --- | --- | --- |
> | SwS-32B | 31.4 | 25.1 | 14.4 | **45.4** | 19.9 | **4.2** | 23.4 | 66.7 | 53.3 | 40.0 | 81.0 | 57.0 | 23.0 | 53.5 |
> | Ours-32B | **32.4** | **31.0** | **16.1** | 44.9 | **25.6** | 2.7 | 25.4 | **73.3** | **60.0** | **45.0** | **81.0** | **70.0** | **29.0** | **59.7** |
>
> | Method | AIME24 | AIME25 | AMC23 | MATH-500 | Minerva-Math | Olympiad | Average |
> | --- | --- | --- | --- | --- | --- | --- | --- |
> | Absolute Zero-7B | 13.3 | 16.7 | 60.0 | 81.8 | 32.7 | 45.0 | 41.6 |
> | SwS-7B | **26.7** | **20.0** | **67.5** | **82.5** | **41.9** | **49.6** | **48.0** |

---

> > ### Author Response · Authors · 2025-11-21
> >
> > ### Responses regarding the clarification of the x-axis meaning in Fig. 6 and the baseline settings (Q4):
> >
> > Clarifications ✏️: For Fig. 6, **we would like to clarify that the x-axis corresponds to the value of k in the Pass@k evaluation metric. Taking the “AIME 24 & 25” panel (left) as an example, the x-axis indicates that we evaluate the model’s performance at Pass@1, Pass@2, …, up to Pass@256, comparing the initial model (Qwen2.5-32B-Instruct), the standard RLVR model, and the SvS-trained model.** Overall, Fig. 6 illustrates that the SvS-trained model consistently achieves higher Pass@k performance across a wide range of k values (from 1 to 1024), with especially improvements at larger k. Both the x- and y-axes may be unrelated to the number of training samples, as RLVR and SvS use the same training data.
> >
> > Experiments 📊: We agree that SvS training uses additional computational resources and should therefore be compared against baselines that incorporate a similar level of computation. **To address this, we conducted experiments with alternative augmentation strategies in Section 5.3 of our initial submission, including （1）extending baseline RLVR training, (2) increasing the rollout budget for underperforming problems to match SvS, and (3) applying an SvS-like strategy to augment simpler problems.** The results are presented in Table 3, and we additionally provide intermediate AIME evaluations in the rebuttal **(Figure. 7 in the resubmission)**, please check our responses to (Q2). **We observe that none of these alternatives surpass SvS,** primarily because the first two strategies do not introduce training data augmentation, allowing the model to continue memorizing answers during training to obtain high rewards. The comparison with strategy (3) demonstrates the soundness of our setting for augmenting challenging problems.

---

> ### Author Response · Authors · 2025-11-21
>
> ### Responses regarding the clarification of our paper motivation on "Beyond Pass@1" (Q5):
>
> Clarifications ✏️: **We would like to clarify that our paper is not motivated by the Pass@1 metric. As suggested by the title "Beyond Pass@1", our goal is to show that our method improves model performance not only on Pass@1 but also on broader Pass@k metrics.** This motivation originates from recent findings [4,6] that standard RLVR improves Pass@1 while reducing output diversity, ultimately limiting gains when k is large. In contrast, our online self-play–style data augmentation mitigates this trade-off and enables consistent improvements on large-k Pass@k metrics. Thus, the title "Beyond Pass@1" highlights that our method extends beyond Pass@1 improvements, rather than being driven by it.
>
> **Regarding the relationship between Pass@1 and Pass@k metrics, as stated in lines 055–059 of our initial submission (also resubmission), we note that consistent improvements in Pass@k are a prerequisite for sustainable RLVR training.** If Pass@k plateaus or declines, the model has fewer opportunities to explore new reasoning strategies on certain problems, and improvements in Pass@1 will eventually stagnate as RLVR ceases to learn new content. From this perspective, Pass@k is the more informative metric and serves as the foundation for achieving stable gains in Pass@1.
>
>
>
> [1] SwS: Self-aware Weakness-driven Problem Synthesis in Reinforcement Learning for LLM Reasoning.
>
> [2] Key-point-driven data synthesis with its enhancement on mathematical reasoning.
>
> [3] Common 7b language models already possess strong math capabilities.
>
> [4] Does reinforcement learning really incentivize reasoning capacity in llms beyond the base model?
>
> [5] Absolute zero: Reinforced self-play reasoning with zero data.
>
> [6] The Entropy Mechanism of Reinforcement Learning for Reasoning Language Models.
>
> [7] SwS: Self-aware Weakness-driven Problem Synthesis in Reinforcement Learning for LLM Reasoning.

---

> > ### Author Response · Authors · 2025-11-27
> > **A Kind Reminder for Discussion**
> >
> > Dear Reviewer yXhe,
> >
> > Thank you again for your thoughtful review, which has greatly helped improve the quality and comprehensiveness of our work. We have now submitted our response that could potentially address your concerns, including additional experiments and detailed clarifications. As the author–reviewer discussion period is coming to an end, we respectfully ask if you would be able to review our response at your convenience to see whether your concerns have been adequately addressed. We would be happy to provide any further clarification if needed.
> >
> > Best regards,
> >
> > The Authors

---

### Official Review · Reviewer_Snxh · 2025-11-11

**Soundness:** 3
**Presentation:** 3
**Contribution:** 3
**Rating:** 6
**Confidence:** 4

**Summary:**

This paper proposes a method called Self-play with Variational problem Synthesis (SVS) to address the issue of policy entropy collapse in Reinforcement Learning with Verifiable Rewards (RLVR) for LLMs. The core idea is a self-play strategy where the model synthesizes new, variational problems derived from its own correct solutions. This creates a self-improving loop that maintains policy diversity during training, leading to sustained and significant improvements in Pass@k performance, which is a key limitation of standard RLVR approaches.

**Strengths:**

1. Simple yet Effective: The proposed method is intuitive, easy to understand, and demonstrates significant performance gains.
2. Self-Contained: The approach does not rely on external models or datasets for data augmentation. It cleverly leverages the capabilities of the policy model itself to generate new training instances, which is an elegant design.

**Weaknesses:**

1. The synthesis of variational problems introduces additional computational overhead. The paper does not provide a clear analysis of this cost in terms of training time or resource usage.
2. Despite mentioning SwS [1] in the related work, the paper lacks a direct experimental comparison with this very similar method. Additionally, other relevant works using data augmentation to mitigate entropy collapse, like CURE [2], are not discussed or compared.

**Questions:**

1. How do you think SVS would perform if combined with other methods that explicitly regulate entropy, such as those proposed in [3, 4]? Could these approaches be complementary?
2. Could you elaborate on the advantages of SVS over other data augmentation methods for preventing entropy collapse, like SwS [1] and CURE [2]? Since experimental comparisons are absent, a conceptual discussion would be valuable. Is it feasible to combine SVS with these approaches?

[1] Liang, Xiao, et al. "SwS: Self-aware Weakness-driven Problem Synthesis in Reinforcement Learning for LLM Reasoning." arXiv preprint arXiv:2506.08989 (2025).

[2] Li, Qingbin, et al. "Cure: Critical-token-guided re-concatenation for entropy-collapse prevention." arXiv preprint arXiv:2508.11016 (2025).

[3] Cui, Ganqu, et al. "The entropy mechanism of reinforcement learning for reasoning language models." arXiv preprint arXiv:2505.22617 (2025).

[4] Cheng, Daixuan, et al. "Reasoning with exploration: An entropy perspective." arXiv preprint arXiv:2506.14758 (2025).

---

> ### Author Response · Authors · 2025-11-21
> **Thank you for your valuable feedback!**
>
> We appreciate that Reviewer Snxh recognizes the elegance of our method and its significant performance gains. We summarize that your concerns primarily focus on **the computational overhead of SvS, comparisons and potential combinations with entropy-regulation methods, and comparisons with other data-augmentation strategies.** We address each of these points through additional experiments and clarifications below, and we sincerely thank you for your time and careful evaluation.
>
> ### Response the clear analysis of SvS’s computational overhead (W1):
>
> **Experiments:** We report the **iteration step time**—including response generation, answer verification, old log-probability recomputation, and policy updating—**as well as the isolated policy-update time** in the following figure, using experiments from our 32B models trained on both the MATH-12k and DAPO-17k datasets. Additionally, we report the number of variational problems and their corresponding responses throughout training. For the resource usage, **all our experiments are conducted on 32 H100 GPUs. We added Section 5.5 to address this topic in our revision submission, with the corresponding experiments illustrated in Figure 9.**
>
>
> When training on the challenging DAPO-17k dataset, **the initial stages exhibit a large number of synthetic variational problems with responses** because the policy’s accuracy on most problems lies between 0.125 and 0.500. **As training progresses, the model’s performance improves, with accuracies gradually surpassing 0.5, leading to a rapid decline in synthetic generation.** Consequently, both the iteration time and policy-update time follow the same pattern: **SvS incurs higher costs initially, but the iteration time decreases as training proceeds.**
>
> However, when training on the MATH-12k dataset, the Qwen-32B model already achieves an initial accuracy of roughly 80% on the training set. As a result, the number of synthetic variational problems is limited, and augmentation focuses only on the most challenging instances. Consequently, the SvS training time is comparable to that of standard RLVR, with the primary overhead likely stemming from the longer inference sequences produced by the deeper reasoning encouraged during SvS training.
>
> **Clarifications**: Based on these observations in Figure 9, **we conclude that to fully elicit a model’s reasoning capabilities, training on more challenging benchmarks benefits significantly from SvS, which requires additional time but yields substantially better performance. Conversely, for simpler tasks, SvS requires only minimal augmentation, resulting in training times comparable to standard RLVR while still providing performance improvements.**
>
> **Clarifications**: We further argue that the additional time required by SvS is well-justified by comparing with other data-augmentation strategies. **We evaluated alternative augmentation strategies that use the same or a similar number of rollouts as RLVR in Table 3 of our initial submission (also Figure. 7 in the resubmission),** such as extending RLVR training or augmenting rollouts for challenging problems. None of these approaches matched the performance of SvS, indicating that its advantage does not stem merely from increased computation but from its augmentation design and online self-play synthesis strategy.

---

> ### Author Response · Authors · 2025-11-21
>
> ### Response to the comparison with entropy-regulation methods (CURE) and other augmentation methods (SwS) (W2, Q2):
>
> **Clarifications:** In comparison with SwS [1], although both approaches aim to enhance the model’s capability on challenging problems, **SwS relies on a series of external instruction and reasoning models for generating new problems and labeling answers, and it suffers from two critical limitations:** the labeled answers are not guaranteed to be correct, and the augmentation process is entirely offline, while in SvS the reference answers for synthetic problems remain identical to the originals and are therefore fixed and ensured. Besides, as you mentioned that our approach "does not rely on external models or datasets for data augmentation", **it relies solely on the policy model itself for self-evolving, providing a much cleaner setting without external assistance. This approach is applicable even when training the strongest models where no superior external guidance exists.** Moreover, online augmentation ensures sustained diversity in generated samples and prevents the model from memorizing previous correct rollouts to secure high rewards in later iterations.
>
> **Experiments:** To demonstrate its superiority over SwS, we compare 32B models trained from Qwen2.5-32B using both methods. On average, SvS-32B attains comparable overall performance across general reasoning benchmarks, with slightly better results on AIME-level tasks. However, **for the Avg@32 and Pass@32 metrics on challenging problems—especially Pass@32, which better reflects the upper bound of reasoning capability—SvS outperforms SwS by an absolute margin of 6.2%.** This advantage stems from online augmentation, which drives the model to continually explore new reasoning strategies for newly generated problems rather than obtaining high rewards through memorization of previous correct responses, a behavior that harms exploration.
>
> | Model | GSM8K | MATH-500 | Minerva Math | Olympiad-Bench | GaoKao-2023 | AMC23 | AIME24 | AIME25 | Beyond-AIME | Avg |
> | --- | --- | --- | --- | --- | --- | --- | --- | --- | --- | --- |
> | SwS-32B | **96.3** | **89.4** | 47.1 | **60.5** | **80.3** | **90.0** | 43.3 | 40.0 | 14.0 | 62.3 |
> | SvS-32B-D25k | 95.2 | 88.6 | **47.8** | 59.9 | 79.2 | 87.5 | **50.0** | 40.0 | **17.0** | 62.8 |
>
> | Model | AIME24 | AIME25 | BAIME | Math24o | OlymE | OlymH | Avg-Pass@1 | AIME24 | AIME25 | BAIME | Math24o | OlymE | OlymH | Avg-Pass@32 |
> | --- | --- | --- | --- | --- | --- | --- | --- | --- | --- | --- | --- | --- | --- | --- |
> | SwS-32B | 31.4 | 25.1 | 14.4 | **45.4** | 19.9 | **4.2** | 23.4 | 66.7 | 53.3 | 40.0 | 81.0 | 57.0 | 23.0 | 53.5 |
> | SvS-32B-D25k | **32.4** | **31.0** | **16.1** | 44.9 | **25.6** | 2.7 | 25.4 | **73.3** | **60.0** | **45.0** | **81.0** | **70.0** | **29.0** | **59.7** |
>
>
> **Clarifications:** For comparison with Cure [2], it offers an effective mechanism for promoting exploration through high-entropy token expansion, which also helps prevent entropy collapse. We believe that SvS could be integrated with this approach, as SvS's primary goal is to enhance model diversity through training data rather than through entropy-based mechanisms. However, because our method and CURE were concurrent works and our computational resources are limited, we were unable to include new experiments within the available timeframe. **We have added the conceptual discussion to the introduction and related work sections of our updated submission.** Instead, to investigate the feasibility of integrating SvS with entropy-based methods, we additionally experiment with combining SvS and Clip-Cov [3], and the details are provided in the next response.

---

> ### Author Response · Authors · 2025-11-21
>
> ### Response to the combination of SvS with other entropy-regulation methods (Q1):
>
> **Clarifications:** Yes, we believe that **SvS can be seamlessly combined with explicit entropy-regulation methods, as its primary benefit arises from data augmentation through online generation of variants for problems that the model finds challenging.** Since the training set challenging problem generates different augmented variants across epochs based on the model’s current responses, this online augmentation prevents the model from memorizing standard training answers, encourages continuous exploration of new solution strategies, and helps maintain the model’s entropy within a stable range. With additional explicit entropy regulation, the policy may further explore richer and more diverse solution strategies for each augmented problem. However, we are unable to discuss or experiment with all related approaches, and therefore adopt the widely used Clip-Cov method referenced in [3] for our experiments.
>
> **Experiments:** We conduct experiments incorporating Clip-Cov into SvS using the LLaMA-3.1-8B-Instruct model on the MATH-12k training set. The Clip-Cov parameters follow the default settings in the original paper, with a clip ratio $r = 2 \times 10^{-4}$, $\omega_{\text{low}} = 1$, and $\omega_{\text{high}} = 5$. The hyperparameters regarding the SvS augmentation are fixed as the settings in our initial submission. We evaluate intermediate results on six benchmarks: GSM8k, MATH-500, Minerva-Math, Olympiad-Bench, Gaokao-2023, and AMC-23, along with their average scores. **We added Appendix G to address this topic in our revision submission, with the corresponding experiments illustrated in Figure 14 and 15.** As shown, **incorporating SvS augmentation consistently improves policy performance over the Clip-CoV baseline, demonstrating that entropy-regulation methods can be further enhanced through integration with SvS.**
>
>
> [1] Liang, Xiao, et al. "SwS: Self-aware Weakness-driven Problem Synthesis in Reinforcement Learning for LLM Reasoning." arXiv preprint arXiv:2506.08989 (2025).
>
> [2] Li, Qingbin, et al. "Cure: Critical-token-guided re-concatenation for entropy-collapse prevention." arXiv preprint arXiv:2508.11016 (2025).
>
> [3] Cui, Ganqu, et al. "The entropy mechanism of reinforcement learning for reasoning language models." arXiv preprint arXiv:2505.22617 (2025).
>
> [4] Cheng, Daixuan, et al. "Reasoning with exploration: An entropy perspective." arXiv preprint arXiv:2506.14758 (2025).

---

> > ### Comment · Reviewer_Snxh · 2025-11-23
> >
> > Thank you for your response. The additional clarifications and experiments have addressed most of my concerns. I will maintain my positive score. On a minor note, I don't like the emoji icons used; they give me the impression of style over substance.

---

> > > ### Author Response · Authors · 2025-11-23
> > >
> > > Thank you for your response and for maintaining a positive assessment of our paper. Regarding the emojis in our rebuttal, our intention was to help reviewers distinguish which paragraphs contained new experiments and which provided clarifications. We apologize for any unfavorable impression or discomfort they may have caused while reading the response, and we have removed them for the potential that you wish to revisit the rebuttal in the future.

---

### Author Response · Authors · 2025-11-23
**Summary of our Contributions and Revisions**

We sincerely thank you to all reviewers for their constructive feedback, which has greatly helped us improve the quality and comprehensiveness of our work. We truly appreciate the time and effort you devoted to the review process, and we genuinely consider that your insights have substantially contributed to a better understanding of SvS for both us and the broader community. Below, we provide a brief overview of (1) the key contributions and strengths of our paper, as highlighted by the reviewers, and (2) the revisions incorporated in our resubmission.

## Summary of Strengths and Contributions

We propose a simple online Self-Play with Variational Problem Synthesis strategy (SvS) for RLVR training that iteratively leverages model responses to generate synthetic problems for augmentation. This approach effectively sustains RLVR training by **maintaining stable policy entropy and generation diversity**, and it achieves substantial improvements over the baseline in **mathematical reasoning (18.3% and 22.8% Pass@32 on AIME 24 & 25)** and **code generation during the rebuttal (average +7% in early-stage).**

+ The SvS method is **simple and general**, and its formulation is clean: the framework is fully self-contained, relying on neither external models nor additional datasets. (From Reviewer Snxh, rsiR, kCjs)

+ The SvS framework **mitigates the entropy collapse** issue in RLVR, while its **problem selection and filtering strategy remains effective** for eliciting stronger reasoning capabilities from the policy. (From Reviewer kCjs, zAc9)

+ Our **experiments demonstrate the method's strong performance.** The evaluation is thorough and well supported by solid empirical results, while the paper also includes several fully fair baselines to clarify the effectiveness of SvS. (From Reviewer Snxh, yXhe, rsiR, zAc9).

+ Our paper is **well written**, clearly detailed, and easy to follow. (From Reviewer rsiR, zAc9).


## Revisions Incorporated in the Resubmission

We have added detailed descriptions, conceptual discussions, and additional experiments that the reviewers expressed interest in and that we believe improve the comprehensiveness of the paper in the resubmission. In the revised version, the revised text in the main artical are **labled in blue color**, while we add the section of F.5, G, H in the Appendix. Specifically, these modifications focus on:

+ Section 5.3: We added the **intermediate AIME evaluation for all three baselines with fair-rollout budgets** and SvS (as Table 3 in the initial submission), presented as Figure 7.

+ Section 5.4: We extended the SvS augmentation strategy to the **code generation** domain beyond the mathematical reasoning setting used in our initial submission, and the **results also appear promising**, demonstrating the generalizability and robustness of our method.

+ Section 5.5: We provided a **computational overhead analysis of SvS** compared with the baselines, along with the specific rollout budgets for synthetic problems and their corresponding responses throughout training.

+ Appendix F.5: We further **analyzed the genuine logical correctness of the synthetic problems** using human inspection and statistics from state-of-the-art LLMs.

+ Appendix G: We compared **SvS with existing entropy-regulation methods and attempted to combine them.** The results demonstrated the superiority, generalizability, and robustness of the SvS augmentation strategy.

+ Appendix H: We present our **initial failed attempt during the development of SvS,** which could help readers better understand the rationale behind the reward assignment strategy used for the synthetic problem generation task.

Finally, we would like to once more express our sincere gratitude to all the reviewers for their insightful comments, and we warmly welcome any further discussion that could help address remaining concerns and potentially improve our work.

---

### Author Response · Authors · 2025-12-03
**A Summary of Discussion by Authors**

Dear Reviewers and ACs:

Thank you all for your efforts during this challenging time for the ICLR community. We are grateful to the reviewers for their thoughtful feedback, which has helped improve our work. During the rebuttal period, **three of the five reviewers maintained their positive assessments (8,8,6), and one reviewer raised their initial score (4 to 6).** We conducted extensive additional experiments, carried out further analysis, and provided detailed clarifications to strengthen the paper. In the revised manuscript, we summarized all new experimental results in Sections 5.3–5.5 and Appendices F–H, which we believe address the majority of the reviewers’ concerns.

Here we summarize our discussions with the reviewers and track the score changes below:

### Reviewer Snxh, kept the score of 6, November 22, 2025 at 23:00.

+ **W1.** During the rebuttal, we additionally provide Section 5.5 and Figure 9 in our revision for analyzing the computational cost of SvS on 32B models (MATH-12k, DAPO-17k, 32 H100 GPUs). Overall, SvS incurs additional cost primarily on challenging tasks, while delivering substantially improved performance. Compute-matched baselines (Table 3 / Figure 7) still underperform SvS, confirming that its gains come from the proposed augmentation strategy.

+ **W2,Q2.** As suggested by the reviewer, we compared our method with SwS, which relies on external models for problem generation and answer labeling, whereas our approach guarantees answer correctness and using only the policy itself. **On 32B models, SvS significantly outperforms SwS on challenging tasks under Pass@32 by an absolute margin of 6.2%.** With respect to CURE, which promotes exploration through entropy regularization, we discuss its relationship to our method conceptually in the revised submission.

+ **Q1.** Based on the new experiments in Appendix G, we find that SvS is complementary to explicit entropy-regularization methods: SvS improves exploration through online data augmentation, while entropy-based methods promote diversity at the policy level. Experiments combining SvS with Clip-Cov consistently outperform Clip-Cov alone, demonstrating that SvS further enhances entropy-regularized training.



### Reviewer yXhe, currently did not respond.

+ **W1,Q1.** We clarified that SvS introduces a novel online self-play framework that generates problems from model responses and guarantees answer correctness without external supervision, a setting not explored in prior RLVR work. Empirically, SvS consistently surpasses the Pass@k performance of standard RLVR on large-scale models and multiple datasets, demonstrating substantial methodological and practical significance.

+ **Q2.** We clarified that SvS focuses augmentation on challenging problems, as augmenting easy ones quickly saturates and leads to overfitting. Besides, to ensure synthetic problems are both valid and useful, we reward only those with intermediate policy accuracy [1/8,5/8], which prevents trivial, hint-based exploitation and enables stable and effective training.

+ **Q3.** We compared SvS with Absolute Zero, which relies on a verifiable code-execution environment to generate RLVR training data. Empirical comparisons via a shared baseline (SwS) show that SvS outperforms SwS at 32B while SwS substantially outperforms Absolute Zero at 7B, providing indirect but consistent evidence that SvS is stronger in reasoning tasks.

+ **Q4.** We clarified that Figure 6 reports Pass@k as a function of k (x-axis), rather than the number of training samples. We also clarified that our paper already includes comparisons with compute-matched baselines and we additionally provides intermediate evaluations in Figure 7.

+ **Q5.** We clarified that our work is motivated by improving broad Pass@k performance rather than Pass@1, and further explained that improvements of Pass@k at large k are a prerequisite for sustaining long-term gains in Pass@1.

---

> ### Author Response · Authors · 2025-12-03
>
> ### Reviewer rsiR, kept the score of 8, November 28, 2025 at 01:21.
>
> + **W1.** We clarified that SvS’s performance degradation on OOD benchmarks is largely driven by the answer-format distribution of the training data and by model capacity. We further demonstrated that this issue can be effectively mitigated by augmenting training set with external open-ended problems, leading to substantially improved generalization across diverse benchmarks (Table 2).
>
> + **W2.** We discussed with the reviewer that both RLVR and SvS benefit substantially from stronger base models, as limited model capacity constrains rollout diversity and learning effectiveness. We further supported this claim with corroborating findings from DeepSeek-R1 report.
>
> + **Q1,Q3.** We added Section 5.5 and Figure 9 to provide a detailed analysis of SvS’s training cost on 32B models, showing that SvS incurs higher overhead primarily in the early stages on challenging datasets but that this cost decreases rapidly as the model learns to solve challenging problems in the training set. We further show that extended SvS training exhibits no performance degradation in our experiments, indicating its stable and efficient long-term behavior.
>
> + **Q2.** We demonstrated that SvS can be naturally extended beyond reasoning to other domains **by conducting additional code-generation experiments (Section 5.4 and Figure 8), where SvS consistently outperformed RLVR with an average improvement of 6.9% across four benchmarks,** highlighting its strong potential for generalization.
>
>
>
> ### Reviewer kCjs, increased their score from 4 to 6 on November 24, 2025 at 20:48.
> + **W1.** We added code-generation experiments (Section 5.4 and Figure 8) comparing SvS with the RLVR baseline, where SvS achieved substantially better performance, providing evidence of its potential to generalize beyond mathematical reasoning.
>
> + **W2.** We clarified that we incorporated compute-matched RLVR augmentation baselines and further added intermediate evaluations (Section 5.3, Table 3, Figure 7), which shows that none of them outperformed SvS. **We further compared SvS with the entropy-regularization method Clip-Cov (Appendix G, Figures 14–15) and demonstrated its consistently superior performance.**
>
> + **W3.** We added a detailed cost analysis on 32B models (Section 5.5, Figure 9) showing that SvS introduces higher overhead mainly at early stages on challenging datasets, while remaining comparable to RLVR when training on simpler problems (MATH-12k). We further demonstrated that this cost is justified, as entropy-based baselines and compute-matched alternatives still underperform SvS.
>
> + **W4,Q2.** We clarified that SvS does not rely on extensive hyperparameter tuning and a simple, fixed augmentation strategy works robustly across all experiments and can be directly adapted to code-generation tasks. We further demonstrated that augmenting challenging problems is critical, as augmenting easier ones leads to rapid saturation and overfitting (Table 3, Figure 7, Appendix H).
>
>
>
> ### Reviewer zAc9, kept the score of 8, November 23, 2025 at 21:02.
>
> + **W1,Q1.** We conducted validation of synthetic problems using SOTA LLMs and manual inspection (Appendix F.5, Figure 13), showing that over 80% of generated problems are judged as valid. We further clarified that SvS does not aim to perfectly curate problems but to generate policy-adaptive, challenging instances that effectively improve downstream performance.
>
> + **W2,Q2.** We clarified that SvS can directly generalize to domains with reliable verifiers or reward models and added code-generation experiments as a representative case (Section 5.4, Figure 8), where SvS consistently outperformed RLVR and exhibited more stable policy entropy.
>
> + **W3.** We added a detailed cost analysis on 32B models (Section 5.5, Figure 9), showing that SvS incurs higher overhead mainly in early stages on challenging datasets but remains comparable to RLVR on easier tasks. We further demonstrated that this cost is well justified, as compute-matched alternatives still underperform SvS.
>
> We hope this summary of our rebuttal discussions with the reviewers is helpful in your evaluation of our work.
>
> Best regards,
>
> The Authors

---

### Meta-Review · Area_Chair_czRX · 2025-12-18

**Summary:**

The paper received a mix of scores, with the majority of reviewers leaning toward a positive assessment. Notably, Reviewer kCjs—who initially gave a score of 4—indicated an intent to increase their rating to 6 after the rebuttal.

The authors have addressed the primary concerns raised during the review process. Regarding the issue of novelty highlighted by Reviewer yXhe, the concern remains somewhat subjective, particularly as the reviewer did not provide specific references or prior work to substantiate the claim. Furthermore, other points like the lack of baselines and discussions were either fully resolved or sufficiently mitigated through the detailed rebuttal responses.

**Reviewer Concerns:**

The authors conducted extensive work during the rebuttal phase, which has successfully addressed (or partially addressed) the following concerns:
- Analysis of computational costs introduced by the variational synthesis module. [Snxh, risR, kCjs, zAc9]
- Lack of discussion and comparison with closely relevant methods like SwS and CURE. [Snxh]
- Lack of novelty and insights. [yXhe]
- Lack of the justification of the filtering designs. [yXhe]
- Lack of baselines. [yXhe, kCjs]
- Lack of discussion of the relationship between pass@1 metric and data augmentation. [yXhe]
- Concern if the method generalizes to small (weak) base models. [risR]
- What’s the filtering rate of generated problem variants? [risR]
- Concern about the generalization to domains other than math reasoning. For example, improvements are inconsistent comparing math reasoning to code and general QA. [rsiR, kCjs, zAc9]
- How would metrics change with more training steps? [rsiR]
- Concern about the difficulty of hyperparameter tuning. [kCjs]
- Lack of investigation if the heuristically selected synthetic problems are logically sound. [zAc9]
- Concern if the performance gain is primarily an artifact of the experimental setup. [zAc9]
- Lack of more analysis on the synthetic problems. [zAc9]

There are currently no significant concerns that remain outstanding.

**Reviewer Scores:**

Reviewer yXhe would have considered changing the score if they they had been able to participate fully in the discussion, as some of the main concerns like missing baselines and missing discussion on filtering designs have been addressed by the authors.
All other reviewers have acknowledged that the rebuttal has mostly resolved their concerns.

---

### Decision · Program_Chairs · 2026-01-26

Accept (Poster)